# Temperature is All You Need for Generalization in Langevin Dynamics and other Markov Processes

**Itamar Harel**[*]
Technion

**Yonathan Wolanowsky**
Technion

**Gal Vardi**
Weizmann Institute of Science

**Nathan Srebro**
Toyota Technological Institute at Chicago

**Daniel Soudry**
Technion

## Abstract

We analyze the generalization gap (gap between the training and test errors) when training a potentially over-parametrized model using a Markovian stochastic training algorithm, initialized from some distribution $\boldsymbol{\theta}_0 \sim p_0$. We focus on Langevin dynamics with a positive temperature $\beta^{-1}$, i.e. gradient descent on a training loss $L$ with infinitesimal step size, perturbed with $\beta^{-1}$-variances Gaussian noise, and lightly regularized or bounded. There, we bound the generalization gap, *at any time during training*, by $\sqrt{(\beta \mathbb{E} L(\boldsymbol{\theta}_0) + \ln(1/\delta))/N}$ with probability $1 - \delta$ over the dataset, where $N$ is the sample size, and $\mathbb{E} L(\boldsymbol{\theta}_0) = O(1)$ with standard initialization scaling. In contrast to previous guarantees, we have no dependence on either training time or reliance on mixing, nor a dependence on dimensionality, gradient norms, or any other properties of the loss or model. This guarantee follows from a general analysis of any Markov process-based training that has a Gibbs-style stationary distribution. The proof is surprisingly simple, once we observe that the marginal distribution divergence from initialization remains bounded, as implied by a generalized second law of thermodynamics.

## 1 Introduction

One main goal of contemporary machine learning theory is to predict a model's behavior before training occurs. A commonly desired metric is the generalization of overparameterized models, such as neural networks (NN). For these models, such a predictive theory of generalization is still lacking, despite great empirical success [71, 23]. In particular, a significant line of work aimed to explain the role of optimization in generalization (e.g. [23, 64, 40, 66]), and specifically the effect of stochasticity (e.g. [59, 49, 10, 8]).

Data-dependent Markov processes are a common optimization approach. These include stochastic gradient descent (SGD), as well as other stochastic gradient methods either studied theoretically [30, 59], or used in practice such as SGD with momentum [52], ADAM [34], and many more. Of particular interest are continuous Langevin dynamics (CLD) and discrete analogues of it, which have been studied extensively as models for SGD (see Section 4.1).

In Section 2 we develop, for the first time, a generalization bound applicable to *any data-dependent Markov process* with a Gibbs-type stationary distribution (i.e. whose finite density exists and is nonzero w.r.t. some data-independent base measure). An important feature of our analysis is that it is *entirely independent of the training time $t$*, both in that we do not rely on training for only a

---

[*]Corresponding author: `itamarharel01@gmail.com`

39th Conference on Neural Information Processing Systems (NeurIPS 2025).

small number of steps, nor that we rely on mixing — the guarantees are valid at any time, with no dependence at all on $t$. Furthermore, it is also completely trajectory independent.

In Section 3 we apply these general results to the particular case where training is done with CLD with loss $L$ and inverse temperature $\beta$, deriving a particularly simple generalization bound for CLD, which we compare to previous generalization bounds for CLD in Section 4, as well as discussing other related work. Finally, we address limitations and future work in Section 5.

To prove these results, we first show in Section 2 how, for the marginal distribution at time $t$, $p_t$, its divergence (either KL or the Rényi infinity divergence) from initialization is bounded due to its monotonicity, i.e. a generalized second law of thermodynamics [11, 46]. This surprisingly simple derivation[2] leads to our key technical result (Corollary 2.5). Standard PAC-Bayes generalization bounds [43] then yield our generalization bounds (Theorem 2.7 and Corollary 3.1).

## 2   Generalization Bounds for General Markov Process

In this Section, we consider general data-dependent Markov processes over predictors and obtain a bound on their generalization gap. Importantly, although the bound only depends on the initialization distribution and a stationary distribution, it will apply to predictors at any time $t \geq 0$ along the Markov process. Our main goal is to apply these bounds to stochastic training methods, such as Langevin dynamics, where the iterates form a data-dependent Markov process. But to emphasize the broad generality of the results, in this section we consider a generic stochastic optimization framework and general data-dependent Markov processes.

We obtain generalization guarantees by bounding the KL-divergence (or, for high probability bounds, the Rényi infinity divergence, see Definition 2.1) between the data-dependent marginal distribution $p_t$ of the predictors at time $t$, and some data-independent base measure $\nu$ (the PAC-Bayes "prior"). The crux of the analysis is therefore bounding the divergence between $p_t$ and $\nu$, based only on assumptions on the initial distribution $p_0$ (specifically, the divergence between $p_0$ and $\nu$) and a stationary distribution $p_\infty$ (specifically, requiring that $p_\infty$ can be expressed as a Gibbs distribution with bounded potential or expected potential, see Definition 2.2) — we do this in Section 2.1. Then, in Section 2.2 we plug these bounds on the divergence between $p_t$ and $\nu$ into standard PAC-Bayes bounds to obtain the desired generalization guarantees.

Detailed proofs of all the results in this section can be found in Appendix B.

### 2.1   Bounding the Divergence of a Markov Process

In this subsection, we consider a general time-invariant Markov process[3] $h_t \in \mathcal{H}$ over a state space $\mathcal{H}$. The Markov process can be either in discrete or continuous time, i.e. we can think of $t$ as either an integer or a real index. We denote by $p_t$ the marginal distribution at time $t$, i.e. $h_t \sim p_t$. We do *not* assume that the Markov process is ergodic, and all our results will rely on the existence of *some* stationary distribution $p_\infty$. The main goal of this subsection is to bound the divergence $D\left(p_t \,\|\, \nu\right)$ between the marginal distribution at time $t$ and some reference distribution $\nu$. We can think of a bound on the divergence as ensuring high entropy relative to $\nu$, or in other words that $p_t$ does not concentrate too much relative to $\nu$, i.e. does not have too much probability mass in a small $\nu$-region. We present all bounds for both the KL-divergence $\mathrm{KL}\left(p \,\|\, q\right)$ and the Rényi infinity divergence $D_\infty\left(p \,\|\, q\right)$, defined below.

**Divergences and Gibbs distributions.** We recall the definitions of our two divergences, and also relate them to the Gibbs distribution. It will also be convenient for us to introduce "relative" versions of divergences.

---

[2]e.g. to bound the KL divergence of a Markov process having a stationary distribution with potential $\Psi \in [0, \infty)$, i.e. $\mathrm{d}p_\infty / \mathrm{d}p_0 \propto e^{-\Psi}$ (e.g., $\Psi = \beta L$ for CLD), the second law implies the first inequality below

$$\mathrm{KL}(p_t \| p_0) = \int p_t \ln \frac{p_t}{p_0} = \int p_t \ln \frac{p_t}{p_\infty} + \int p_t \ln \frac{p_\infty}{p_0} \leq \int p_0 \ln \frac{p_0}{p_\infty} + \int p_t \ln \frac{p_\infty}{p_0} = E_{p_0} \Psi - E_{p_t} \Psi \leq E_{p_0} \Psi.$$

[3]Formally stated: we require that for any $0 \leq t_1 < t_2 < t_3$ we have that $h_{t_3}$ is independent of $h_{t_1}$ conditioned on $h_{t_2}$ (Markov property) and that for any $0 \leq t_1, t_2, \Delta$ we have that $h_{t_1+\Delta}|h_{t_1}$ has the same conditional distribution as $h_{t_2+\Delta}|h_{t_2}$ (time-invariance).

**Definition 2.1** (Divergences [4]). For probability distributions $p, q$ and $\mu$:

1. The $\mu$-**weighted Kullback-Leibler (KL) divergence** (a.k.a. relative cross-entropy) is[5] $\mathrm{KL}_\mu(p \,\|\, q) = \int \mathrm{d}\mu \ln \frac{\mathrm{d}p}{\mathrm{d}q}$, and the **KL-divergence** is then $\mathrm{KL}(p \,\|\, q) = \mathrm{KL}_p(p \,\|\, q)$.

2. The **Rényi infinity divergence** is[6] $D_\infty^\mu(p \,\|\, q) = \mathrm{ess\,sup}_\mu \ln \frac{\mathrm{d}p}{\mathrm{d}q}$, with $D_\infty(p \,\|\, q) = D_\infty^p(p \,\|\, q)$.

**Definition 2.2** (Gibbs distribution). A distribution $p$ is **Gibbs** w.r.t. a **base distribution** $q$ with **potential** $\Psi : \mathcal{H} \to \mathbb{R}$ if $Z = \int e^{-\Psi} \, \mathrm{d}q < \infty$ and

$$\mathrm{d}p = Z^{-1} e^{-\Psi} \, \mathrm{d}q \,.$$

**Claim 2.3.** *If $p, q, \mu, \nu$ are probability measures, and $p$ is Gibbs w.r.t. $q$ with potential $\Psi < \infty$, then*

*1. $\mathrm{KL}_\mu(p \,\|\, q) + \mathrm{KL}_\nu(q \,\|\, p) = \mathbb{E}_\nu \Psi - \mathbb{E}_\mu \Psi$,*

*2. $D_\infty^\mu(p \,\|\, q) + D_\infty^\nu(q \,\|\, p) = \mathrm{ess\,sup}_\nu \Psi - \mathrm{ess\,inf}_\mu \Psi$.*

*So, $\mathrm{KL}(p \,\|\, q) + \mathrm{KL}(q \,\|\, p) = \mathbb{E}_q \Psi - \mathbb{E}_p \Psi$, and $D_\infty(p \,\|\, q) + D_\infty(q \,\|\, p) = \mathrm{ess\,sup}_q \Psi - \mathrm{ess\,inf}_p \Psi$.*

That is, the potential of a Gibbs distribution $p$ allows us to bound the divergence *in both directions* between $p$ and the base measure $q$. A generalized converse of Claim 2.3 also holds, and we have that bounding on the *symmetrized* divergences (but not just on one direction!) is also sufficient for $p$ being Gibbs with a bounded potential.[7]

**Second Law of Thermodynamics.** Central to our analysis is the following monotonicity property on the divergence between the marginal distribution of a Markov process and *any* stationary distribution.

**Claim 2.4** (Cover's Second Law of Thermodynamics). *Let $p_t$ be the marginal distribution of a time-invariant Markov process, and $p_\infty$ a stationary distribution for the transitions of the Markov process (the process need not be ergodic, and $p_t$ need not converge to $p_\infty$). Then for any $t \geq 0$*

$$\mathrm{KL}(p_t \,\|\, p_\infty) \leq \mathrm{KL}(p_0 \,\|\, p_\infty) \qquad and \qquad D_\infty(p_t \,\|\, p_\infty) \leq D_\infty(p_0 \,\|\, p_\infty) \,.$$

When the stationary distribution is uniform (thus having maximal entropy), the KL-form of Claim 2.4 recovers the familiar second law of thermodynamics, i.e. that the entropy is monotonically non-decreasing. The more general form, as in Claim 2.4, is a direct consequence of the data processing inequality, as pointed out by Theorem 4 of Cover [11] (see also [12, 46]) and the generalization to Rényi divergences in [65, Theorem 9 and Example 2] —for completeness we provide a proof in Appendix A.2).

In our case, the stationary distribution $p_\infty$ will not be uniform, but rather will be very data-dependent (we are interested mostly in processes that aim to optimize some data-dependent quantity, such as Langevin dynamics). Nevertheless, we do want to use Claim 2.4 to control the entropy of $p_t$ relative to some benign data-independent base distribution $\nu$ (which we can informally think of as "uniform"). To do so, we can use the chain rule and plug in Claim 2.4 to obtain that for any distribution $\nu$ and at any time $t$ we have (see Lemma B.1 in Appendix B for the full derivation):

$$\mathrm{KL}(p_t \,\|\, \nu) = \mathrm{KL}(p_t \,\|\, p_\infty) + \mathrm{KL}_{p_t}(p_\infty \,\|\, \nu) \leq \mathrm{KL}(p_0 \,\|\, p_\infty) + \mathrm{KL}_{p_t}(p_\infty \,\|\, \nu)$$
$$= \mathrm{KL}(p_0 \,\|\, \nu) + \mathrm{KL}_{p_0}(\nu \,\|\, p_\infty) + \mathrm{KL}_{p_t}(p_\infty \,\|\, \nu) \,, \quad (1)$$

and similarly,

$$D_\infty(p_t \,\|\, \nu) \leq D_\infty(p_0 \,\|\, \nu) + D_\infty^{p_0}(\nu \,\|\, p_\infty) + D_\infty^{p_t}(p_\infty \,\|\, \nu) \,. \quad (2)$$

---

[4] The term "divergence" is a slight abuse of notation, as the following definitions are not strictly non-negative, without specifying $\mu$.

[5] For two measures $p$ and $q$, $\mathrm{d}p/\mathrm{d}q$ is the Radon-Nikodym derivative (i.e. the density of $p$ w.r.t. $q$) when it exists (i.e. when $p \ll q$, i.e. $p$ is absolutely continuous w.r.t. $q$), or $\infty$ otherwise.

[6] The essential supremum of a function $f$ w.r.t. a measure $\mu$ is $\mathrm{ess\,sup}_\mu f = \inf \{b \in \mathbb{R} \mid \mu(f > b) = 0\}$, i.e. the smallest (infimum) number that bounds $f$ from above almost everywhere. The essential infimum is defined similarly.

[7] More formally: $\mathrm{KL}(p \,\|\, q) + \mathrm{KL}(q \,\|\, p) \leq \beta$ iff there exists a potential $\Psi$ such that $p$ is Gibbs w.r.t. $q$ with potential $\Psi$ and $\mathbb{E}_q \Psi - \mathbb{E}_p \Psi \leq \beta$, and similarly $D_\infty(p \,\|\, q) + D_\infty(q \,\|\, p) \leq \beta$ iff there exists a potential $0 \leq \Psi \leq \beta$ such that $p$ is Gibbs w.r.t. $q$ with potential $\Psi$. See Claim B.8 for a proof.

Bounding the last two terms in (1) and (2) using Claim 2.3 we obtain the main result of this subsection:

---

**Corollary 2.5.** *For any distribution $\nu$ and any time-invariant Markov process, and any stationary distribution $p_\infty$ that is Gibbs w.r.t. $\nu$ with potential $\Psi \geq 0$ (the Markov chain need not be ergodic, and need not converge to $p_\infty$), at any time $t \geq 0$:*

$$\mathrm{KL}\left(p_t \,\|\, \nu\right) \leq \mathrm{KL}\left(p_0 \,\|\, \nu\right) + \mathbb{E}_{p_0}\Psi - \mathbb{E}_{p_t}\Psi \leq \mathrm{KL}\left(p_0 \,\|\, \nu\right) + \mathbb{E}_{p_0}\Psi \tag{3}$$

$$D_\infty\left(p_t \,\|\, \nu\right) \leq D_\infty\left(p_0 \,\|\, \nu\right) + \operatorname{ess\,sup}_{p_0}\Psi \tag{4}$$

---

The important feature of Corollary 2.5 is that it bounds the divergence *at any time* $t$, in terms of a right-hand side that depends only on the initial distribution $p_0$ and a stationary distribution $p_\infty$. Interpreting the divergence $D\left(p_t \,\|\, \nu\right)$ as a measure of concentration, the Corollary ensures that at no point during its run, and regardless of mixing, does the Markov process concentrate too much, and it always maintains high entropy (relative to the base measure $\nu$).

*Remark* 2.6. In order to bound the divergence $D\left(p_t \,\|\, \nu\right)$ at finite time $t$, it is not enough to rely only on the divergences $D\left(p_0 \,\|\, \nu\right)$ and $D\left(p_\infty \,\|\, \nu\right)$ from the initial and stationary distributions, and it is necessary to rely also on the reverse divergence $D\left(\nu \,\|\, p_\infty\right)$ — see Appendix C.

## 2.2 From Divergences to Generalization

Corollary 2.5 can be directly used to obtain PAC-Bayes type generalization guarantees. Specifically, we consider a **generic stochastic optimization setting** specified by a bounded instantaneous objective $f : \mathcal{H} \times \mathcal{Z} \to [0, 1]$ over a class $\mathcal{H}$, which we will refer to as the "predictor" class, and instance domain $\mathcal{Z}$. For example, in supervised learning $\mathcal{Z} = \mathcal{X} \times \mathcal{Y}$, $\mathcal{H} \subseteq \mathcal{Y}^{\mathcal{X}}$ and $f(h, (x, y)) = \mathbb{I}\{h(x) \neq y\}$ measures the error of predicting $h(x)$ when the correct label is $y$. For a source distribution $D$ over $\mathcal{Z}$ and data $S \sim D^N$ of size $N$ we would like to relate the population and empirical objectives

$$E_D\left(h\right) = \mathbb{E}_{z \sim D}[f(h, z)] \qquad E_S\left(h\right) = \frac{1}{N} \sum_{z \in S} f(h, z). \tag{5}$$

In our case, we are interested in predictors generated by a **data-dependent Markov process** $h_t$. That is, conditioned on the data $S$, $\{h_t\}_{t \geq 0}$ is a time-invariant Markov process, specified by some (possibly data-dependent) initial distribution $p_0(h_0; S)$, and a transition distribution that would also depend on the data $S$, and specifies a (randomized) rule for generating the next iterate $h_{t+1}$ (if in discrete time) from the current iterate $h_t$ and the data $S$ (as in, e.g., stochastic gradient descent or stochastic gradient Langevin dynamics; SGLD).

We present two types of generalization guarantees: guarantees that hold *in expectation* over a draw from the Markov process ((6) below) and guarantees that hold *with high probability* over a single draw from the Markov process (as in (7), e.g. a single run of CLD). In both cases, the guarantees hold with high probability over the training set.

---

**Theorem 2.7.** *Consider any distribution $D$ over $\mathcal{Z}$, function $f : \mathcal{H} \times \mathcal{Z} \to [0, 1]$, sample size $N \geq 8$, and any distribution $\nu$ over $\mathcal{H}$. Let $\{h_t \in \mathcal{H}\}_{t \geq 0}$ be a discrete or continuous time process (i.e. $t \in \mathbb{Z}_+$ or $t \in \mathbb{R}_+$) that is time-invariant Markov conditioned on $S$, that starts from an initial distribution $p_0(\cdot; S)$ (that may depend on $S$), and admits a stationary distribution conditioned on $S$, $p_\infty(\cdot; S)$. Let $\Psi_S(h) \geq 0$ be a non-negative potential function and assume that $p_\infty(\cdot; S)$ is Gibbs w.r.t. $\nu$ with potential $\Psi_S$. Then:*

*1. with probability $1 - \delta$ over $S \sim D^N$,*

$$\mathbb{E}\left[E_D(h_t) - E_S(h_t)|S\right] \leq \sqrt{\frac{\mathrm{KL}\left(p_0(\cdot; S) \,\|\, \nu\right) + \mathbb{E}\left[\Psi_S(h_0)|S\right] + \ln^{N}/\delta}{2N}}, \tag{6}$$

*2. with probability $1 - \delta$ over $S \sim D^N$ and over $h_t$:*

$$E_D(h_t) - E_S(h_t) \leq \sqrt{\frac{D_\infty\left(p_0(\cdot; S) \,\|\, \nu\right) + \operatorname{ess\,sup}_{h \sim p_0(\cdot; S)} \Psi_S(h) + \ln^{N}/\delta}{2N}}. \tag{7}$$

---

*Proof.* The Theorem follows immediately by plugging the divergence bounds of Corollary 2.5 into standard PAC-Bayes guarantees, which we do in Appendix B. $\qquad\square$

*Remark* 2.8. A simplified variant of Theorem 2.7 can be stated when the initial distribution $p_0$ is data-independent and always equal to $\nu$. In this case the divergence between $p_0$ and $\nu$ vanishes, and (6) and (7) become

$$\mathbb{E}\left[E_D(h_t) - E_S(h_t)|S\right] \leq \sqrt{\frac{\mathbb{E}_{p_0}\left[\Psi_S \mid S\right] + \ln{N/\delta}}{2N}}, \ E_D(h_t) - E_S(h_t) \leq \sqrt{\frac{\operatorname{ess\,sup}_{p_0} \Psi_S + \ln{N/\delta}}{2N}}.$$

(8)

But allowing $p_0 \neq \nu$ is more general, as it both allows using a data-dependent initialization (recall that $\nu$ must be data independent) and it allows initializing to a distribution where $D\left(p_\infty \,\|\, p_0\right)$ is infinite — e.g., we can allow initializing to a degenerate initial distribution $p_0$ whose support is a strict subset of the support of $p_\infty$ (in which case $p_\infty$ will definitely *not* be Gibbs w.r.t. $p_0$), as long as the $\nu$-mass of the support of $p_0$ is not too small.

*Remark* 2.9. In Theorem 2.7, the Markov process need not be ergodic, and need not converge to $p_\infty$, or converge at all. If there are multiple stationary distributions, the theorem holds for all of them, and so we can take $p_\infty$ to be any stationary distribution we want. And in any case, there is no mixing requirement, and the theorem holds at any time $t$.

*Remark* 2.10. Our data-dependent Markov process of interest, and in particular CLD and SGD, might aim to minimize $E_S(h_t)$, and the potential $\Psi$ might also be related to it (as in, e.g., CLD). This is allowed, but is in no way required in Theorem 2.7. Even for CLD, these might be related but not the same, as we might be minimizing a surrogate loss, such as a logistic loss, but are interested in bounding the generalization gap for a zero-one error. In stating Theorem 2.7 we intentionally refer to an arbitrary stochastic optimization problem and an arbitrary data-dependent Markov process, that are allowed to be related or dependent in arbitrary ways.

*Remark* 2.11. In Appendix C we show that in order to ensure generalization at every intermediate $t$, it is not sufficient to only bound $\mathrm{KL}\left(p_\infty \,\|\, \nu\right)$ or $D_\infty\left(p_\infty \,\|\, \nu\right)$, and we do need the stronger symmetric bound ensured by the Gibbs potential and Claim 2.3; and that it is also necessary to relate both $p_0$ and $p_\infty$ to the *same* data independent distribution $\nu$, as relating them to different data-independent distributions ensures generalization at the beginning and at the end, but not the middle of training.

*Remark* 2.12. In Theorem 2.7 we plugged Corollary 2.5 into a simplified PAC-Bayes bound that allows for easy interpretation and comparison with other results. But once we have the divergence bounds of Corollary 2.5, we can just as easily plug them into tighter PAC-Bayes bounds — see Appendix B. For example, when $E_S\left(h_t\right) \approx 0$, these yield a rate of $O\left(1/N\right)$.

## 3 Special Case: Continuous Langevin Dynamics

Clearly, given Theorem 2.7 all we need to do in order to derive explicit generalization bounds for *any* Markovian training procedure, is to find a stationary distribution, and bound its potential (or its expectation at $p_0$). In this section, we will exemplify our results in a few special cases of continuous-time Langevin dynamics (CLD), a commonly studied approximation for NN training with "infinitesimal learning rate" (e.g. [41], see Section 4.1 for additional references), which have a normalized stationary distribution that we can write analytically.

**Additional notation.** In the following, it will be convenient to consider a parametric model. Specifically, we assume that there exists some parameter space $\Theta \subseteq \mathbb{R}^d$ that parameterizes a hypothesis class $\mathcal{H} \subseteq \mathcal{Y}^{\mathcal{X}}$ via a mapping $\Theta \ni \boldsymbol{\theta} \mapsto h_{\boldsymbol{\theta}} \in \mathcal{H}$, and assume Markovian dynamics *in parameter space*, instead of in the hypothesis space (note that Markov processes in parameter space may not be Markovian in hypothesis space, but the same generalization results apply ). We shall also use, with some abuse of notation, $\varphi\left(\boldsymbol{\theta}\right) = \varphi\left(h_{\boldsymbol{\theta}}\right)$ for any data-dependent or data-independent function $\varphi$ over hypotheses, e.g. a training loss/objective $L_S$ w.r.t a training set $S$. Finally, we use $\mathcal{C}^2$ to denote the space of twice continuously differentiable functions on $\Theta$.

**CLD in a bounded domain.** Let $\Theta$ be a box in $\mathbb{R}^d$, and suppose that training is modeled with CLD in a bounded domain, i.e. that the parameters evolve according to the stochastic differential equation with reflection at the boundary (SDER)

$$d\boldsymbol{\theta}_t = -\nabla L_S\left(\boldsymbol{\theta}_t\right) dt + \sqrt{2\beta^{-1}} d\mathbf{w}_t + d\mathbf{r}_t,$$

(9)

where $L_S \geq 0$ is twice continuously differentiable, $\mathbf{w}_t$ is a standard Brownian motion, and $\mathbf{r}_t$ is a reflection process that constrains $\boldsymbol{\theta}_t$ within $\Theta$. Such weight clipping is quite common in practical scenarios such as NN training. For simplicity, we assume that $\mathbf{r}_t$ has normal reflection, meaning that the reflection is perpendicular to the boundary. An established result in the analysis of SDERs states that under these assumptions (9) has a stationary distribution $p_\infty(\boldsymbol{\theta}) \propto e^{-\beta L_S(\boldsymbol{\theta})} \mathbb{I}_\Theta \{\boldsymbol{\theta}\}$ (see Appendix H.2). Thus, when $p_0 = \mathrm{Uniform}(\Theta)$, we have $p_0 = \nu$.

**Regularized CLD in $\mathbb{R}^d$.** Suppose that the parameters evolve according to the stochastic differential equation (SDE) with weight decay (i.e. $\ell^2$ regularization)

$$d\boldsymbol{\theta}_t = -\nabla L_S(\boldsymbol{\theta}_t)\, dt - \lambda\beta^{-1}\boldsymbol{\theta}_t dt + \sqrt{2\beta^{-1}}d\mathbf{w}_t\,, \tag{10}$$

where $L_S \geq 0$ is twice continuously differentiable, $\mathbf{w}_t$ is a standard Brownian motion. Such weight decay is also quite common in practical scenarios such as NN training. Similar to the previous case, with the regularization and twice continuous differentiability of $L_S$ this process has a unique stationary distribution $p_\infty(\boldsymbol{\theta}) \propto e^{-\beta L_S(\boldsymbol{\theta})}\phi_\lambda(\boldsymbol{\theta})$, where $\phi_\lambda$ is the density of the multivariate Gaussian $\mathcal{N}(\mathbf{0}, \lambda^{-1}\mathbf{I}_d)$. Thus, when $p_0 = \mathcal{N}(\mathbf{0}, \lambda^{-1}\mathbf{I}_d)$, we also have $p_0 = \nu$.

We can now formulate a generalization bound for both cases.

**Corollary 3.1.** *Assume that the parameters evolve according to either* (9) *with* $p_0 = \mathrm{Uniform}(\Theta)$, *or* (10) *with* $p_0 = \mathcal{N}(\mathbf{0}, \lambda^{-1}\mathbf{I}_d)$. *Then for any time* $t \geq 0$, *and* $\delta \in (0,1)$,

*1. w.p.* $1 - \delta$ *over* $S \sim D^N$,

$$\mathbb{E}_{\boldsymbol{\theta}_t \sim p_t}[E_D(\boldsymbol{\theta}_t) - E_S(\boldsymbol{\theta}_t) \mid S] \leq \sqrt{\frac{\beta\mathbb{E}_{\boldsymbol{\theta} \sim p_0}[L_S(\boldsymbol{\theta}) \mid S] + \ln(N/\delta)}{2N}}\,. \tag{11}$$

*2. w.p.* $1 - \delta$ *over* $S \sim D^N$ *and* $\boldsymbol{\theta}_t \sim p_t$

$$E_D(\boldsymbol{\theta}_t) - E_S(\boldsymbol{\theta}_t) \leq \sqrt{\frac{\beta \operatorname{ess\,sup}_{p_0} L_S(\boldsymbol{\theta}) + \ln(N/\delta)}{2N}}\,. \tag{12}$$

The proof is simple — by assumption, in both cases $p_0 = \nu$ so $D_\infty(p_0 \| \nu) = 0$. The rest is a direct substitution into Theorem 2.7, and in particular, using $\beta L_S$ as potential $\Psi_S$.

## 3.1 Interpreting Corollary 3.1

Corollary 3.1 raises questions on the relevance of this setting, which we address below: (1) How large is $\mathbb{E}_{p_0} L_S(\boldsymbol{\theta})$ in practically relevant cases? (2) Can we attribute the generalization to the regularization (either with the $\ell_2$ regularization term, or the bounded domain)? (3) Can models successfully train in the presence of noise with a variance large enough to make the bounds non-vacuous?

**Magnitude of the initial loss.** Commonly, the dependence on $\mathbb{E}_{p_0} L_S(\boldsymbol{\theta})$ with realistic $p_0$ and $L_S$ is relatively mild. For example, using standard initialization schemes, Gaussian process approximations [50, 42, 35, 25] imply that the output of an infinitely wide fully connected neural network converges to a Gaussian with mean 0 and $O(1)$ variance at initialization. So in many cases $\mathbb{E}_{p_0} L_S(\boldsymbol{\theta}) = O(1)$, such as for the scalar square and logistic losses. In the multi-output case, $\mathbb{E}_{p_0} L_S(\boldsymbol{\theta})$ may also depend on the number of outputs (e.g., logarithmically so in softmax-cross-entropy). A more difficult question is concerned with the case that $\operatorname{ess\,sup}_{p_0} L_S = \infty$, which is common when $p_0$ has infinite support. This can be mitigated by clipping the loss, which is standard in practice (e.g. in reinforcement learning [48, 62]) and in the theory of optimization [37, 33]. Moreover, this clipping can be done in a differentiable way (e.g. using either softmin, tanh (e.g. $c \cdot \tanh(L/c)$), etc) and at values only slightly higher than the typical loss at the initialization (since the loss is roughly monotonically decreasing in CLD with small noise, the optimization process would typically operate below the clipping and will not be affected by it).

**Magnitude of regularization.** In the above result we must use regularization (or a bounded domain) that matches the initialization $p_0$ (this can be somewhat relaxed, see Section 3.2). The same assumption, that the regularization matches the initialization, was also made in other theoretical works on CLD [49, 38, 19]. Note that, NN models regularized this way remain highly expressive, both empirically (Appendix F) and theoretically (Appendix G), and therefore we cannot use this regularization

alone, together with classical uniform convergence approaches to show generalization. Intuitively, this is because the regularization term can be tiny, for example, in (10) the regularization term is divided by $\beta$. Therefore, when $\beta = O(N)$ (which is sufficient for a non-vacuous result), $p_0 = \nu$, and we use a standard deep nets initialization distribution $p_0$ (e.g., [21, 28], where $\lambda \propto \text{layer width}$), the regularization coefficient is $O\left(\frac{\text{layer width}}{N}\right)$ that is rather small in realistic cases. Therefore, we found (empirically) that it does not seem to have a large effect at practical timescales. In addition, one can always increase the regularization by modifying the loss $L_S \leftarrow L_S + c\|\boldsymbol{\theta}\|^2$ in (10). Under standard initializations, this changes the loss in the bound by an $O(c\tilde{d})$ factor, where $\tilde{d}$ is the depth of the neural network and so $c\tilde{d}$ is small, for common values of $c$ and $\tilde{d}$. Therefore, combining these observations, we do not see the magnitude of the regularization as a significant practical issue.

**Magnitude of noise: theoretical perspective.** In the above result we must use $\beta = O(N)$ to obtain a non-vacuous bound. This requirement is standard in many theoretical works. For example, as we will discuss below in Section 4.1, all previous generalization bounds for CLD and SGLD also required, to generalize well, $\beta = O(N)$ and potentially much worse (lower $\beta$). In addition, other theoretical works on noisy training also typically had $\beta = O(N)$ or worse. For example, when considering the ability of noisy gradient descent to escape saddle points, Jin et al. [30] uses noise sampled uniformly from a ball with a radius that depends on the dimensionality and smoothness of the problem, and thus cannot decay with $N$. Moreover, it is known that the Gibbs posterior[8] generalizes well with $\beta = O\left(\sqrt{N}\right)$ (e.g. see Theorem 2.8 in [1]), which is significantly smaller than $\beta = O(N)$. Lastly, in Appendix E we examine the impact of $\beta$ in the simple model of linear regression with i.i.d. standard Gaussian input, labels produced by a constant-magnitude teacher label noise, trained using regularized CLD as in (10), with $\lambda \propto d$ to match standard initialization. We find there that whenever $d \ll \beta \ll N$, the added noise does not significantly affect the training or population losses, and our bound is useful, i.e., it implies a vanishing generalization gap (since $\beta \ll N$ and $\mathbb{E}_{p_0} L = O(1)$). Note that $d \ll N$ is not a major constraint, since $d \ll N$ is required to obtain low population loss in this setting, even if we did not add noise to the training process (i.e. $\beta = \infty$).

**Magnitude of noise: empirical perspective.** An inverse temperature of $\beta = O(N)$ is also relevant in many practical settings. For example, in Bayesian settings, when we wish to (approximately) sample from the posterior, it is quite common to use variants of SGLD; then inverse temperatures of order $\beta = O(N)$ are commonly used to achieve good generalization [69], which matches our results. In the standard practical training settings, the inverse temperature is a hyperparameter tuned to best fit a given problem. Empirically, in Appendix F we find that $\beta = O(N)$ can be tuned to obtain non-vacuous generalization bounds for overparameterized NNs in a few small binary classification datasets (binary MNIST, Fashion MNIST, SVHN, and a parity problem), i.e. the sum of the generalization gap bound and the training error is smaller than $0.5$. Importantly, these non-vacuous bounds do not use any trajectory-dependent quantities as other non-vacuous bounds (e.g. [15, 39]), which can make them arguably more useful as they can be calculated before training. The bounds are still not very tight (at noise levels that allow for non-vacuous bounds), but we believe there is still much room for improvement in future work.

## 3.2 Extensions and Modifications

**State dependent diffusion coefficient.** Consider a state-dependent diffusion coefficient

$$d\boldsymbol{\theta}_t = -\nabla L_S\left(\boldsymbol{\theta}_t\right)dt + \sqrt{2\beta^{-1}\sigma^2\left(\boldsymbol{\theta}_t\right)}d\mathbf{w}_t + d\mathbf{r}_t\,,$$

where $\sigma^2 \in \mathcal{C}^2$. For example, in Appendix D.1 we derive the explicit form of stationary distributions when $\sigma^2\left(\boldsymbol{\theta}\right) = \left(L_S\left(\boldsymbol{\theta}\right) + \alpha\right)^k$ or $\sigma^2\left(\boldsymbol{\theta}\right) = e^{\alpha L_S\left(\boldsymbol{\theta}\right)}$, for some $k \in \mathbb{N}$ and $\alpha > 0$. In both cases, the analytic form of the stationary potential $\Psi$ can be used directly with Theorem 2.7 to derive generalization bounds.

**Restricted initialization.** In Appendix D.2 we present generalizations of Corollary 3.1 to cases where $p_0$ and $\nu$ are different. Specifically, for the bounded case we consider $p_0$ that is uniform in a subset $\Theta_0 \subset \Theta$ of the domain, and for the regularized case we consider general diagonal Gaussian initialization and regularization. In particular, this means that some of the parameters can be more

---

[8]Generalization bounds for the Gibbs posterior typically assume that it is "trained" and "tested" on the same function, while here the distribution is defined by the loss and "tested" on the error.

Table 1: **Comparison of generalization bounds for CLD.** We compare the main bounds in settings similar to the CLD setting considered here. All the bounds here consider different functions for training and evaluation, as was done in this paper with $L_S$ and $E_S, E_D$, respectively. For simplicity, we assume that $E_S, E_D$ are bounded in $[0, 1]$, and are therefore $1/2$-subGaussian via Hoeffding's inequality. We use $g_t$ to denote *trajectory-dependent* statistics of the gradients, $K$ for the Lipschitz constant, and $C$ for a bound on the loss, or the expected loss at initialization, when they are required. For compactness, low-order terms are omitted, time-dependent quantities are simplified to an approximate asymptotic value, and trajectory dependent integrals are solved by considering the statistics $g_t$ constant w.r.t. the variable of integration. Finally, all bounds assume a Gaussian initialization $\mathcal{N}\left(\mathbf{0}, \lambda^{-1}\mathbf{I}_d\right)$ and regularization term $\frac{\lambda}{2\beta} \|\boldsymbol{\theta}_t\|^2$, both with the same $\lambda$.

| Paper | Trajectory dependent | dimension dependence | Bound (big $O$) |
|---|---|---|---|
| Mou et al. [49] | ✓ | through gradients | $\sqrt{\frac{\beta}{N}} \cdot \sqrt{\frac{1}{\lambda}g_t^2}$ |
| Li et al. [38] | ✗ | through $K$ | $\frac{e^{4\beta C}\sqrt{\beta}}{N} \cdot \frac{2K}{\sqrt{\lambda}}$ |
| Futami and Fujisawa [19] | ✓ | through gradients | $\sqrt{\frac{\beta}{N}e^{8\beta C}} \cdot \sqrt{\frac{1}{\lambda}g_t^2}$ |
| Ours (11) | ✗ | ✗ | $\sqrt{\frac{\beta}{N}} \cdot \sqrt{C}$ |

loosely regularized/bounded at a cost proportional to their number. For example, in a deep NN, if only a single layer is loosely regularized/bounded, the KL-divergence cost will be proportional only to the number of parameters in that layer, not the entire $d$.

## 4 Related Work

**Information theoretic guarantees and PAC-Bayes theory.** A common type of generalization bounds consists of a measure of the dependence between the learned model and the dataset used to train it, such as the mutual information between the data and algorithm [58, 70, 61] or the KL-divergence between the predictor's distribution and any data-independent distribution [44, 9, 1]. In particular, recent works were able to estimate such dependence measures from trained models to derive non-vacuous generalization bounds, even for deep overparameterized models. For example, Dziugaite et al. [17] used held-out data to bound the KL-divergence in a PAC-Bayes bound with a data-dependent prior. Other works used some property of the trained model to estimate the information content, adding valuable insight to the mechanisms facilitating the successful generalization, such as the size of the compressed model after training, due to noise stability [3], and data structure [39].

**Generalization of the Gibbs posterior.** One classic result in the PAC-Bayesian theory of generalization is that the Gibbs posterior with properly tuned temperature minimizes the PAC-Bayes bound of McAllester [44], i.e. the KL-regularized expected loss. Raginsky et al. [59] used uniform stability [7] to derive a different generalization bound for sampling from the Gibbs distribution. Due to these known generalization capabilities, some works relied on it to derive bounds for related algorithms.

### 4.1 Explicit Comparison for CLD

Many previous works [59, 49, 38, 18, 19, 14] derived generalization bounds specifically for CLD, under different assumptions. Our bound offers some improvements over previous ones:

- It is trajectory independent, and does not require gradient statistics [49, 19].
- It does not require very large time scales to make sure we have already converged near Gibbs [59], nor does it deteriorate with time, as is common for stability-based bounds [49, 14].
- It does not depend on the dimension of the parameters, neither explicitly through constants [18], nor implicitly, e.g. through the Lipschitz constant or the norms of the gradients [49, 38, 19]. In particular, as previously discussed, using standard initialization, our in-expectation bound in (11) is dimension independent. However, our high-probability bound (12) relies on the effective supremum at $t = 0$, and may also depend on the dimension if the loss is not bounded.
- The dependence on the inverse temperature $\beta$ and loss' (or expected loss) bound $C$ is polynomial ($\sqrt{\beta C}$) instead of exponential [38, 18, 19].

- The bounded expectation assumption in (11) is weaker than a uniform bound on the loss [38, 19].
- Theorem 2.7 and Corollary 3.1 demonstrate that our results hold for general initialization-regularization pairs, beyond Gaussian initialization with matching $\ell^2$ regularization.

In Table 1 we compare in more detail Corollary 3.1 to other bounds that remain bounded as $t \to \infty$.

Finally, Dupuis et al. [14] recently derived bounds on the generalization gap for *all* intermediate times $0 \leq s \leq t$ *simultaneously*. Naturally, as avoiding parameters with large generalization gap is increasingly less likely as the process mixes, their bounds grow with time. Therefore, Dupuis et al. [14]'s bounds are qualitatively different, and higher than most other bounds, including ours.

### 4.2 Technical Novelty

As a representative example, we first focus on Raginsky et al. [59], which provided a bound for CLD (as an intermediate step for deriving a generalization bound for SGLD, a discretized version of CLD). Using spectral methods [e.g. 5], they bound the distance between the process' distribution to the Gibbs posterior, which, when combined with the generalization bound for the Gibbs distribution, results in generalization bounds for intermediate times. Our Corollary 2.5 and the preceding arguments are similar to the proof of Lemma 3.4 of Raginsky et al. [59] that bounds the divergence between the initialization and the Gibbs distribution, where their dissipativity coefficient $m$ corresponds to our explicit $\ell^2$ regularization coefficient $\lambda$. We use some significant observations that make the bound simpler, and time/dimension/Lipschitz/smoothness independent.

- Instead of a bound on the convergence of intermediate time distributions to Gibbs, which restricts the result to very large times and introduces exponential dependence on dimensionality through the spectral gap, we only require the monotonic convergence to it. As a result, we do not use a spectral gap, but a complexity term for the initial distribution. This also enables us to generalize the result to *any* Markov process, relying on $\mathbb{E}_{p_0} \Psi$ as a complexity term for the Gibbs posterior, which is also included in Lemma 3.4 of Raginsky et al. [59] along other quantities.
- By using a symmetric version of the divergence (e.g. by summing $\mathrm{KL}\left(p \, \| \, q\right)$ and $\mathrm{KL}\left(q \, \| \, p\right)$) we were able to *completely remove the partition function* from the analysis, avoiding the complications arising from it.
- By separating the regularization from the loss we were able to disentangle their effects.

This approach also sidesteps the main difficulties encountered by other works, e.g., using stability-based bounds [49, 38, 19] which either diverge with training time or have dimension dependence.

### 4.3 Generalization Guarantees Applicable for Neural Networks

Many additional lines of work established generalization guarantees applicable for NNs, but are less directly related to our work. These results have some limitations that do not exist in ours. For example, NTK analysis [29] can imply generalization guarantees in certain settings, but they do not allow for feature learning; Mean-field results [45] require non-standard initialization and specific architectures; Algorithmic stability analysis Bousquet and Elisseeff [7], Hardt et al. [26], Richards and Rabbat [60], Lei et al. [36], Wang et al. [67] only apply when the number of iterations is sufficiently small; Norm-based generalization bounds [6, 22] ignore optimization aspects and depend exponentially on the network's depth; And bounds for random interpolators [8] involve impractical training procedures.

A closely related setting to the one studied here is SGLD, i.e. a discretized version of CLD. There is an extensive line of work bounding the generalization gap of such models (see [59, 49, 55, 51, 18, 19, 13] for a partial list). These results typically have a significant dependence on hyperparameter stemming from the discretization such as the learning rate and batch size, or suffer from constraints similar to the ones discussed in Section 4.1, such as dependence on trajectory or dimensionality (e.g. via smoothness, parameter norms, log-Sobolev or spectral gap constants).

## 5 Discussion, Limitations, and Future Work

**Summary.** We derived a simple generalization bound for general parametric models trained using a Markov-process-based algorithm, where the dynamics have a stationary distribution with bounded

potential or expected potential. For CLD with regularization/boundedness constraint matching the initial distribution, we proved that the model generalizes well when the inverse temperature is of order $\beta = O(N)$. There are several interesting directions to extend this result.

**Non-isotropic noise.** We can consider a more general model for training, such as

$$\mathrm{d}\boldsymbol{\theta}_t = -\nabla L_S(\boldsymbol{\theta}_t)\,\mathrm{d}t + \boldsymbol{\Sigma}(\boldsymbol{\theta}_t)\,\mathrm{d}\mathbf{w}_t + \mathrm{d}\mathbf{r}_t\,,$$

where $\boldsymbol{\Sigma}$ is a matrix-valued dispersion coefficient, and $\mathbf{r_t}$ is some regularization process, such as $\ell^2$ regularization or a reflection process in a bounded domain. In contrast, in this paper, to derive concrete generalization bounds, we focused on CLD with isotropic noise, i.e. such that $\boldsymbol{\Sigma}$ is a scalar multiple of the identity matrix. The reason for this was that our bound (Corollary 3.1) relies on explicit analytical expressions or bounds on stationary distributions, which are difficult to find in the general case. In addition, in typical overparameterized settings, the noise induced by the randomness of SGD may not only be non-isotropic, but also singular with low-rank. The analysis of such processes poses various challenges beyond the ability to derive an analytic form for their stationary distribution. For example, they may concentrate on low-dimensional manifolds, possibly making the KL-divergence term infinite, or making some of the assumptions unrealistic (e.g. the choice of initial distribution).

**No regularization.** In this work, we only considered processes that have stationary *probability* measures. For this reason, in the examples in Section 3 we used either a bounded domain or regularization. This seems essential for generalization at $t \to \infty$, unless there are other architectural constraints. For example, consider training a model for the classification of randomly labeled data. Without regularization, a sufficiently expressive model is likely to arrive (at some point) at high training accuracy, yet it cannot generalize in this setting. Nonetheless, it might be possible to ensure generalization as a function of time, but here we focus on time-independent bounds.

**Discrete time steps.** The behavior of SGD with a large step size may be qualitatively different than that of the continuous process considered here. Specifically, Azizian et al. [4] showed that while the asymptotic distribution of SGD resembles the Gibbs posterior, it is influenced by the step size and geometry of the loss surface. While an extension of our analysis to this setting is straightforward *given* a stationary distribution, such stationary distributions are typically hard to find explicitly (except in simple cases, such as quadratic potentials), and the error terms coming from their approximations are typically detrimental to finding non-vacuous generalization bounds, as they may depend on the dimension of the parameters through the model's Lipschitz or smoothness coefficients, etc. (49, 38, 19, 14). Hence, a direct application of our approach to such algorithms requires additional considerations. An alternative approach is to incorporate a Metropolis-Hastings type rejection [47, 27], ensuring that the stationary distribution is indeed the Gibbs posterior.

**Can noise be useful for generalization?** There is a long line of work in the literature (e.g. see [20] and references therein), debating the effect of noise on generalization. Our work does not imply that higher noise improves the test error, only that it decreases the gap between training and testing. Since higher noise *could hurt* the training error, the overall effect depends on the specific situation. Even if introducing noise does not improve test performance, there could still be an advantage to introducing noise, based on our results, in that it reduces the *gap* and thus could *increase* the training error to match the test error in cases we cannot hope to learn (i.e. to get a small test error). This is a good thing since it prevents being mislead by overfitting, hopefully without hurting the test error when we can generalize well (i.e. in learnable regimes, both training and test errors are low, perhaps also without noise, but in non-learnable regimes, where the test error is necessarily high, noise forces the training error to be high as well, so that the gap is small). Indeed, in our small-scale experiments in Appendix F, we noticed that a small amount of noise can decrease the generalization gap, without significantly harming the test error (e.g. see the bottom half of Tables 2 to 4). Further analysis is necessary in order to establish general conditions under which test performance is not significantly hurt by noise, while ensuring a small gap. This, in particular, requires studying the effect of noise on the training loss, and what noise level still ensures obtaining a small training loss in learnable regimes.

## Acknowledgments and Disclosure of Funding

The research of DS was Funded by the European Union (ERC, A-B-C-Deep, 101039436). Views and opinions expressed are however those of the author only and do not necessarily reflect those of the European Union or the European Research Council Executive Agency (ERCEA). Neither the

European Union nor the granting authority can be held responsible for them. DS also acknowledges the support of the Schmidt Career Advancement Chair in AI. GV is supported by the Israel Science Foundation (grant No. 2574/25), by a research grant from Mortimer Zuckerman (the Zuckerman STEM Leadership Program), and by research grants from the Center for New Scientists at the Weizmann Institute of Science, and the Shimon and Golde Picker – Weizmann Annual Grant. Part of this work was done as part of the NSF-Simons funded Collaboration on the Mathematics of Deep Learning. NS was partially supported by the NSF TRIPOD Institute on Data Economics Algorithms and Learning (IDEAL) and an NSF-IIS award.

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

**Appendix structure:**

- In Appendix A we recap and establish notation and conventions, and present some well-known lemmas.

- In Appendix B we prove Theorem 2.7 and its related claims in Section 2.

- In Appendix C we discuss the tightness and necessity of the divergence conditions found in Appendix B.

- In Appendix D we prove a generalized version of Corollary 3.1.

- The bounds found in this paper only bound the generalization gap, and not the absolute error of a model. In Appendix E and Appendix F we study the applicability of our bound in realistic settings. Specifically, whether the regime in which the bound on the generalization gap is non-vacuous allows for meaningful learning, i.e. coincides with a regime in which the absolute training error is also small. In Appendix E we study linear regression trained with CLD, for which we can analytically characterize the training loss, and in Appendix F we experiment with NNs trained with SGLD (discretized version of CLD) on standard training sets.

- As Section 3 deals only with models trained with some form of regularization, it is natural to ask whether the regularization alone is sufficient for the use of uniform convergence to arrive the desired generalization bounds. In Appendix G we show that the regularization used in Section 3 is not sufficient for such bounds, and that the models can remain highly expressive.

- Finally, for completeness, in Appendix H we recall some definitions and properties related to SDEs used throughout the paper.

## A  Preliminary and Auxiliary Results

### A.1  Preliminaries

We start by restating and introducing notation.

**Notation.**  We use bold lowercase letters (e.g. $\mathbf{x} \in \mathbb{R}^d$) to denote vectors, bold capital letters to denote matrices (e.g. $\mathbf{A} \in \mathbb{R}^{d \times d}$), and regular capital letters to denote random elements (e.g. $S, X, Y$). We may deviate from these conventions when it does not create confusion. Unless stated otherwise, all vectors are assumed to be column vectors. Specifically, we use $\mathbf{e}_i \in \mathbb{R}^d$, $i = 1, \ldots, d$, to denote the standard basis vector with 1 in the $i^{th}$ entry, and 0 elsewhere. For a subset $\Omega \subseteq \mathbb{R}^d$, we denote by $\overline{\Omega}$, $\partial \Omega$, and $\Omega^\circ$, the closure, boundary, and interior of $\Omega$, respectively. In addition, we denote the volume of $B \subset \Omega$, when it is defined, by $|B|$. With some abuse of notation, when $B$ is finite we denote its cardinality by $|B|$. We use $\|\cdot\|$ for the standard Euclidean norm on $\mathbb{R}^d$. Then, the open Euclidean ball centered at $\mathbf{x} \in \mathbb{R}^d$ with radius $r > 0$ is $B_r(\mathbf{x}) = \left\{ \mathbf{y} \in \mathbb{R}^d \mid \|\mathbf{y} - \mathbf{x}\| < r \right\}$. In addition, we use $\mathbb{I}\{\cdot\}$ for the indicator function, and specifically for $A \subset \mathbb{R}^d$ and $\mathbf{x} \in \mathbb{R}^d$, $\mathbb{I}_A\{x\} = \mathbb{I}\{\mathbf{x} \in A\}$. We denote the set of all probability measures over $\Omega$ by $\Delta(\Omega)$. For some $\mu \in \Delta(\Omega)$ with density $p$, with some abuse of notation we denote $p \in \Delta(\Omega)$, and $p(B) = \mu(B)$ for measurable $B \subseteq \Omega$. In addition, we use $\mathbb{E}_{X \sim p}$ or $\mathbb{E}_p$ to denote the expectation w.r.t $p$, and omit the subscript when it can be inferred. For two distributions $\mu, \nu$ with densities $p, q$ we denote by $\mathrm{KL}(\mu \| \nu) = \mathrm{KL}(p \| q)$ their KL-divergence (relative entropy). Furthermore, we use $H(\delta) = -\delta \ln(\delta) - (1 - \delta) \ln(1 - \delta)$, $\delta \in [0, 1]$, for the binary entropy function (in nats). We denote the divergence of a vector field by $\nabla \cdot$, and the gradient and Laplacian of a scalar function by $\nabla$ and $\Delta = \nabla \cdot \nabla$, respectively. Given a domain $E \subset \mathbb{R}^k$ and $k \in \mathbb{Z}_+ \cup \{\infty\}$, we denote by $\mathcal{C}^k(E)$ the set of real valued functions that are continuous over $\overline{E}$, and $k$-times continuously differentiable with continuous partial derivatives in $E$. In particular, $\mathcal{C} = \mathcal{C}^0$ is the set of continuous functions.

**Conventions.**  Unless stated otherwise, we use $\Omega \subset \mathbb{R}^d$ to denote a non-empty, connected, and open domain. In addition, we follow the following naming conventions for probability distributions.

- For a discrete/continuous-time Markov process, we use $p_n$ or $p_t$ for its marginal distribution at time $n \in \mathbb{N}$ or $t \in \mathbb{R}_+$.

- We denote stationary distributions of Markov processes by $p_\infty$.
- In the context of PAC-Bayesian theory, we denote prior distributions by $\rho$, and data dependent posteriors by $\hat{\rho} = \hat{\rho}_S$.
- In case some stationary distribution is also data-dependent, we use $p_\infty$.
- We also use $p, q$ for generic distributions, or modify the pervious notation.

## A.2 General Lemmas: Data processing inequality and generalized second laws of thermodynamics

For completeness, we start by proving some well known results in probability and the theory of Markov processes.

**Lemma A.1** (Data processing inequality). *Let $p(x, y)$ and $q(x, y)$ be the densities of two joint distributions over a product measure space $\mathcal{X} \times \mathcal{Y}$. Denote by $p_X(x), q_X(x)$ the marginal densities, e.g.*

$$p_X(x) = \int_{\mathcal{Y}} p(x, y) \, dy,$$

*and by $p(y \mid x), q(y \mid x)$ the conditional densities, so $p(x, y) = p(y \mid x) p_X(x)$, and similarly for $q$. Then*

$$\mathrm{KL}(p_X \,\|\, q_X) \le \mathrm{KL}(p \,\|\, q).$$

*Proof.* By definition of the KL divergence

$$\mathrm{KL}(p \,\|\, q) = \int_{\mathcal{X} \times \mathcal{Y}} p(x, y) \ln\left(\frac{p(x, y)}{q(x, y)}\right) dx dy$$

$$= \int_{\mathcal{X} \times \mathcal{Y}} p(x, y) \ln\left(\frac{p(y \mid x) p_X(x)}{q(y \mid x) q_X(x)}\right) dx dy$$

$$= \int_{\mathcal{X} \times \mathcal{Y}} p(x, y) \ln\left(\frac{p_X(x)}{q_X(x)}\right) dx dy + \int_{\mathcal{X} \times \mathcal{Y}} p(x, y) \ln\left(\frac{p(y \mid x)}{q(y \mid x)}\right) dx dy$$

$$= \int_{\mathcal{X} \times \mathcal{Y}} p(y \mid x) p_X(x) \ln\left(\frac{p_X(x)}{q_X(x)}\right) dx dy + \int_{\mathcal{X} \times \mathcal{Y}} p_X(x) p(y \mid x) \ln\left(\frac{p(y \mid x)}{q(y \mid x)}\right) dx dy$$

$$[\text{Fubini}] = \int_{\mathcal{X}} p_X(x) \ln\left(\frac{p_X(x)}{q_X(x)}\right) dx + \mathbb{E}_{X \sim p_X} \int_{\mathcal{Y}} p(y \mid X) \ln\left(\frac{p(y \mid X)}{q(y \mid X)}\right) dy$$

$$= \mathrm{KL}(p_X \,\|\, q_X) + \mathbb{E}_{X \sim p_X} \mathrm{KL}(p(\cdot \mid X) \,\|\, q(\cdot \mid X)).$$

The KL divergence is non-negative and therefore the expectation in the last line is non-negative as well, and we conclude that

$$\mathrm{KL}(p \,\|\, q) \ge \mathrm{KL}(p_X \,\|\, q_X).$$

$\square$

Let $X_n = \{X_n\}_{n=0}^{\infty}$ be a discrete-time Markov chain on $\Omega \subset \mathbb{R}^d$, with transition kernel $P(y \mid x)$ such that for all $n \in \mathbb{N}_0$,

$$p_{n+1}(y) = \int_{\Omega} P(y \mid x) p_n(x) \, dx.$$

In addition, assume that the there exists an invariant distribution $p_\infty$ such that

$$p_\infty(y) = \int_{\Omega} P(y \mid x) p_\infty(x) \, dx.$$

We proceed to present a generalized form of the second law of thermodynamics, regarding the monotonicity of the (relative) entropy of Markov processes with possibly non-uniform stationary distributions [11, 12].

**Lemma A.2** (Generalized second law of thermodynamics). *For all $n \geq 0$,*

$$\mathrm{KL}\left(p_{n+1} \,\|\, p_\infty\right) \leq \mathrm{KL}\left(p_n \,\|\, p_\infty\right) .$$

*Proof.* First, note that we can assume that $\mathrm{KL}\left(p_n \,\|\, p_\infty\right) < \infty$, since otherwise the claim holds trivially. Let $q\left(x, y\right) = p_n\left(x\right) P\left(y \mid x\right)$ be the joint densities of $\left(X_n, X_{n+1}\right)$ where $X_n \sim p_n$, and let $r\left(x, y\right) = p_\infty\left(x\right) P\left(y \mid x\right)$ be the joint distribution under $X_n \sim p_\infty$. By definition of $p_{n+1}$,

$$q_Y\left(y\right) = p_{n+1}\left(y\right) ,$$

and by definition of the stationary distribution,

$$r^Y\left(y\right) = p_\infty\left(y\right) .$$

Therefore according to Lemma A.1,

$$\mathrm{KL}\left(p_{n+1} \,\|\, p_\infty\right) \leq \mathrm{KL}\left(q \,\|\, r\right) .$$

In addition,

$$
\begin{aligned}
\mathrm{KL}\left(q \,\|\, r\right) &= \int_{\Omega \times \Omega} q\left(x, y\right) \ln\left(\frac{q\left(x, y\right)}{r\left(x, y\right)}\right) dx dy \\
&= \int_{\Omega \times \Omega} q\left(x, y\right) \ln\left(\frac{p_n\left(x\right) P\left(y \mid x\right)}{p_\infty\left(x\right) P\left(y \mid x\right)}\right) dx dy \\
&= \int_{\Omega \times \Omega} q\left(x, y\right) \ln\left(\frac{p_n\left(x\right)}{p_\infty\left(x\right)}\right) dx dy \\
&= \int_{\Omega \times \Omega} p_n\left(x\right) P\left(y \mid x\right) \ln\left(\frac{p_n\left(x\right)}{p_\infty\left(x\right)}\right) dx dy \\
[\text{Fubini}] &= \int_{\Omega} p_n\left(x\right) \ln\left(\frac{p_n\left(x\right)}{p_\infty\left(x\right)}\right) dx \\
&= \mathrm{KL}\left(p_n \,\|\, p_\infty\right) ,
\end{aligned}
$$

and overall

$$\mathrm{KL}\left(p_{n+1} \,\|\, p_\infty\right) \leq \mathrm{KL}\left(p_n \,\|\, p_\infty\right) .$$

$\square$

A similar result can be obtained form $D_\infty\left(\cdot \,\|\, \cdot\right)$.

**Lemma A.3** (The Pointwise Second Law). *For all $n > 0$ :*

$$D_\infty\left(p_{n+1} \,\|\, p_\infty\right) \ \leq \ D_\infty\left(p_n \,\|\, p_\infty\right).$$

*Proof.* Let $p, q$ be some probability measures such that $\frac{dp}{dq}$ exists. By definition,

$$D_\infty\left(p \,\|\, q\right) = \operatorname{ess\,sup}_q \ln \frac{dp}{dq} = \inf\left\{c \in \mathbb{R} \ \middle| \ q\left(\left\{x \ \middle| \ \ln \frac{dp}{dq} > c\right\}\right) = 0\right\} .$$

Let $C \in \mathbb{R}$ and suppose that for all measurable $A \subset \mathcal{X}$, $p\left(A\right) \leq e^C q\left(A\right)$. Assume by way of contradiction that $D_\infty\left(p \,\|\, q\right) > C$, that is, that there exists $c > C$ such that

$$q\left(\left\{x \ \middle| \ \ln \frac{dp}{dq} > c\right\}\right) > 0 .$$

Denote

$$A = \left\{x \ \middle| \ \ln \frac{dp}{dq} > c\right\} ,$$

then

$$p\left(A\right) = \int_A \frac{\mathrm{d}p}{\mathrm{d}q}\,\mathrm{d}q > e^c q\left(A\right) > e^C q\left(A\right)\ ,$$

in contradiction to the assumption. Therefore, for all $C$ such that $p\left(A\right) \le e^C q\left(A\right)$ for all measurable $A$, $C \ge D_\infty\left(p \,\|\, q\right)$. We can now show the claim.

Let $P(\mathrm{d}y \mid x)$ be the processes' transition kernel (in measure form). We can assume $D_\infty\left(p_n \,\|\, p_\infty\right) < \infty$, since otherwise the claim holds trivially. Let $A$ be measurable, then by definition,

$$p_{n+1}\left(A\right) = \int P\left(A \mid x\right) dp_n\left(x\right) = \int P\left(A \mid x\right) \frac{\mathrm{d}p_n}{\mathrm{d}p_\infty}\left(x\right)\mathrm{d}p_\infty\left(x\right)$$

$$\le e^{D_\infty\left(p_n \,\|\, p_\infty\right)} \int P\left(A \mid x\right)\mathrm{d}p_\infty\left(x\right) = e^{D_\infty\left(p_n \,\|\, p_\infty\right)}p_\infty\left(A\right)\ ,$$

so $D_\infty\left(p_{n+1} \,\|\, p_\infty\right) \le D_\infty\left(p_n \,\|\, p_\infty\right)$. □

We can now state the relevant results for continuous-time processes.

**Corollary A.4.** *Let $X_t$ be a Markov process with marginals $p_t$ and stationary distribution $p_\infty$. Then, for all $t > 0$ :*

$$\mathrm{KL}\left(p_t \,\|\, p_\infty\right) \le \mathrm{KL}\left(p_0 \,\|\, p_\infty\right) \ \text{ or } \ D_\infty\left(p_t \,\|\, p_\infty\right) \le D_\infty\left(p_0 \,\|\, p_\infty\right)$$

*Proof.* Let $0 < t$ and let $\Delta t > 0$ such that $t \in \Delta t \cdot \mathbb{N}$. Define $Y_n = X_{n\Delta t}$, then $Y_n$ is a discrete time Markov chain with marginals $p_{n \cdot \Delta t}$ and stationary distribution $p_\infty$, so Lemma A.2 and Lemma A.3 imply the results. □

# B  Proof of Theorem 2.7 and its Related Claims in Section 2

In this section, we present the proof of Theorem 2.7, the claims leading to it, and some of its generalizations.

## B.1  Derivation of Corollary 2.5

**Recall Claim 2.3.** If $p, q, \mu, \nu$ are probability measures, and $p$ is Gibbs w.r.t $q$ with potential $\Psi < \infty$, then

1. $\mathrm{KL}_\mu \left( p \,\|\, q \right) + \mathrm{KL}_\nu \left( q \,\|\, p \right) = \mathbb{E}_\nu \Psi - \mathbb{E}_\mu \Psi$,

2. $D_\infty^\mu \left( p \,\|\, q \right) + D_\infty^\nu \left( q \,\|\, p \right) = \operatorname{ess\,sup}_\nu \Psi - \operatorname{ess\,inf}_\mu \Psi$.

In particular, $\mathrm{KL} \left( p \,\|\, q \right) + \mathrm{KL} \left( q \,\|\, p \right) = \mathbb{E}_q \Psi - \mathbb{E}_p \Psi$, and $D_\infty \left( p \,\|\, q \right) + D_\infty \left( q \,\|\, p \right) = \operatorname{ess\,sup}_q \Psi - \operatorname{ess\,inf}_p \Psi$.

*Proof.* By definition $\frac{\mathrm{d}p}{\mathrm{d}q} = Z^{-1} e^{-\Psi}$ where $Z < \infty$ is the appropriate partition function. Then we have

$$\mathrm{KL}_\mu \left( p \,\|\, q \right) + \mathrm{KL}_\nu \left( q \,\|\, p \right) = \int \mathrm{d}\mu \ln \frac{\mathrm{d}p}{\mathrm{d}q} + \int \mathrm{d}\nu \ln \frac{\mathrm{d}q}{\mathrm{d}p}$$
$$= \int \left( -\Psi - \ln Z \right) \mathrm{d}\mu + \int \left( \Psi + \ln Z \right) \mathrm{d}\nu = \mathbb{E}_\nu \Psi - \mathbb{E}_\mu \Psi \,.$$

Also,

$$D_\infty^\mu \left( p \,\|\, q \right) + D_\infty^\nu \left( q \,\|\, p \right) = \ln \left( \operatorname{ess\,sup}_\mu \frac{\mathrm{d}p}{\mathrm{d}q} \right) + \ln \left( \operatorname{ess\,sup}_\nu \frac{\mathrm{d}q}{\mathrm{d}p} \right)$$
$$= \operatorname{ess\,sup}_\mu \left( -\Psi - \ln Z \right) + \operatorname{ess\,sup}_\nu \left( \Psi + \ln Z \right) = \operatorname{ess\,sup}_\nu \Psi - \operatorname{ess\,inf}_\mu \Psi \,,$$

where in the last equality we used the fact that $\operatorname{ess\,sup} \left( -\Psi \right) = -\operatorname{ess\,inf} \Psi$, and that $Z$ is a constant. $\qquad\square$

Using the Chain Rule and Claim 2.4, we derive the bounds of (1) and (2), as re-stated and established in the following lemma.

**Lemma B.1.** *If $p_t$ is the marginal distribution of a Markov process with initial distribution $p_0$ at time $t$, $p_\infty$ is a stationary distribution, and $\nu$ is a probability measure, then*

$$\mathrm{KL} \left( p_t \,\|\, \nu \right) \leq \mathrm{KL} \left( p_0 \,\|\, \nu \right) + \mathrm{KL}_{p_0} \left( \nu \,\|\, p_\infty \right) + \mathrm{KL}_{p_t} \left( p_\infty \,\|\, \nu \right) \,,$$

*and similarly,*

$$D_\infty \left( p_t \,\|\, \nu \right) \leq D_\infty \left( p_0 \,\|\, \nu \right) + D_\infty^{p_0} \left( \nu \,\|\, p_\infty \right) + D_\infty^{p_t} \left( p_\infty \,\|\, \nu \right) \,.$$

*Proof.* This is a simple application of the chain rule,

$$\mathrm{KL} \left( p_t \,\|\, \nu \right) = \int \mathrm{d}p_t \ln \frac{\mathrm{d}p_t}{\mathrm{d}\nu} = \int \mathrm{d}p_t \ln \frac{\mathrm{d}p_t}{\mathrm{d}p_\infty} \frac{\mathrm{d}p_\infty}{\mathrm{d}\nu} = \mathrm{KL} \left( p_t \,\|\, p_\infty \right) + \mathrm{KL}_{p_t} \left( p_\infty \,\|\, \nu \right)$$
$$\leq \mathrm{KL} \left( p_0 \,\|\, p_\infty \right) + \mathrm{KL}_{p_t} \left( p_\infty \,\|\, \nu \right) = \mathrm{KL} \left( p_0 \,\|\, \nu \right) + \mathrm{KL}_{p_0} \left( \nu \,\|\, p_\infty \right) + \mathrm{KL}_{p_t} \left( p_\infty \,\|\, \nu \right) \,,$$

where in the first inequality we used Claim 2.4. Similarly,

$$D_\infty \left( p_t \,\|\, \nu \right) = \operatorname{ess\,sup}_{p_t} \ln \frac{\mathrm{d}p_t}{\mathrm{d}\nu} = \operatorname{ess\,sup}_{p_t} \ln \frac{\mathrm{d}p_t}{\mathrm{d}p_\infty} \frac{\mathrm{d}p_\infty}{\mathrm{d}\nu} \leq D_\infty \left( p_t \,\|\, p_\infty \right) + D_\infty^{p_t} \left( p_\infty \,\|\, \nu \right)$$
$$\leq D_\infty \left( p_0 \,\|\, p_\infty \right) + D_\infty^{p_t} \left( p_\infty \,\|\, \nu \right) = D_\infty \left( p_0 \,\|\, \nu \right) + D_\infty^{p_0} \left( \nu \,\|\, p_\infty \right) + D_\infty^{p_t} \left( p_\infty \,\|\, \nu \right) \,.$$
$\qquad\square$

Corollary 2.5 now follows from plugging in Claim 2.3 into Lemma B.1.

Given these bounds on the divergences, All that remains in order to prove Theorem 2.7 is plugging Corollary 2.5 into a PAC-Bayes bound.

## B.2 In-Expectation PAC-Bayes Bounds

**Theorem B.2** (Theorem 5 from Maurer [43]). *For any $\delta \in (0,1)$ and any $N \geq 8$, for any data-independent prior distribution $\rho$:*

$$\mathbb{P}_{S \sim D^N} \left( \forall_{\hat{\rho}} \, \mathrm{kl} \left( \mathbb{E}_{h \sim \hat{\rho}} E_S(h) \, \| \, \mathbb{E}_{h \sim \hat{\rho}} E_D(h) \right) \leq \frac{\mathrm{KL}(\hat{\rho} \, \| \, \rho) + \ln \frac{2\sqrt{N}}{\delta}}{N} \right) \geq 1 - \delta \,,$$

*where $\mathrm{kl}(a \, \| \, b) = a \ln \frac{a}{b} + (1-a) \ln \frac{1-a}{1-b}$ for $0 \leq a, b \leq 1$ is the KL divergence for a Bernoulli random variable, and $\hat{\rho}$ denotes a posterior distribution.*

## B.3 Single-Sample PAC-Bayes Bounds

Theorem B.2 can be viewed as a bound in expectation over the draw from the posterior, which corresponds to the traditional PAC-Bayes view of considering the expected error of a randomized predictor. But it is actually possible to get guarantees for a single draw from this predictor, which is more appropriate when we view the randomness as part of the training algorithm, that then outputs a single deterministic predictor (chosen at random). High probability guarantees for a single draw from the posterior were shown by Alquier et al. [1] based on Catoni [9] and also discussed by Dziugaite and Roy [16]. Here we present a tight version based on a simple modification to Maurer's proof [43].

**Theorem B.3.** *For any $\delta \in (0,1)$ and $N \geq 8$, for any data independent prior $\rho$, and any learning rule specified by a conditional probability $h|S \sim \hat{\rho}_S$ such that $\rho \ll \hat{\rho}_S$ $S$-a.s.,*

$$\mathbb{P}_{S \sim D^N, h \sim \hat{\rho}_S} \left( \mathrm{kl}\left( E_S(h) \, \| \, E_D(h) \right) \leq \frac{\ln \frac{\mathrm{d}\hat{\rho}_S}{\mathrm{d}\rho}(h) + \ln \frac{2\sqrt{N}}{\delta}}{N} \right) \geq 1 - \delta \,,$$

*and so, by the definition of $D_\infty(\hat{\rho}_S \, \| \, \rho)$,*

$$\mathbb{P}_{S \sim D^N, h \sim \hat{\rho}_S} \left( \mathrm{kl}\left( E_S(h) \, \| \, E_D(h) \right) \leq \frac{D_\infty(\hat{\rho}_S \, \| \, \rho) + \ln \frac{2\sqrt{N}}{\delta}}{N} \right) \geq 1 - \delta \,.$$

*Proof.* Following and modifying the proof of Theorem 5 of Maurer [43], we start with the inequality $\mathbb{E}_S \left[ e^{N \mathrm{kl}(E_S(h) \| E_D(h))} \right] \leq 2\sqrt{N}$ [43, Theorem 1], which holds for any $h$, and so also in expectation over $h$ w.r.t. $\rho$:

$$2\sqrt{N} \geq \mathbb{E}_{h \sim \rho} \mathbb{E}_S \left[ \exp\left( N \, \mathrm{kl}\left( E_S(h) \, \| \, E_D(h) \right) \right) \right] = \mathbb{E}_S \mathbb{E}_{h \sim \rho} \left[ \exp\left( N \, \mathrm{kl}\left( E_S(h) \, \| \, E_D(h) \right) \right) \right]$$

with a change of measure from $\rho$ to $\hat{\rho}_S$,

$$= \mathbb{E}_S \mathbb{E}_{h \sim \hat{\rho}_S} \left[ \exp\left( N \, \mathrm{kl}\left( E_S(h) \, \| \, E_D(h) \right) \right) \frac{\mathrm{d}\rho}{\mathrm{d}\hat{\rho}_S}(h) \right] \tag{13}$$

$$= \mathbb{E}_{S, h \sim \hat{\rho}_S} \left[ \exp\left( N \, \mathrm{kl}\left( E_S(h) \, \| \, E_D(h) \right) - \ln \frac{\mathrm{d}\hat{\rho}_S}{\mathrm{d}\rho}(h) \right) \right] \tag{14}$$

Now applying Markov's inequality, we get:

$$\mathbb{P}_{S, h \sim \hat{\rho}_S} \left( \exp\left( N \, \mathrm{kl}\left( E_S(h) \, \| \, E_D(h) \right) - \ln \frac{\mathrm{d}\hat{\rho}_S}{\mathrm{d}\rho}(h) \right) \leq \frac{2\sqrt{N}}{\delta} \right) \geq 1 - \delta \,. \tag{15}$$

Rearranging terms, we get the desired bound. $\qquad \square$

## B.4 Arriving at Theorem 2.7

**Theorem B.4.** *Consider any distribution $D$ over $\mathcal{Z}$, function $f : \mathcal{H} \times \mathcal{Z} \to [0,1]$, and sample size $N \geq 8$, any distribution $\nu$ over $\mathcal{H}$, and any discrete or continuous time process $\{h_t \in \mathcal{H}\}_{t \geq 0}$ (i.e. $t \in \mathbb{Z}_+$ or $t \in \mathbb{R}_+$) that is time-invariant Markov conditioned on $S$. Denote $p_0(\cdot; S)$ the initial distribution of the Markov process (that may depend on $S$). Let $p_\infty(\cdot; S)$ be any stationary distribution of the process conditioned on $S$, and $\Psi_S(h) \geq 0$ a non-negative potential function that can depend arbitrarily on $S$, such that $p_\infty(\cdot; S)$ is Gibbs w.r.t. $\nu$ with potential $\Psi_S$. Then:*

1. With probability $1 - \delta$ over $S \sim D^N$,

$$\mathrm{kl}\left(\mathbb{E}\left[E_S(h_t)|S\right] \| \mathbb{E}\left[E_D(h_t)|S\right]\right) \leq \frac{\mathrm{KL}\left(p_0(\cdot; S) \| \nu\right) + \mathbb{E}\left[\Psi_S(h_0)|S\right] + \ln 2\sqrt{N}/\delta}{N} \quad (16)$$

and so

$$\mathbb{E}\left[E_D(h_t) - E_S(h_t)|S\right] \leq \sqrt{2\mathbb{E}\left[E_S(h_t) \mid S\right] \frac{\mathrm{KL}\left(p_0(\cdot; S) \| \nu\right) + \mathbb{E}\left[\Psi_S(h_0)|S\right] + \ln 2\sqrt{N}/\delta}{N}}$$
$$+ 2\frac{\mathrm{KL}\left(p_0(\cdot; S) \| \nu\right) + \mathbb{E}\left[\Psi_S(h_0)|S\right] + \ln 2\sqrt{N}/\delta}{N} \quad (17)$$

2. With probability $1 - \delta$ over $S \sim D^N$ and over $h_t$:

$$\mathrm{kl}\left(E_S(h_t) \| E_D(h_t)\right) \leq \frac{D_\infty\left(p_0(\cdot; S) \| \nu\right) + \mathrm{ess\,sup}_{p_0} \Psi_S(h_0) + \ln 2\sqrt{N}/\delta}{N} \quad (18)$$

and so, when $E_S(h_t) < E_D(h_t)$

$$E_D(h_t) - E_S(h_t) \leq \sqrt{2E_S(h_t) \frac{D_\infty\left(p_0(\cdot; S) \| \nu\right) + \mathrm{ess\,sup}_{p_0} \Psi_S(h_0) + \ln 2\sqrt{N}/\delta}{N}}$$
$$+ 2\frac{D_\infty\left(p_0(\cdot; S) \| \nu\right) + \mathrm{ess\,sup}_{p_0} \Psi_S(h_0) + \ln 2\sqrt{N}/\delta}{N} \quad (19)$$

**Lemma B.5.** *Let $a, b \in [0, 1]$. Then*

$$b \leq a + \sqrt{2a\mathrm{kl}\left(a \| b\right)} + 2\mathrm{kl}\left(a \| b\right). \quad (20)$$

*Proof.* The KL divergence is non-negative, so it suffices to consider the case that $b \geq a$. Defining $\varphi : [0, 1 - a] \to \mathbb{R}$ as

$$\varphi\left(u\right) = \frac{u^2}{2\left(a + u\right)},$$

it can be readily checked by differentiation that for all $u \in [0, 1 - a]$,

$$\mathrm{kl}\left(a \| a + u\right) \geq \varphi\left(u\right).$$

In particular, for $u = b - a \in [0, 1 - a]$,

$$\mathrm{kl}\left(a \| b\right) \geq \frac{\left(b - a\right)^2}{2b}. \quad (21)$$

Next, we consider the following inequality

$$2u^2 + \sqrt{2a}u + a - b \geq 0, \ u \geq 0. \quad (22)$$

Solving for $u$, it turns out that the inequality holds when

$$u \geq \frac{\sqrt{8b - 6a} - \sqrt{2a}}{4}. \quad (23)$$

In addition, under the assumption that $b \geq a$,

$$\frac{\sqrt{8b - 6a} - \sqrt{2a}}{4} \leq \sqrt{\frac{\left(b - a\right)^2}{2b}}. \quad (24)$$

Combining (21), (23), and (24), $u = \sqrt{\mathrm{kl}\left(a \| b\right)}$ solves (22) implying (20). $\quad \square$

*Proof.* The inequalities (16) and (18) follow by plugging Corollary 2.5 into Theorems B.2 and B.3. For inequalities (17) and (19), we use (20). For (17), we use $a = E_D(h_t)$ and $b = E_S(h_t)$, which yields:

$$E_D(h_t) \leq E_S(h_t) + \sqrt{2E_S(h_t)\mathrm{kl}\left(E_S(h_t) \| E_D(h_t)\right)} + 2\mathrm{kl}\left(E_S(h_t) \| E_D(h_t)\right)$$
$$\leq E_S(h_t) + \sqrt{2E_S(h_t)\frac{\mathrm{KL}\left(p_0(\cdot; S) \| \nu\right) + \mathbb{E}_{p_0}\Psi_S(h_0) + \ln 2\sqrt{N}/\delta}{N}}$$
$$+ 2\frac{\mathrm{KL}\left(p_0(\cdot; S) \| \nu\right) + \mathbb{E}_{p_0}\Psi_S(h_0) + \ln 2\sqrt{N}/\delta}{N},$$

and similarly for (19). $\quad \square$

*Remark* B.6. Notice that when $h_t$ has a small training error $\mathbb{E}\left[E_S\left(h_t\right) \mid S\right] \approx 0$, the effective generalization gap decays as $O\left(1/N\right)$ instead of as $O\left(1/\sqrt{N}\right)$.

*Remark* B.7. In order to get the version in Theorem 2.7 we use the upper bound of Pinsker's inequality, i.e. that for all $a, b \in (0, 1)$

$$|a - b| \le \sqrt{\frac{1}{2}\mathrm{kl}\left(a \,\|\, b\right)},$$

and simplify $\ln \frac{2\sqrt{N}}{\delta} \le \ln \frac{N}{\delta}$ as $N \ge 8$.

Finally, we prove the equivalence statement made in Footnote 7:

**Claim B.8.** $\mathrm{KL}\left(p \,\|\, q\right) + \mathrm{KL}\left(q \,\|\, p\right) \le \beta$ *iff there exists a potential* $\Psi$ *such that* $p$ *is Gibbs w.r.t.* $q$ *with potential* $\Psi$ *and* $\mathbb{E}_q\Psi - \mathbb{E}_p\Psi \le \beta$, *and similarly* $D_\infty\left(p \,\|\, q\right) + D_\infty\left(q \,\|\, p\right) \le \beta$ *iff there exists a potential* $0 \le \Psi \le \beta$ *such that* $p$ *is Gibbs w.r.t.* $q$ *with potential* $\Psi$.

*Proof.* The first direction follows directly from Claim 2.3, so we only need to prove the converse. Assume that either $\mathrm{KL}\left(p \,\|\, q\right) + \mathrm{KL}\left(q \,\|\, p\right) \le \beta$, or $D_\infty\left(p \,\|\, q\right) + D_\infty\left(q \,\|\, p\right) \le \beta$. In these cases, both $\mathrm{d}p/\mathrm{d}q$ and $\mathrm{d}q/\mathrm{d}p$ exist, and for any measurable event $B$, $p\left(B\right) = 0 \iff q\left(B\right) = 0$, or equivalently, $p\left(B\right) > 0 \iff q\left(B\right) > 0$. Therefore, $\mathrm{supp}\left(p\right) = \mathrm{supp}\left(q\right)$, and $\mathrm{d}p/\mathrm{d}q > 0$ on $\mathrm{supp}\left(p\right)$. Denote $\Psi = -\ln \mathrm{d}p/\mathrm{d}q$, then $p$ is Gibbs w.r.t. $q$ with potential $\Psi$. The same derivation as in the proof of Claim 2.3 results in the bounds $\mathbb{E}_q\Psi - \mathbb{E}_p\Psi \le \beta$ and $\mathrm{ess\,sup}_q \Psi - \mathrm{ess\,inf}_p \Psi \le \beta$. In particular, if the latter holds then $\Psi$ can be shifted such that essentially $0 \le \Psi \le \beta$. $\qquad\square$

## C   Tightness and Necessity of the Divergence Conditions

If we are only interested in ensuring generalization at time $t \to \infty$, and when we converge to the stationary distribution $p_\infty$, then it is enough to bound the divergence $D\left(p_\infty \,\|\, \nu\right)$. If we are interested in bounding $D\left(p_t \,\|\, \nu\right)$ (and consequently, the generalization gap) at all times $t$, then we need also to limit $p_0$'s dependence on $S$, since $p_0$ (as well as $p_t$ for small $t$) can be completely different from a stationary $p_\infty$, and just bounding $D\left(p_\infty \,\|\, \nu\right)$ does not say anything about it. Bounding $D\left(p_0 \,\|\, \mu\right)$, for some data-independent distribution $\mu$, ensures generalization at $p_0$. This leaves the following questions regarding the proof of Theorem 2.7:

- Why do we need to bound the divergences $D\left(p_\infty \,\|\, \nu\right)$ and $D\left(p_0 \,\|\, \nu\right)$ from the same distribution $\nu$? That is, we do we need to require $\mu = \nu$? Bounding the divergences of $p_0$ and $p_\infty$ to two different divergences $\mu \neq \nu$ is sufficient to get generalization at the beginning (i.e. initialization) and end (i.e. after mixing)–is it sufficient for generalization in the middle (i.e. at any $t$)?

- Why do we need to also bound the reverse divergence $D\left(\nu \,\|\, p_\infty\right)$? I.e., why do we need to require $p_\infty$ is Gibbs w.r.t. $\nu$ with a bounded potential, instead of just controlling the divergence $D\left(p_\infty \,\|\, \nu\right)$, which is a weaker requirement and sufficient for generalization after mixing?

As we now show, both are necesairy, and without requiring both, i.e. if we drop either one of these, we cannot ensure generalization at intermediate times $t \geq 0$.

**Construction.**   Consider a supervised learning problem with $\mathcal{Z} = \mathcal{X} \times \mathcal{Y}$, $\mathcal{X} = [0, 1]$, $\mathcal{Y} = \{0, 1\}$, $\mathcal{H} = $ all measurable functions from $\mathcal{X}$ to $\mathcal{Y}$, and the zero-one loss $f(h, (x, y)) = \mathbb{I}\{h(x) \neq y\}$, with $D$ being the uniform distribution over $\mathcal{X}$, and $y$ being Bernoulli($\frac{1}{2}$) independent of $x$. For all $h$, $E_D(h) = 0.5$. Let $p_0$ be the constant zero function with probability $\frac{1}{2}$ and the constant one function with probability $\frac{1}{2}$. Consider the following deterministic $S$-dependent transition function over $h$: if $h_t$ is the constant zero function, then $h_{t+1} = h_S$ which memorizes $S$, i.e. $h_S(x) = y$ for $(x, y) \in S$, and $h_S(x) = 1$ otherwise. If $h_t$ is not the constant zero function, then $h_{t+1}$ is the constant ones function. We have that $p_\infty$ is deterministic at the constant one function, and KL$\left(p_\infty \,\|\, p_0\right) = \ln 2$, and in fact $p_t = p_\infty$ for $t > 1$. But with probability half, $h_1 = h_S$, for which for any sample size $N > 0$, $E_S(h_S) = 0$ while $E_D(h_S) = \frac{1}{2}$.

**How does this show it is not enough to bound $D\left(p_0 \,\|\, \nu\right)$ and $D\left(p_\infty \,\|\, \nu\right)$, but that we also need the reverse $D\left(\nu \,\|\, p_\infty\right)$?**   Since $p_0$ is data independent, we can take $\nu = p_0$, in which case KL$\left(p_0 \,\|\, \nu\right) = D_\infty\left(p_0 \,\|\, \nu\right) = 0$ and KL$\left(p_\infty \,\|\, \nu\right) = D_\infty\left(p_\infty \,\|\, \nu\right) = \ln 2$, but even as $N \to \infty$, the gap for $h_1$ does not diminish. Indeed, $D\left(\nu \,\|\, p_\infty\right) = \infty$, and so $p_\infty$ is not Gibbs w.r.t. $\nu$ and Theorem 2.7 does not apply.

**How does this show it is not enough to bound $D\left(p_\infty \,\|\, \nu\right) + D\left(\nu \,\|\, p_\infty\right)$ and $D\left(p_0 \,\|\, \mu\right)$ for $\mu \neq \nu$?**
Since in this example $p_\infty$ is also data independent, we can take $\nu = p_\infty$ and $\mu = p_0$, in which case $D\left(p_0 \,\|\, \mu\right) = 0$ and $D\left(p_\infty \,\|\, \nu\right) + D\left(\nu \,\|\, p_\infty\right) = 0$. We are indeed ensured a small gap for $h_0$ and $h_\infty$, but not for $h_1$.

# D Generalized Version of Corollary 3.1

We start by characterizing the stationary distributions of SDERs in a box with different noise scales $\sigma^2$. The stationary distributions for Gaussian initialization can be found similarly. Then, we extend Corollary 3.1 to scenarios where $p_0 \neq \nu$, as an immediate consequence of Theorem 2.7.

## D.1 Stationary distributions of CLD

We first derive the stationary distribution of SDERs of the form

$$d\mathbf{x}_t = -\nabla L\left(\mathbf{x}_t\right) dt + \sqrt{2\beta^{-1}\sigma^2\left(\mathbf{x}_t\right)}d\mathbf{w}_t + d\mathbf{r}_t\,, \tag{25}$$

with normal reflection in a box domain (for a full definition see (45)-(47) in Appendix H.2), where $L \geq 0$ is some $\mathcal{C}^2$ loss function, $\beta > 0$ is an inverse temperature parameter, and $\sigma^2$ is a diffusion coefficient. First, we present a well known characterization of the stationary distribution of (25).

**Lemma D.1.** *If $L, \sigma^2 \in \mathcal{C}^2$, $\sigma^2\left(\cdot\right) > 0$ is uniformly bounded away from 0 in $\overline{\Omega}$,*

$$Z = \int_{\overline{\Omega}} \frac{1}{\sigma^2\left(\mathbf{x}\right)} \exp\left(-\beta \int \frac{\nabla L\left(\mathbf{x}\right)}{\sigma^2\left(\mathbf{x}\right)} d\mathbf{x}\right) < \infty\,,$$

*the integrals exist, and the field $\nabla L/\sigma^2$ is conservative (curl-free), then*

$$p_\infty\left(\mathbf{x}\right) = \frac{1}{Z} \frac{1}{\sigma^2\left(\mathbf{x}\right)} \exp\left(-\beta \int \frac{\nabla L\left(\mathbf{x}\right)}{\sigma^2\left(\mathbf{x}\right)} d\mathbf{x}\right) \tag{26}$$

*is a stationary distribution of* (25).

For completeness, the proof is presented in Appendix H.2.1, following additional results and definitions in Appendix H. We can now calculate explicit stationary distributions for some choices of $\sigma^2$. Specifically, we focus on cases where $\sigma^2\left(\mathbf{x}\right) = g\left(L\left(\mathbf{x}\right)\right)$ for some scalar function $g$, as it guarantees the curl-free condition, and is convenient to integrate.

**Example D.2** (Uniform noise scale). Assuming that $\sigma^2\left(\mathbf{x}\right) \equiv 1$, the stationary distribution becomes the well-known Gibbs distribution

$$p_\infty\left(\mathbf{x}\right) = \frac{1}{Z} e^{-\beta L\left(\mathbf{x}\right)}\,, \tag{27}$$

so

$$\Psi_{\text{uniform}}\left(\mathbf{x}\right) = \beta L\left(\mathbf{x}\right)\,. \tag{28}$$

**Example D.3** (Linear noise scale). Let $\alpha > 0$, and suppose that $\sigma^2\left(\mathbf{x}\right) = \left(L\left(\mathbf{x}\right) + \alpha\right)$. Then

$$\frac{\nabla L\left(\mathbf{x}\right)}{\sigma^2\left(\mathbf{x}\right)} = \nabla \ln\left(L\left(\mathbf{x}\right) + \alpha\right)$$

so the stationary distribution is

$$p_\infty\left(\mathbf{x}\right) \propto \frac{1}{L\left(\mathbf{x}\right) + \alpha} \exp\left(-\beta \ln\left(L\left(\mathbf{x}\right) + \alpha\right)\right) = \frac{1}{L\left(\mathbf{x}\right) + \alpha}\left(L\left(\mathbf{x}\right) + \alpha\right)^{-\beta} = \left(L\left(\mathbf{x}\right) + \alpha\right)^{-\beta-1}\,, \tag{29}$$

which is integrable in a bounded domain. Recall that we want to represent $p_\infty$ using a potential $\Psi$ with $\inf \Psi \geq 0$. In this case, we can start from $\tilde{\Psi}\left(\mathbf{x}\right) = \left(\beta + 1\right) \ln\left(L\left(\mathbf{x}\right) + \alpha\right)$. Since $L \geq 0$ it clearly holds that $\tilde{\Psi} \geq \left(\beta + 1\right) \ln\left(\alpha\right)$, so we can use the shifted version

$$\Psi_{\text{linear}}\left(\mathbf{x}\right) = \left(\beta + 1\right)\left(\ln\left(L\left(\mathbf{x}\right) + \alpha\right) - \ln\left(\alpha\right)\right) = \left(\beta + 1\right) \ln\left(\frac{L\left(\mathbf{x}\right)}{\alpha} + 1\right)\,. \tag{30}$$

**Example D.4** (Polynomial noise scale). Let $\alpha > 0$, and $k > 1$. Suppose that $\sigma^2\left(\mathbf{x}\right) = \left(L\left(\mathbf{x}\right) + \alpha\right)^k$. Then

$$\frac{\nabla L\left(\mathbf{x}\right)}{\sigma^2\left(\mathbf{x}\right)} = \nabla L\left(\mathbf{x}\right)\left(L\left(\mathbf{x}\right) + \alpha\right)^{-k} = \frac{1}{1-k}\nabla\left(L\left(\mathbf{x}\right) + \alpha\right)^{1-k}$$

so

$$p_\infty\left(\mathbf{x}\right) \propto \left(L\left(\mathbf{x}\right) + \alpha\right)^{-k} \exp\left(\frac{\beta}{k-1}\left(L\left(\mathbf{x}\right) + \alpha\right)^{1-k}\right).$$

As before, the potential is monotonically increasing with $L\left(\mathbf{x}\right)$, so we can make a shift

$$\Psi_{\text{poly}} = k\ln\left(\frac{L\left(\mathbf{x}\right)}{\alpha} + 1\right) + \frac{\beta}{k-1}\left(\alpha^{1-k} - \left(L\left(\mathbf{x}\right) + \alpha\right)^{1-k}\right).$$

**Example D.5** (Exponential noise scale). Let $\alpha > 0$ and suppose that $\sigma^2\left(\mathbf{x}\right) = e^{\alpha L(\mathbf{x})}$. Then

$$\frac{\nabla L\left(\mathbf{x}\right)}{\sigma^2\left(\mathbf{x}\right)} = -\frac{1}{\alpha}\nabla\left(e^{-\alpha L(\mathbf{x})}\right)$$

so

$$p_\infty\left(\mathbf{x}\right) \propto e^{-\alpha L(\mathbf{x})}\exp\left(\frac{\beta}{\alpha}e^{-\alpha L(\mathbf{x})}\right) = \exp\left(\frac{\beta}{\alpha}e^{-\alpha L(\mathbf{x})} - \alpha L\left(\mathbf{x}\right)\right).$$

Denote $\psi\left(\tau\right) = \alpha\tau - \frac{\beta}{\alpha}e^{-\alpha\tau}$, then $\psi'\left(\tau\right) = \alpha + \beta e^{-\alpha\tau} \geq 0$. Therefore, $\min_{\tau \geq 0}\psi\left(\tau\right) = \psi\left(0\right) = -\frac{\beta}{\alpha}$, and we can take

$$\Psi_{\text{exp}}\left(\mathbf{x}\right) = \alpha L\left(\mathbf{x}\right) - \frac{\beta}{\alpha}e^{-\alpha L(\mathbf{x})} + \frac{\beta}{\alpha} = \alpha L\left(\mathbf{x}\right) + \frac{\beta}{\alpha}\left(1 - e^{-\alpha L(\mathbf{x})}\right) \tag{31}$$

## D.2 Generalization bounds

**Bounded domain with uniform initialization.** Assume that training follows a CLD in a bounded domain as described in (25) with uniform initialization $p_0 = \text{Uniform}\left(\Theta_0\right)$, where $\Theta_0 \subseteq \Theta$. For simplicity we take $\sigma^2 \equiv 1$. In that case Theorem 2.7 implies the following.

**Lemma D.6.** *Assume that the parameters evolve according to (25) with $\sigma^2 \equiv 1$ and uniform initialization $p_0 = \text{Uniform}\left(\Theta_0\right)$, where $\Theta_0 \subseteq \Theta$. Then for any time $t \geq 0$, and $\delta \in \left(0, 1\right)$,*

*1. w.p. $1 - \delta$ over $S \sim D^N$,*

$$\mathbb{E}_{\boldsymbol{\theta}_t \sim p_t}\left[E_D(\boldsymbol{\theta}_t) - E_S(\boldsymbol{\theta}_t) \mid S\right] \leq \sqrt{\frac{\beta\mathbb{E}_{p_0}\left[L_S(\boldsymbol{\theta}) \mid S\right] + \ln|\Theta|/|\Theta_0| + \ln\left(N/\delta\right)}{2N}}. \tag{32}$$

*2. w.p. $1 - \delta$ over $S \sim D^N$ and $\boldsymbol{\theta}_t \sim p_t$*

$$E_D(\boldsymbol{\theta}_t) - E_S(\boldsymbol{\theta}_t) \leq \sqrt{\frac{\beta\,\text{ess sup}_{p_0}L_S(\boldsymbol{\theta}) + \ln|\Theta|/|\Theta_0| + \ln\left(N/\delta\right)}{2N}}. \tag{33}$$

*Proof.* This is a direct corollary of Theorem 2.7 with $\text{KL}\left(p_0 \parallel \nu\right) = \ln|\Theta|/|\Theta_0|$. $\qquad\square$

$\ell^2$ **regularization with Gaussian initialization.** Let $\boldsymbol{\lambda} \in \mathbb{R}^d_{>0}$ be regularization terms, and consider the unconstrained SDE

$$d\boldsymbol{\theta}_t = -\nabla L\left(\boldsymbol{\theta}_t\right)dt - \beta^{-1}\text{diag}\left(\boldsymbol{\lambda}\right)\boldsymbol{\theta}_t dt + \sqrt{2\beta^{-1}\sigma^2\left(\boldsymbol{\theta}_t\right)}d\mathbf{w}_t. \tag{34}$$

Notice that $-\beta^{-1}\text{diag}\left(\boldsymbol{\lambda}\right)\boldsymbol{\theta}_t dt$ corresponds to an additive regularization of the form $\frac{1}{2\beta}\boldsymbol{\theta}_t^\top\text{diag}\left(\boldsymbol{\lambda}\right)\boldsymbol{\theta}_t$, so each parameter can have a different regularization coefficient. We shall denote by $\phi_{\boldsymbol{\lambda}}$ a multivariate Gaussian distribution with mean $\mathbf{0}$ and covariance matrix $\text{diag}\left(\boldsymbol{\lambda}^{-1}\right)$, where $\boldsymbol{\lambda}^{-1} = \left(\lambda_1^{-1}, \ldots, \lambda_d^{-1}\right)$. For simplicity, we present the results with $\sigma^2 \equiv 1$.

**Lemma D.7.** *Let $\boldsymbol{\lambda}_0, \boldsymbol{\lambda}_1 > 0$, and let $\boldsymbol{\theta}_t$ evolve according to (34) with $\sigma^2 \equiv 1$ and $\boldsymbol{\lambda} = \boldsymbol{\lambda}_1$, and start from a Gaussian initialization $p_0 = \phi_{\boldsymbol{\lambda}_0}$. Then for any time $t \geq 0$, and $\delta \in \left(0, 1\right)$,*

*1. w.p. $1 - \delta$ over $S \sim D^N$,*

$$\mathbb{E}_{\boldsymbol{\theta}_t \sim p_t}\left[E_D(\boldsymbol{\theta}_t) - E_S(\boldsymbol{\theta}_t) \mid S\right] \leq \sqrt{\frac{\beta\mathbb{E}_{p_0}\left[L_S(\boldsymbol{\theta}) \mid S\right] + \text{KL}\left(\phi_{\boldsymbol{\lambda}_0} \parallel \phi_{\boldsymbol{\lambda}_1}\right) + \ln\left(N/\delta\right)}{2N}}. \tag{35}$$

2. *w.p.* $1 - \delta$ *over* $S \sim D^N$ *and* $\boldsymbol{\theta}_t \sim p_t$

$$E_D(\boldsymbol{\theta}_t) - E_S(\boldsymbol{\theta}_t) \leq \sqrt{\frac{\beta \operatorname{ess\,sup}_{p_0} L_S(\boldsymbol{\theta}) + \mathrm{KL}\left(\phi_{\boldsymbol{\lambda}_0} \,\|\, \phi_{\boldsymbol{\lambda}_1}\right) + \ln\left(N/\delta\right)}{2N}}, \qquad (36)$$

*where* $\mathrm{KL}\left(\phi_{\boldsymbol{\lambda}_0} \,\|\, \phi_{\boldsymbol{\lambda}_1}\right) = \frac{1}{2} \sum_{i=1}^{d} \left( \ln\left(\frac{\lambda_{1,i}}{\lambda_{0,i}}\right) - 1 + \frac{\lambda_{0,i}}{\lambda_{1,i}} \right).$[9]

*Proof.* This is a direct corollary of Theorem 2.7 with the explicit expression for the KL divergence between two Gaussians. ☐

*Remark* D.8 (Dependence on the parameters' dimension). While the bound in Lemma D.7 depends on the dimension of the parameters $d$, this can be mitigated in practice. For example, by matching the regularization coefficient and initialization variance, the KL-divergence term vanishes and we lose the dependence on dimension. Furthermore, we can control each parameter separately by using parameter specific initialization variances and regularization coefficients. Then, the KL-divergence can have different dependencies, if any, on the dimension $d$.

---

[9]For $\boldsymbol{\lambda}_0 = \lambda_0 \mathbf{I}, \boldsymbol{\lambda}_1 = \lambda_1 \mathbf{I}, \lambda_0, \lambda_1 > 0$, this simplifies to $\mathrm{KL}\left(\phi_{\lambda_0} \,\|\, \phi_{\lambda_1}\right) = \frac{d}{2}\left(\ln\frac{\lambda_0}{\lambda_1} - 1 + \frac{\lambda_1}{\lambda_0}\right)$

# E   Linear Regression with CLD

Theorem 2.7 and Corollary 3.1 only bound the *gap* between the population and training errors, yet this does not necessarily bound the population error itself. One way to do this is by separately bounding the training error and showing that in the regime in which the generalization gap is small, the training error can be small as well. In Appendix F we show empirically that deep NNs can reach low training error when trained with SGLD in the regime in which Corollary 3.1 is not vacuous. Here, we look at the particular case of the asymptotic behavior of ridge regression with CLD training with Gaussian i.i.d. data, for which we can analytically study the training and population *losses*.

**Setup.**   Let $\boldsymbol{\theta}^\star \in \mathbb{R}^d$, $y = \mathbf{x}^\top \boldsymbol{\theta}^\star + \varepsilon$ with $\|\boldsymbol{\theta}^\star\| = 1$ and $\varepsilon \sim \mathcal{N}\left(0, \sigma^2\right)$ independent of $\mathbf{x}$. We assume that $\mathbf{x}$ has i.i.d. entries with $\mathbb{E}\mathbf{x} = \mathbf{0}$ and covariance $\mathbb{E}\left[\mathbf{x}\mathbf{x}^\top\right] = \mathbf{I}$. Let $\mathbf{X} \in \mathbb{R}^{N \times d}$ be the data (design) matrix, $\mathbf{y} \in \mathbb{R}^N$ the training targets, $\boldsymbol{\varepsilon} \in \mathbb{R}^N$ the pointwise perturbations, and $\boldsymbol{\theta} \in \mathbb{R}^d$ the parameters in a linear regression problem. In what follows, we focus on the overdetermined case $N > d$, where $\mathbf{X}$ has full column rank with probability 1, so the empirical covariance $\mathbf{A} = \frac{1}{N}\mathbf{X}^\top\mathbf{X} \succ 0$ a.s. In addition, we denote $\boldsymbol{\theta}_{\mathrm{LS}} = \frac{1}{N}\mathbf{A}^{-1}\mathbf{X}^\top\mathbf{y}$, and $\tilde{\boldsymbol{\theta}} = \boldsymbol{\theta} - \boldsymbol{\theta}_{\mathrm{LS}}$. The training objective is then the minimization of the regularized empirical loss

$$L_S\left(\boldsymbol{\theta}\right) + \frac{\lambda}{2\beta}\|\boldsymbol{\theta}\|^2 = \frac{1}{2N}\|\mathbf{X}\boldsymbol{\theta} - \mathbf{y}\|^2 + \frac{\lambda}{2\beta}\|\boldsymbol{\theta}\|^2 = \frac{1}{2}\tilde{\boldsymbol{\theta}}^\top\mathbf{A}\tilde{\boldsymbol{\theta}} + C_S + \frac{\lambda}{2\beta}\|\boldsymbol{\theta}\|^2 \,,$$

where $C_S = L_S\left(\boldsymbol{\theta}_{\mathrm{LS}}\right) = \frac{1}{2N}\|\mathbf{y}\|^2 - \frac{1}{2}\boldsymbol{\theta}_{\mathrm{LS}}\mathbf{A}\boldsymbol{\theta}_{\mathrm{LS}} = \frac{1}{2N}\|\mathbf{y}\|^2 - \frac{1}{2N}\mathbf{y}^\top\mathbf{X}\left(\mathbf{X}^\top\mathbf{X}\right)^{-1}\mathbf{X}^\top\mathbf{y}$, is the empirical irreducible error.

**CLD training.**   Assume that training is performed by CLD with inverse temperature $\beta > 0$, which, because $L_S$ is quadratic, takes the form

$$\mathrm{d}\boldsymbol{\theta}_t = -\mathbf{A}\left(\boldsymbol{\theta}_t - \boldsymbol{\theta}_{\mathrm{LS}}\right)\mathrm{d}t - \lambda\beta^{-1}\boldsymbol{\theta}_t\,\mathrm{d}t + \sqrt{\frac{2}{\beta}}\,\mathrm{d}\mathbf{w}_t\,. \tag{37}$$

Since $\mathbf{A} \succ 0$ and $\lambda > 0$, the Gibbs distribution

$$p_\infty\left(\boldsymbol{\theta}\right) \propto \exp\left(-\frac{1}{2}\left(\left(\boldsymbol{\theta} - \boldsymbol{\theta}_{\mathrm{LS}}\right)^\top \beta\mathbf{A}\left(\boldsymbol{\theta} - \boldsymbol{\theta}_{\mathrm{LS}}\right) + \lambda\boldsymbol{\theta}^\top\boldsymbol{\theta}\right)\right)$$

is the unique stationary distribution, and furthermore, it is the asymptotic distribution of (37). We can simplify this to a Gaussian. Denote $\alpha = \lambda/\beta$ and

$$\boldsymbol{\Sigma} = \frac{1}{\beta}\left(\mathbf{A} + \alpha\mathbf{I}\right)^{-1} \text{ and } \bar{\boldsymbol{\theta}} = \beta\boldsymbol{\Sigma}\mathbf{A}\boldsymbol{\theta}_{\mathrm{LS}} = \frac{1}{N}\left(\mathbf{A} + \alpha\mathbf{I}\right)^{-1}\mathbf{X}^\top\mathbf{y}\,,$$

then

$$
\begin{aligned}
\left(\boldsymbol{\theta} - \bar{\boldsymbol{\theta}}\right)^\top \boldsymbol{\Sigma}^{-1}\left(\boldsymbol{\theta} - \bar{\boldsymbol{\theta}}\right) &= \beta\boldsymbol{\theta}^\top\left(\mathbf{A} + \alpha\mathbf{I}\right)\boldsymbol{\theta} - 2\boldsymbol{\theta}^\top\boldsymbol{\Sigma}^{-1}\bar{\boldsymbol{\theta}} + \bar{\boldsymbol{\theta}}^\top\boldsymbol{\Sigma}^{-1}\bar{\boldsymbol{\theta}} \\
&= \beta\boldsymbol{\theta}^\top\left(\mathbf{A} + \alpha\mathbf{I}\right)\boldsymbol{\theta} - 2\beta\boldsymbol{\theta}^\top\boldsymbol{\Sigma}^{-1}\boldsymbol{\Sigma}\mathbf{A}\boldsymbol{\theta}_{\mathrm{LS}} + \beta^2\boldsymbol{\theta}_{\mathrm{LS}}^\top\mathbf{A}\boldsymbol{\Sigma}\boldsymbol{\Sigma}^{-1}\boldsymbol{\Sigma}\mathbf{A}\boldsymbol{\theta}_{\mathrm{LS}} \\
&= \beta\boldsymbol{\theta}^\top\left(\mathbf{A} + \alpha\mathbf{I}\right)\boldsymbol{\theta} - 2\beta\boldsymbol{\theta}^\top\mathbf{A}\boldsymbol{\theta}_{\mathrm{LS}} + \beta^2\boldsymbol{\theta}_{\mathrm{LS}}^\top\mathbf{A}\boldsymbol{\Sigma}\mathbf{A}\boldsymbol{\theta}_{\mathrm{LS}}\,.
\end{aligned}
$$

Since the last term is constant w.r.t. $\boldsymbol{\theta}$, we deduce that

$$p_\infty\left(\boldsymbol{\theta}\right) \propto \exp\left(-\frac{1}{2}\left(\boldsymbol{\theta} - \bar{\boldsymbol{\theta}}\right)^\top \boldsymbol{\Sigma}^{-1}\left(\boldsymbol{\theta} - \bar{\boldsymbol{\theta}}\right)\right)\,,$$

i.e. the stationary distribution is a Gaussian $\mathcal{N}\left(\bar{\boldsymbol{\theta}}, \boldsymbol{\Sigma}\right)$. We can now calculate the expected training and population losses.

**Goal.**   In the rest of this section, our final aim is to calculate the expected training and population losses in the setup described above, in the case when the data is sampled i.i.d. from standard Gaussian distribution, $\sigma$ is a fixed constant, $\lambda \propto d$ (to match standard initialization), [10] $N$, $\beta$ and $d$ are large, but

---

[10]Since this is a linear model $d = \mathrm{layer\ width}$, and as we assume the regularization matches the standard initialization. This initialization is considered in many works as a Bayesian prior in various settings [35, 68].

$\beta \ll N$, so our generalization bound is small (since $\mathbb{E}_{p_0} L$ is a fixed constant in this case). We will find (in Remark E.2 and Remark E.4) that if also $d \ll \beta$ then the training and expected population loss are not significantly degraded. This is not a major constraint, since we need $d \ll N$ to get good population loss anyway, even without noise (i.e. $\beta \to \infty$). This shows that in this regime $d \ll \beta \ll N$, the randomness required by our generalization bound (the KL bounds in Corollary 3.1) does not significantly harm the training loss or the expected population loss.

**Claim E.1.** *With some abuse of notation, denote $L_S(\boldsymbol{\theta}_\infty) = \mathbb{E}_{\boldsymbol{\theta} \sim p_\infty} L_S(\boldsymbol{\theta})$. Then*

$$\mathbb{E}\left[L_S(\boldsymbol{\theta}_\infty) \mid \mathbf{X}\right] = \frac{1}{2\beta} \operatorname{Tr}\left(\mathbf{A}(\mathbf{A} + \alpha\mathbf{I})^{-1}\right) + \frac{\alpha^2}{2} \boldsymbol{\theta}^{\star\top}(\mathbf{A} + \alpha\mathbf{I})^{-2} \mathbf{A}\boldsymbol{\theta}^\star$$

$$+ \frac{\sigma^2 \alpha^2}{2N} \operatorname{Tr}\left((\mathbf{A} + \alpha\mathbf{I})^{-2}\right) + \frac{\sigma^2}{2}\left(1 - \frac{d}{N}\right).$$

*Proof.* From Petersen and Pedersen [56] (equation 318)

$$L_S(\boldsymbol{\theta}_\infty) = \frac{1}{2}\mathbb{E}(\boldsymbol{\theta} - \boldsymbol{\theta}_{\mathrm{LS}})^\top \mathbf{A}(\boldsymbol{\theta} - \boldsymbol{\theta}_{\mathrm{LS}}) + C_S$$

$$= \frac{1}{2}\operatorname{Tr}(\mathbf{A}\boldsymbol{\Sigma}) + \frac{1}{2}(\bar{\boldsymbol{\theta}} - \boldsymbol{\theta}_{\mathrm{LS}})^\top \mathbf{A}(\bar{\boldsymbol{\theta}} - \boldsymbol{\theta}_{\mathrm{LS}}) + C_S.$$

For the second term, notice that

$$\bar{\boldsymbol{\theta}} - \boldsymbol{\theta}_{\mathrm{LS}} = (\beta\boldsymbol{\Sigma}\mathbf{A} - \mathbf{I})\boldsymbol{\theta}_{\mathrm{LS}}$$

$$= (\beta\boldsymbol{\Sigma}\mathbf{A} + \lambda\boldsymbol{\Sigma} - \lambda\boldsymbol{\Sigma} - \mathbf{I})\boldsymbol{\theta}_{\mathrm{LS}}$$

$$= \left(\boldsymbol{\Sigma}\underbrace{\beta(\mathbf{A} + \alpha\mathbf{I})}_{=\boldsymbol{\Sigma}^{-1}} - \lambda\boldsymbol{\Sigma} - \mathbf{I}\right)\boldsymbol{\theta}_{\mathrm{LS}}$$

$$= -\lambda\boldsymbol{\Sigma}\boldsymbol{\theta}_{\mathrm{LS}} = -\alpha(\mathbf{A} + \alpha\mathbf{I})^{-1}\boldsymbol{\theta}_{\mathrm{LS}}.$$

$\mathbf{A}$ and $\boldsymbol{\Sigma}$ are simultaneously diagonalizable. To see this, let $\mathbf{A} = \mathbf{Q}\boldsymbol{\Lambda}\mathbf{Q}^\top$ be a spectral decomposition of $\mathbf{A}$, then $\mathbf{A} + \alpha\mathbf{I} = \mathbf{Q}(\boldsymbol{\Lambda} + \alpha\mathbf{I})\mathbf{Q}^\top$, so $\boldsymbol{\Sigma} = \beta^{-1}\mathbf{Q}(\boldsymbol{\Lambda} + \alpha\mathbf{I})^{-1}\mathbf{Q}^\top$. This means that $\mathbf{A}$, $\boldsymbol{\Sigma}$, and their inverses all multiplicatively commute. Therefore,

$$L_S(\boldsymbol{\theta}_\infty) = \frac{1}{2}\operatorname{Tr}(\mathbf{A}\boldsymbol{\Sigma}) + \frac{\alpha^2}{2}\boldsymbol{\theta}_{\mathrm{LS}}^\top(\mathbf{A} + \alpha\mathbf{I})^{-1}\mathbf{A}(\mathbf{A} + \alpha\mathbf{I})^{-1}\boldsymbol{\theta}_{\mathrm{LS}} + C_S$$

$$= \frac{1}{2}\operatorname{Tr}(\mathbf{A}\boldsymbol{\Sigma}) + \frac{\alpha^2}{2N^2}\mathbf{y}^\top\mathbf{X}\mathbf{A}^{-1}(\mathbf{A} + \alpha\mathbf{I})^{-1}\mathbf{A}(\mathbf{A} + \alpha\mathbf{I})^{-1}\mathbf{A}^{-1}\mathbf{X}^\top\mathbf{y} + C_S$$

$$= \frac{1}{2\beta}\operatorname{Tr}\left(\mathbf{A}(\mathbf{A} + \alpha\mathbf{I})^{-1}\right) + \frac{\alpha^2}{2N^2}\mathbf{y}^\top\mathbf{X}(\mathbf{A} + \alpha\mathbf{I})^{-2}\mathbf{A}^{-1}\mathbf{X}^\top\mathbf{y} + C_S,$$

Conditioned on $\mathbf{X}$, standard results about the residuals in linear regression imply that,

$$\mathbb{E}[C_S \mid \mathbf{X}] = \frac{\sigma^2}{2}\left(1 - \frac{d}{N}\right).$$

In addition, for any symmetric matrix $\mathbf{M}$ we have

$$\mathbb{E}_\varepsilon\left[\mathbf{y}^\top\mathbf{M}\mathbf{y}\right] = \mathbb{E}_\varepsilon\left[(\mathbf{X}\boldsymbol{\theta}^\star + \varepsilon)^\top\mathbf{M}(\mathbf{X}\boldsymbol{\theta}^\star + \varepsilon)\right]$$

$$= (\mathbf{X}\boldsymbol{\theta}^\star)^\top\mathbf{M}\mathbf{X}\boldsymbol{\theta}^\star + \mathbb{E}_\varepsilon\left[\varepsilon^\top\mathbf{M}\varepsilon\right]$$

$$= (\mathbf{X}\boldsymbol{\theta}^\star)^\top\mathbf{M}\mathbf{X}\boldsymbol{\theta}^\star + \sigma^2\operatorname{Tr}(\mathbf{M}).$$

In particular,

$$\mathbb{E}\left[\mathbf{y}^\top\mathbf{X}(\mathbf{A} + \alpha\mathbf{I})^{-2}\mathbf{A}^{-1}\mathbf{X}^\top\mathbf{y} \mid \mathbf{X}\right]$$

$$= \boldsymbol{\theta}^{\star\top}\mathbf{X}^\top\mathbf{X}(\mathbf{A} + \alpha\mathbf{I})^{-2}\mathbf{A}^{-1}\mathbf{X}^\top\mathbf{X}\boldsymbol{\theta}^\star + \sigma^2\operatorname{Tr}\left(\mathbf{X}(\mathbf{A} + \alpha\mathbf{I})^{-2}\mathbf{A}^{-1}\mathbf{X}^\top\right)$$

$$= \boldsymbol{\theta}^{\star\top}N\mathbf{A}(\mathbf{A} + \alpha\mathbf{I})^{-2}\mathbf{A}^{-1}N\mathbf{A}\boldsymbol{\theta}^\star + \sigma^2\operatorname{Tr}\left(\mathbf{X}^\top\mathbf{X}(\mathbf{A} + \alpha\mathbf{I})^{-2}\mathbf{A}^{-1}\right)$$

$$= N^2\boldsymbol{\theta}^{\star\top}(\mathbf{A} + \alpha\mathbf{I})^{-2}\mathbf{A}\boldsymbol{\theta}^\star + N\sigma^2\operatorname{Tr}\left((\mathbf{A} + \alpha\mathbf{I})^{-2}\right),$$

where we used the definition of $\mathbf{A}$, the joint diagonalizability of $\mathbf{A}$ and $\boldsymbol{\Sigma}$, and the cyclicality of the trace. In total, the expected training loss, conditioned on the data is

$$\mathbb{E}_{\boldsymbol{\varepsilon}} L_S\left(\boldsymbol{\theta}_{\infty}\right) = \frac{1}{2\beta} \operatorname{Tr}\left(\mathbf{A}\left(\mathbf{A} + \alpha\mathbf{I}\right)^{-1}\right) + \frac{\alpha^2}{2}\boldsymbol{\theta}^{\star\top}\left(\mathbf{A} + \alpha\mathbf{I}\right)^{-2}\mathbf{A}\boldsymbol{\theta}^{\star}$$
$$+ \frac{\sigma^2\alpha^2}{2N}\operatorname{Tr}\left(\left(\mathbf{A} + \alpha\mathbf{I}\right)^{-2}\right) + \frac{\sigma^2}{2}\left(1 - \frac{d}{N}\right) .$$

$\square$

*Remark* E.2. We intuitively derive the asymptotic behavior of Claim E.1. Let $\lambda$ be constant, and let $\beta$ grow (so $\alpha$ shrinks). We can decompose $\left(\mathbf{A} + \alpha\mathbf{I}\right)^{-1}$ as

$$\left(\mathbf{A} + \alpha\mathbf{I}\right)^{-1} = \mathbf{A}^{-1} - \alpha\mathbf{A}^{-2} + \alpha^2\mathbf{A}^{-2}\left(\mathbf{A} + \alpha\mathbf{I}\right)^{-1} .$$

This can be readily verified as

$$\mathbf{A}^{-1} - \alpha\mathbf{A}^{-2} + \alpha^2\mathbf{A}^{-2}\left(\mathbf{A} + \alpha\mathbf{I}\right)^{-1}$$
$$= \mathbf{A}^{-2}\left(\mathbf{A} + \alpha\mathbf{I}\right)^{-1}\left(\mathbf{A}\left(\mathbf{A} + \alpha\mathbf{I}\right) - \alpha\left(\mathbf{A} + \alpha\mathbf{I}\right) + \alpha^2\mathbf{I}\right)$$
$$= \mathbf{A}^{-2}\left(\mathbf{A} + \alpha\mathbf{I}\right)^{-1}\left(\mathbf{A}^2 + \alpha\mathbf{A} - \alpha\mathbf{A} - \alpha^2\mathbf{I} + \alpha^2\mathbf{I}\right)$$
$$= \mathbf{A}^{-2}\left(\mathbf{A} + \alpha\mathbf{I}\right)^{-1}\mathbf{A}^2$$
$$= \left(\mathbf{A} + \alpha\mathbf{I}\right)^{-1} ,$$

where we used the multiplicative commutativity, as before. Notice that since $\mathbf{A} \succ 0$, $\mathbf{A} + \alpha\mathbf{I} \succ \mathbf{A}$, so $\left(\mathbf{A} + \alpha\mathbf{I}\right)^{-k} \prec \mathbf{A}^{-k}$ for any $k \in \mathbb{N}$. Denote

$$R_2\left(\alpha\right) = \alpha^2\mathbf{A}^{-2}\left(\mathbf{A} + \alpha\mathbf{I}\right)^{-1} ,$$

then $\|R_2\left(\alpha\right)\|_2 \leq \frac{\alpha^2}{\lambda_{\min}(\mathbf{A})^3}$, where $\lambda_{\min}\left(\mathbf{A}\right)$ is the minimal eigenvalue of $\mathbf{A}$. As the elements of $\mathbf{X}$ are i.i.d. with mean $0$ and variance $1$, the limiting distribution of the spectrum of $\mathbf{A}$ as $N, d \to \infty$ with $d/N \to \gamma \in (0, 1)$ is the Marchenko–Pastur distribution, which is supported on $\left[\left(1 - \sqrt{\gamma}\right)^2, \left(1 + \sqrt{\gamma}\right)^2\right]$. In particular, as $N, d \to \infty$, $\lambda_{\min}\left(\mathbf{A}\right) \geq \left(1 - \sqrt{d/N}\right)^2$, so for $\varepsilon > 0$,

$$\|R_2\left(\alpha\right)\|_2 \leq \frac{\alpha^2}{\left(1 - \sqrt{d/N} - \varepsilon\right)^6}$$

with high probability. Therefore, in the following we shall treat the remainder as $R_2\left(\alpha\right) = O\left(\alpha^2\right)$, even when taking the expectation over $\mathbf{X}$.

Since $\alpha = \lambda/\beta$ and $\lambda \propto d$, then for $d \leq \beta$, we have $\alpha/\beta = O\left(\alpha^2\right)$, and we conclude that

$$\mathbb{E}\left[L_S\left(\boldsymbol{\theta}_{\infty}\right) \mid \mathbf{X}\right] = \frac{d}{2}\left(\frac{1}{\beta} + \sigma^2\left(\frac{1}{d} - \frac{1}{N}\right)\right) + O\left(\alpha^2\right) .$$

Therefore, the added noise does not significantly hurt the training loss when $\frac{1}{\beta} \lesssim \sigma^2\left(\frac{1}{d} - \frac{1}{N}\right)$, or equivalently, $\beta \gtrsim \frac{Nd}{(N-d)\sigma^2}$. In particular, this holds when $d \ll \beta \ll N$, which is a regime where our generalization bound Corollary 3.1 also becomes small (since $\beta \ll N$). This shows that the randomness required by Corollary 3.1 can allow for successful optimization of the training loss.

Moving on to the population loss, we define $L_D$ in the usual way

$$L_D\left(\boldsymbol{\theta}_t\right) = \frac{1}{2}\mathbb{E}_{\mathbf{x},\varepsilon}\left(\mathbf{x}^{\top}\boldsymbol{\theta}_t - y\right)^2 = \frac{1}{2}\mathbb{E}\left(\mathbf{x}^{\top}\boldsymbol{\theta}_t - \mathbf{x}^{\top}\boldsymbol{\theta}^{\star} - \varepsilon\right)^2 .$$

Due to the independence between $\mathbf{x}$ and $\varepsilon$,

$$L_D\left(\boldsymbol{\theta}\right) = \frac{1}{2}\mathbb{E}\left(\mathbf{x}^{\top}\left(\boldsymbol{\theta} - \boldsymbol{\theta}^{\star}\right)\right)^2 + \frac{\sigma^2}{2} = \frac{1}{2}\|\boldsymbol{\theta} - \boldsymbol{\theta}^{\star}\|^2 + \frac{\sigma^2}{2} .$$

**Claim E.3.** *With some abuse of notation, denote $L_D\left(\boldsymbol{\theta}_\infty\right) = \mathbb{E}_{\boldsymbol{\theta}\sim p_\infty} L_D\left(\boldsymbol{\theta}\right)$. Then*

$$\mathbb{E}\left[L_D\left(\boldsymbol{\theta}_\infty\right) \mid \mathbf{X}\right] = \frac{1}{2\beta}\mathrm{Tr}\left(\left(\mathbf{A}+\alpha\mathbf{I}\right)^{-1}\right) + \frac{1}{2}\boldsymbol{\theta}^{\star\top}\mathbf{A}^2\left(\mathbf{A}+\alpha\mathbf{I}\right)^{-2}\boldsymbol{\theta}^\star$$

$$+ \frac{\sigma^2}{2N}\mathrm{Tr}\left(\mathbf{A}\left(\mathbf{A}+\alpha\mathbf{I}\right)^{-2}\right) - \boldsymbol{\theta}^{\star\top}\mathbf{A}\left(\mathbf{A}+\alpha\mathbf{I}\right)^{-1}\boldsymbol{\theta}^\star + \frac{1}{2}\left\|\boldsymbol{\theta}^\star\right\|^2 + \frac{\sigma^2}{2}.$$

*Proof.* Taking the expectation w.r.t $\boldsymbol{\theta}\sim\mathcal{N}\left(\bar{\boldsymbol{\theta}},\boldsymbol{\Sigma}\right)$ we get from Petersen and Pedersen [56]

$$L_D\left(\boldsymbol{\theta}_\infty\right) = \frac{1}{2}\mathrm{Tr}\left(\boldsymbol{\Sigma}\right) + \frac{1}{2}\left\|\bar{\boldsymbol{\theta}} - \boldsymbol{\theta}^\star\right\|^2 + \frac{\sigma^2}{2}$$

$$= \frac{1}{2\beta}\mathrm{Tr}\left(\left(\mathbf{A}+\alpha\mathbf{I}\right)^{-1}\right) + \frac{1}{2}\bar{\boldsymbol{\theta}}^\top\bar{\boldsymbol{\theta}} - \bar{\boldsymbol{\theta}}^\top\boldsymbol{\theta}^\star + \frac{1}{2}\left\|\boldsymbol{\theta}^\star\right\|^2 + \frac{\sigma^2}{2}.$$

We can simplify some of the terms when taking the expectation conditioned on $\mathbf{X}$.

$$\mathbb{E}_\varepsilon\left[\bar{\boldsymbol{\theta}}^\top\bar{\boldsymbol{\theta}}\right] = \frac{1}{N^2}\mathbb{E}\left[\mathbf{y}^\top\mathbf{X}\left(\mathbf{A}+\alpha\mathbf{I}\right)^{-1}\left(\mathbf{A}+\alpha\mathbf{I}\right)^{-1}\mathbf{X}^\top\mathbf{y}\right]$$

$$= \frac{1}{N^2}\mathbb{E}\left[\left(\mathbf{X}\boldsymbol{\theta}^\star + \varepsilon\right)^\top\mathbf{X}\left(\mathbf{A}+\alpha\mathbf{I}\right)^{-2}\mathbf{X}^\top\left(\mathbf{X}\boldsymbol{\theta}^\star + \varepsilon\right)\right]$$

$$= \frac{1}{N^2}\boldsymbol{\theta}^{\star\top}\mathbf{X}^\top\mathbf{X}\left(\mathbf{A}+\alpha\mathbf{I}\right)^{-2}\mathbf{X}^\top\mathbf{X}\boldsymbol{\theta}^\star + \frac{1}{N^2}\mathbb{E}_\varepsilon\left[\varepsilon^\top\mathbf{X}\left(\mathbf{A}+\alpha\mathbf{I}\right)^{-2}\mathbf{X}^\top\varepsilon\right]$$

$$= \boldsymbol{\theta}^{\star\top}\mathbf{A}^2\left(\mathbf{A}+\alpha\mathbf{I}\right)^{-2}\boldsymbol{\theta}^\star + \frac{\sigma^2}{N^2}\mathrm{Tr}\left(\mathbf{X}\left(\mathbf{A}+\alpha\mathbf{I}\right)^{-2}\mathbf{X}^\top\right)$$

$$= \boldsymbol{\theta}^{\star\top}\mathbf{A}^2\left(\mathbf{A}+\alpha\mathbf{I}\right)^{-2}\boldsymbol{\theta}^\star + \frac{\sigma^2}{N}\mathrm{Tr}\left(\mathbf{A}\left(\mathbf{A}+\alpha\mathbf{I}\right)^{-2}\right).$$

In addition,

$$\mathbb{E}_\varepsilon\left[\bar{\boldsymbol{\theta}}^\top\boldsymbol{\theta}^\star\right] = \frac{1}{N}\mathbb{E}_\varepsilon\left[\left(\mathbf{X}\boldsymbol{\theta}^\star + \varepsilon\right)^\top\mathbf{X}\left(\mathbf{A}+\alpha\mathbf{I}\right)^{-1}\boldsymbol{\theta}^\star\right]$$

$$= \frac{1}{N}\boldsymbol{\theta}^{\star\top}\mathbf{X}^\top\mathbf{X}\left(\mathbf{A}+\alpha\mathbf{I}\right)^{-1}\boldsymbol{\theta}^\star + \frac{1}{N}\mathbb{E}_\varepsilon\left[\varepsilon^\top\mathbf{X}\left(\mathbf{A}+\alpha\mathbf{I}\right)^{-1}\boldsymbol{\theta}^\star\right]$$

$$= \boldsymbol{\theta}^{\star\top}\mathbf{A}\left(\mathbf{A}+\alpha\mathbf{I}\right)^{-1}\boldsymbol{\theta}^\star.$$

Combining these we get the desired result. $\qquad\square$

*Remark* E.4. As we have done for the training loss in Remark E.2, we can estimate the expected population loss in some asymptotic regimes. Let $\lambda$ be constant, and let $\beta$ grow (so $\alpha$ shrinks). As in Remark E.2, we use the approximation $\left(\mathbf{A}+\alpha\mathbf{I}\right)^{-1} = \mathbf{A}^{-1} - \alpha\mathbf{A}^{-2} + O\left(\alpha^2\mathbf{I}\right)$, which also implies $\left(\mathbf{A}+\alpha\mathbf{I}\right)^{-2} = \mathbf{A}^{-2} - 2\alpha\mathbf{A}^{-3} + O\left(\alpha^2\mathbf{I}\right)$, and treat the remainders as $O\left(\alpha^2\right)$ even when taking the expectation w.r.t. $\mathbf{X}$. Then,

$$\mathbb{E}\left[L_D\left(\boldsymbol{\theta}_\infty\right) \mid \mathbf{X}\right] = \frac{1}{2\beta}\mathrm{Tr}\left(\mathbf{A}^{-1} - \alpha\mathbf{A}^{-2} + O\left(\alpha^2\mathbf{I}\right)\right) + \frac{1}{2}\boldsymbol{\theta}^{\star\top}\mathbf{A}^2\left(\mathbf{A}^{-2} - 2\alpha\mathbf{A}^{-3} + O\left(\alpha^2\mathbf{I}\right)\right)\boldsymbol{\theta}^\star$$

$$+ \frac{\sigma^2}{2N}\mathrm{Tr}\left(\mathbf{A}\left(\mathbf{A}^{-2} - 2\alpha\mathbf{A}^{-3} + O\left(\alpha^2\mathbf{I}\right)\right)\right)$$

$$- \boldsymbol{\theta}^{\star\top}\mathbf{A}\left(\mathbf{A}^{-1} - \alpha\mathbf{A}^{-2} + O\left(\alpha^2\mathbf{I}\right)\right)\boldsymbol{\theta}^\star + \frac{1}{2}\left\|\boldsymbol{\theta}^\star\right\|^2 + \frac{\sigma^2}{2}$$

$$= \frac{1}{2}\left(\frac{1}{\beta} + \frac{\sigma^2}{N}\right)\mathrm{Tr}\left(\mathbf{A}^{-1}\right) + \frac{\sigma^2}{2}$$

$$- \frac{\alpha}{2\beta}\mathrm{Tr}\left(\mathbf{A}^{-2} + O\left(\alpha\mathbf{I}\right)\right) - \alpha\boldsymbol{\theta}^{\star\top}\left(\mathbf{A}^{-1} + O\left(\alpha\mathbf{I}\right)\right)\boldsymbol{\theta}^\star$$

$$- \frac{\sigma^2\alpha}{N}\mathrm{Tr}\left(\mathbf{A}^{-2} + O\left(\alpha\mathbf{I}\right)\right) + \alpha\boldsymbol{\theta}^{\star\top}\left(\mathbf{A}^{-1} + O\left(\alpha\mathbf{I}\right)\right)\boldsymbol{\theta}^\star.$$

Simplifying, we arrive at

$$\mathbb{E}\left[L_D\left(\boldsymbol{\theta}_\infty\right) \mid \mathbf{X}\right] = \frac{1}{2}\left(\frac{1}{\beta} + \frac{\sigma^2}{N}\right)\mathrm{Tr}\left(\mathbf{A}^{-1}\right) + \frac{\sigma^2}{2} - \alpha\left(\frac{1}{2\beta} + \frac{\sigma^2}{N}\right)\mathrm{Tr}\left(\mathbf{A}^{-2}\right) + O\left(\alpha^2\right).$$

**Assuming** that $\mathbf{x}$ are i.i.d. $\mathcal{N}(\mathbf{0}, \mathbf{I})$, $N \cdot \mathbf{A} \sim \mathcal{W}_d(N, \mathbf{I})$, i.e. has a Wishart distribution. According to Theorem 3.3.16 of [24], if $N > d + 3$ then

$$\mathbb{E}\mathbf{A}^{-1} = \frac{N}{N - d - 1}\mathbf{I},$$

$$\mathbb{E}\mathbf{A}^{-2} = N^2 \cdot \frac{\text{Tr}(\mathbf{I})\mathbf{I}}{(N - d)(N - d - 1)(N - d - 3)} + N^2 \cdot \frac{\mathbf{I}}{(N - d)(N - d - 3)}$$

$$= \frac{N^2 d + N^2(N - d - 1)}{(N - d)(N - d - 1)(N - d - 3)}\mathbf{I}.$$

Then, the expectation over $\mathbf{X}$ and if $\frac{\sigma^2}{N} \lesssim \alpha$ (which is true for $\lambda \propto d$ and $\beta \ll N$ like we assume here),

$$\mathbb{E}L_D(\boldsymbol{\theta}_\infty) = \frac{1}{2}\left(\frac{1}{\beta} + \sigma^2\left(\frac{1}{N} + \frac{N - d - 1}{Nd}\right)\right) \cdot \frac{Nd}{N - d - 1} + O(\alpha^2)$$

$$= \frac{1}{2}\left(\frac{1}{\beta} + \sigma^2 \cdot \frac{N - 1}{Nd}\right) \cdot \frac{Nd}{N - d - 1} + O(\alpha^2).$$

This result is similar to the one in Remark E.2 — for the expected population loss not to be significantly hurt by the added noise, it must hold that $\beta \gtrsim \frac{Nd}{(N-1)\sigma^2}$. In particular, this holds when $d \ll \beta \ll N$, which is a regime where our generalization bound Corollary 3.1 also becomes small (since $\beta \ll N$). This shows that the randomness required by Corollary 3.1 does not harm the expected population loss.

## F  Numerical Experiments

### F.1  Experimental results

The following are results of training with SGLD (a discretized version of the CLD in (10)) on a few benchmark datasets. Notice we use the regularized version where regularization coefficient is $\lambda \cdot \beta^{-1}$ and the $\lambda$ hyperparameter is dictated by the initialization from the normal distribution $p_0 = \mathcal{N}\left(\mathbf{0}, \lambda^{-1}\mathbf{I}_d\right)$. We used a common initialization of $\mathcal{N}\left(\mathbf{0}, \frac{1}{d_{\mathrm{in}}}\right)$, i.e. $\lambda = d_{\mathrm{in}}$.

We use several different values of $\beta$ relative to $N$ (the number of training samples). For simplicity, we focused on binary classification cases. In all datasets with more than 2 classes, we constructed a binary classification task by partitioning the original label set into 2 disjoint sets of the same size.

The results demonstrate that learning with SGLD is possible with various values of $\beta$. In fact, in several instances, the injected noise appears to improve the *generalization gap*, e.g, in SVHN [53], in all the tested $\beta$ values between $0.4 \cdot N$ and $2 \cdot N$ the average test error remained almost the same while the training error decreased as $\beta$ increased (i.e. the generalization gap increased). Notably, we also observe that for sufficiently large levels of noise, the generalization bounds are non-vacuous.

Table 2: **MNIST** (binary classification)

| $\beta$ | $E_S$ | $E_D$ | $E_D - E_S$ | Bound (11) w.p 0.99 |
|---|---|---|---|---|
| $0.01 \cdot N$ | $0.2279(\pm 0.0021)$ | $0.1972(\pm 0.0243)$ | -0.0307 | 0.06124 |
| $0.03 \cdot N$ | $0.1161(\pm 0.0028)$ | $0.1074(\pm 0.0035)$ | -0.0087 | 0.10498 |
| $0.1 \cdot N$ | $0.0618(\pm 0.001)$ | $0.062(\pm 0.0041)$ | 0.0002 | 0.19096 |
| $0.15 \cdot N$ | $0.0497(\pm 0.0014)$ | $0.0494(\pm 0.0031)$ | -0.0003 | 0.23376 |
| $0.4 \cdot N$ | $0.0281(\pm 0.0002)$ | $0.0358(\pm 0.0029)$ | 0.0077 | 0.38147 |
| $0.7 \cdot N$ | $0.0202(\pm 0.0006)$ | $0.0284(\pm 0.0024)$ | 0.0082 | 0.50456 |
| $N$ | $0.0162(\pm 0.0006)$ | $0.0278(\pm 0.0023)$ | 0.0116 | 0.60302 |
| $2 \cdot N$ | $0.0092(\pm 0.0004)$ | $0.0262(\pm 0.0016)$ | 0.017 | 0.85273 |
| $\infty$ | $0.0001(\pm 0)$ | $0.0229(\pm 0.0004)$ | 0.0228 | $> 1$ |

Table 3:  **fashionMNIST** (binary classification)

| $\beta$ | $E_S$ | $E_D$ | $E_D - E_S$ | Bound (11) w.p 0.99 |
|---|---|---|---|---|
| $0.01 \cdot N$ | $0.1215(\pm 0.0027)$ | $0.1251(\pm 0.0087)$ | 0.0036 | 0.06833 |
| $0.03 \cdot N$ | $0.0999(\pm 0.001)$ | $0.1087(\pm 0.0167)$ | 0.0088 | 0.11738 |
| $0.1 \cdot N$ | $0.0821(\pm 0.0012)$ | $0.086(\pm 0.001)$ | 0.0039 | 0.21368 |
| $0.15 \cdot N$ | $0.0765(\pm 0.0009)$ | $0.0803(\pm 0.0015)$ | 0.0038 | 0.26159 |
| $0.4 \cdot N$ | $0.0635(\pm 0.0005)$ | $0.0722(\pm 0.002)$ | 0.0087 | 0.42695 |
| $0.7 \cdot N$ | $0.0567(\pm 0.0006)$ | $0.0691(\pm 0.0019)$ | 0.0124 | 0.56473 |
| $N$ | $0.0525(\pm 0.0005)$ | $0.0675(\pm 0.0013)$ | 0.015 | 0.67495 |
| $2 \cdot N$ | $0.043(\pm 0.0007)$ | $0.0672(\pm 0.0023)$ | 0.0242 | 0.95446 |
| $\infty$ | $0.0248(\pm 0.001)$ | $0.0675(\pm 0.0033)$ | 0.0427 | $> 1$ |

Table 4: **SVHN** (binary classification)

| $\beta$ | $E_S$ | $E_D$ | $E_D - E_S$ | Bound (11) w.p 0.99 |
|---|---|---|---|---|
| $0.01 \cdot N$ | $0.0746(\pm 0.0012)$ | $0.1033(\pm 0.0032)$ | 0.0287 | 0.05898 |
| $0.03 \cdot N$ | $0.0441(\pm 0.0004)$ | $0.067(\pm 0.0026)$ | 0.0229 | 0.10203 |
| $0.1 \cdot N$ | $0.0282(\pm 0.0008)$ | $0.0476(\pm 0.007)$ | 0.0194 | 0.1862 |
| $0.15 \cdot N$ | $0.0251(\pm 0.0005)$ | $0.0445(\pm 0.002)$ | 0.0194 | 0.22803 |
| $0.4 \cdot N$ | $0.0182(\pm 0.0005)$ | $0.0374(\pm 0.0017)$ | 0.0192 | 0.37235 |
| $0.7 \cdot N$ | $0.0146(\pm 0.0004)$ | $0.0363(\pm 0.002)$ | 0.0217 | 0.49256 |
| $N$ | $0.0124(\pm 0.0002)$ | $0.0342(\pm 0.0014)$ | 0.0218 | 0.58872 |
| $2 \cdot N$ | $0.0085(\pm 0)$ | $0.0371(\pm 0.001)$ | 0.0286 | 0.83256 |

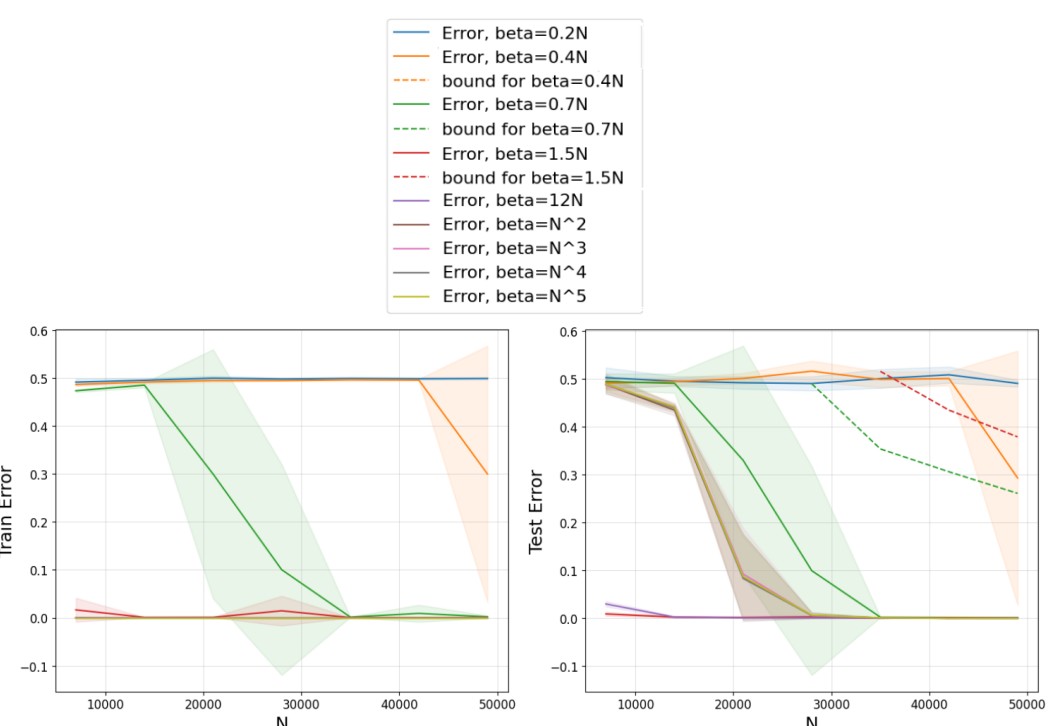

Figure 1: **Parity Results.** Left: Training error. Right: test error and generalization bound.

## F.2 Training details

**MNIST and fashionMNIST.**    We trained a fully connected network with 4 hidden layers of sizes $[256, 256, 256, 128]$ and ReLU activation, $lr = 0.01$, for 60 epochs.

**SVHN.**    The network was trained with a convolutional neural network with 5 convolutional layers, $lr = 0.01$, for 80 epochs. The complete architecture:

- Two convolutional layers (3×3 kernel, padding 1) with 32 channels, followed by ReLU activations and a 2×2 max pooling.

- Two convolutional layers (3×3 kernel, padding 1) with 64 channels, followed by ReLU activations and a 2×2 max pooling.

- A 3×3 convolution with 128 channels, ReLU, and 2×2 max pooling.

- 2 A linear layer $2048 \to 512$, followed by ReLU and another $512 \to 1$ linear layer

**Parity.**    In this experiment, we consider a synthetic binary classification task where each input is a binary vector of length 70 and the target label is defined as the parity of 3 randomly selected input dimensions. We train a neural network using SGLD with varying values of the inverse temperature parameter $\beta$ and different sample sizes.

The network was trained with a fully connected network with 4 hidden layers of sizes $[512, 1028, 2064, 512]$ and ReLU activation, $lr = 0.05$, for 100 epochs.

The results show that injecting noise can improve the generalization gap: specifically, the case of $\beta \geq N^2$ leads to overfitting, while smaller values of $\beta$ (e.g., $1.5 \cdot N$ to $12 \cdot N$) yield better generalization. Moreover, as well as in the benchmark datasets, in this setting, our generalization bound is non-vacuous in several cases.

### F.3 Comparison with the bound of Mou et al. [49]

The bound proposed by Mou et al. [49] has demonstrated non-vacuous results. To further assess the effectiveness of our bound and evaluate its relative tightness, we conducted a series of numerical experiments on the MNIST binary classification task (see Tables 5-8).

It is worth emphasizing that our bound offers a distinct advantage: it can be evaluated directly at initialization, whereas the bound of Mou et al. [49] depends on gradients and therefore cannot be computed before training. When testing their bound we used the continuous version, i.e.

$$\mathbb{E}_{p_T}[E_D(\boldsymbol{\theta}) - E_S(\boldsymbol{\theta})] \leq s \left( \frac{\beta}{2n} \int_0^T e^{\frac{\lambda}{2}(T-t)} \mathbb{E}_{p_t} \left[ \|\nabla L_S(\boldsymbol{\theta})\|^2 \right] dt + \frac{\log(1/\delta) + \log\log M}{n} \right)^{0.5}.$$

For simplicity, we omitted the term involving $M$ (which makes the bound more favorable). In addition, we set $s = 0.5$ since the zero–one loss (denoted here by $f(w)$, unlike [49]) is bounded within the interval $[0, 1]$. We observed that the relative tightness of the two bounds varies across different values of $\beta$ and at different points in time. Consequently, in some instances, the bound of Mou et al. [49] is tighter, while in others our bound performs better, and we could not draw any further conclusions.

Table 5: **20 training epochs**

| $\beta$ | Train Error | Test Error | Generalization Gap | Mou et al. [49] | Our bound |
|---|---|---|---|---|---|
| $0.03N = 1800$ | 0.1224 | 0.137 | 0.0146 | 0.0539 | 0.1144 |
| $0.15N = 9000$ | 0.0515 | 0.0747 | 0.0232 | 0.1279 | 0.2548 |
| $0.4N = 24000$ | 0.0335 | 0.058 | 0.0245 | 0.2845 | 0.4157 |
| $0.7N = 42000$ | 0.0278 | 0.0498 | 0.0220 | 0.4930 | 0.5499 |
| $N = 60000$ | 0.0249 | 0.0428 | 0.0179 | 0.7032 | 0.6572 |
| $2N = 120000$ | 0.0209 | 0.0356 | 0.0147 | 1.4044 | 0.9294 |

Table 6: **50 training epochs**

| $\beta$ | Train Error | Test Error | Generalization Gap | Mou et al. [49] | Our bound |
|---|---|---|---|---|---|
| $0.03N = 1800$ | 0.1156 | 0.1697 | 0.0541 | 0.0637 | 0.1144 |
| $0.15N = 9000$ | 0.0491 | 0.0615 | 0.0124 | 0.1324 | 0.2548 |
| $0.4N = 24000$ | 0.0295 | 0.0348 | 0.0053 | 0.2992 | 0.4157 |
| $0.7N = 42000$ | 0.0217 | 0.0283 | 0.0066 | 0.4903 | 0.5499 |
| $N = 60000$ | 0.0173 | 0.0277 | 0.0104 | 0.6827 | 0.6572 |
| $2N = 120000$ | 0.0108 | 0.0265 | 0.0157 | 1.3153 | 0.9294 |

Table 7: **250 training epochs**

| $\beta$ | Train Error | Test Error | Generalization Gap | Mou et al. [49] | Our bound |
|---|---|---|---|---|---|
| $0.03N = 1800$ | 0.122 | 0.1049 | $-0.0171$ | 0.1273 | 0.1144 |
| $0.15N = 9000$ | 0.0502 | 0.0476 | $-0.0026$ | 0.1503 | 0.2548 |
| $0.4N = 24000$ | 0.0284 | 0.0296 | 0.0011 | 0.2853 | 0.4157 |
| $0.7N = 42000$ | 0.0178 | 0.0247 | 0.0069 | 0.4595 | 0.5499 |
| $N = 60000$ | 0.0127 | 0.0240 | 0.0113 | 0.6478 | 0.6572 |
| $2N = 120000$ | 0.0050 | 0.0234 | 0.0184 | 1.2158 | 0.9294 |

Table 8: **400 training epochs**

| $\beta$ | Train Error | Test Error | Generalization Gap | Mou et al. [49] | Our bound |
|---|---|---|---|---|---|
| $0.03N = 1800$ | 0.1224 | 0.1105 | $-0.0119$ | 0.1900 | 0.1144 |
| $0.15N = 9000$ | 0.0499 | 0.0556 | 0.0057 | 0.1774 | 0.2548 |
| $0.4N = 24000$ | 0.0261 | 0.0357 | 0.0096 | 0.3005 | 0.4157 |
| $0.7N = 42000$ | 0.0161 | 0.0271 | 0.0110 | 0.4548 | 0.5499 |
| $N = 60000$ | 0.0112 | 0.0255 | 0.0143 | 0.6247 | 0.6572 |
| $2N = 120000$ | 0.0038 | 0.0249 | 0.0211 | 1.1455 | 0.9294 |

## G   Mild Overparametrization Prevents Uniform Convergence

In this section, we consider fully-connected ReLU networks, where the weights are bounded, such that for each layer $j$ the absolute values of all weights are bounded by $\frac{1}{\sqrt{d_{j-1}}}$, where $d_{j-1}$ is the width of layer $j-1$. Moreover, we assume that the input $\mathbf{x}$ is such that each coordinate $x_i$ is bounded in $[-1, 1]$. We show that $m$ training examples do not suffice for learning constant depth networks with $O(m)$ parameters. Thus, even a mild overparameterization prevents uniform convergence in our setting.

Our result follows by bounding the fat-shattering dimension, defined as follows:

**Definition G.1.** Let $\mathcal{F}$ be a class of real-valued functions from an input domain $\mathcal{X}$. We say that $\mathcal{F}$ shatters $m$ points $\{\mathbf{x}_i\}_{i=1}^m \subseteq \mathcal{X}$ with margin $\epsilon > 0$ if there are $r_1, \ldots, r_m \in \mathbb{R}$ such that for all $y_1, \ldots, y_m \in \{0, 1\}$ there exists $f \in \mathcal{F}$ such that

$$\forall i \in [m], \ f(\mathbf{x}_i) \leq r_i - \epsilon \ \text{if} \ y_i = 0 \ \text{and} \ f(\mathbf{x}_i) \geq r_i + \epsilon \ \text{if} \ y_i = 1 .$$

The fat-shattering dimension of $\mathcal{F}$ with margin $\epsilon$ is the maximum cardinality $m$ of a set of points in $\mathcal{X}$ for which the above holds.

The fat-shattering dimension of $\mathcal{F}$ with margin $\epsilon$ lower bounds the number of samples needed to learn $\mathcal{F}$ within accuracy $\epsilon$ in the distribution-free setting (see, e.g., [2, Part III]). Hence, to lower bound the sample complexity by some $m$ it suffices to show that we can shatter a set of $m$ points with a constant margin.

**Theorem G.2.** *We can shatter $m$ points $\{\mathbf{x}_i\}_{i=1}^m$ where $\|\mathbf{x}_i\|_\infty \leq 1$, with margin 1, using ReLU networks of constant depth and $O(m)$ parameters, such that for each layer $j$ the absolute values of all weights are bounded by $\frac{1}{\sqrt{d_{j-1}}}$, where $d_{j-1}$ is the width of layer $j-1$.*

*Proof.* Consider input dimension $d_0 = 1$. For $1 \leq i \leq m$, consider the points $x_i = \frac{i}{m}$, and let $\{y_i\}_{i=1}^m \subseteq \{0, 1\}$. Consider the following one-hidden-layer ReLU network $N$, which satisfies $N(x_i) = \frac{y_i}{m}$ for all $i$. First, the network $N$ includes a neuron with weight 0 and bias $\frac{y_1}{m}$, i.e., $[0 \cdot x + \frac{y_1}{m}]_+$. Now, for each $i$ such that $y_i = 0$ and $y_{i+1} = 1$ we add two neurons: $[x - y_i]_+ - [x - y_{i+1}]_+$, and for $i$ such that $y_i = 1$ and $y_{i+1} = 0$ we add $-[x - y_i]_+ + [x - y_{i+1}]_+$. It is easy to verify that this construction has width at most $2m - 1$ and allows us to shatter $m$ points with margin $\frac{1}{2m}$. However, the output weights of the neurons are $\pm 1$, and thus it does not satisfy the theorem's requirement. Consider the network $N'(x) = N(x) \cdot \frac{1}{\sqrt{2m-1}}$ obtained from $N$ by modifying the output weights. The network $N'$ satisfies the theorem's requirement on the weight magnitudes, and allows for shattering with margin $\frac{1}{2m\sqrt{2m-1}}$. We will now show how to increase this margin to 1 using a constant number of additional layers.

Let $\tilde{N}$ be a network obtained from $N'$ as follows. First, we add a ReLU activation to the output neuron of $N'$. Since for every $x_i$ we have $N'(x_i) \geq 0$, it does not affect these outputs. Next, we add $L = 8$ additional layers (layers $3, \ldots, 3 + L - 1$) of width $\sqrt{m}$ and without bias terms, where the incoming weights to layer 3 are all 1 and the weights in layers $4, \ldots, 3 + L - 1$ are $\frac{1}{m^{1/4}}$. Finally, we add an output neuron (layer $3 + L$) with incoming weights $\frac{1}{m^{1/4}}$. The network $\tilde{N}$ satisfies the theorem's requirements on the weight magnitudes, and it has depth $3 + L = 11$ and $O(m)$ parameters. Now, suppose that all neurons in a layer $3 \leq j \leq 3 + L - 1$ have values (i.e., activations) $z \geq 0$, then the values of all neurons in layer $j + 1$ are $z \cdot \frac{1}{m^{1/4}} \cdot \sqrt{m} = z \cdot m^{1/4}$.

Hence, if the value of the neuron in layer 2 is $\frac{1}{2m\sqrt{2m-1}}$, then the output of the network $\tilde{N}$ is $\frac{1}{2m\sqrt{2m-1}} \cdot (m^{1/4})^L = \frac{m^{L/4}}{2m\sqrt{2m-1}} = \frac{m^2}{2m\sqrt{2m-1}} \geq 2$ for large enough $m$. If the value of the neuron in layer 2 is 0 then the output of $\tilde{N}$ is also 0. Hence, this construction allow for shattering $m$ points with margin at least 1, using $O(m)$ parameters and weights that satisfy the theorem's conditions. $\square$

# H  Background on Stochastic Differential Equations with Reflection

We supply an introduction to the theory of stochastic differential equations with reflection (SDERs), then proceed to characterize the stationary distribution of a family of SDERs in a box. The background of standard (non-reflective) SDEs is similar and more common, and is therefore not included here. See for example [54] for more.

## H.1  SDEs with reflection

One of the main analytical tools of this work is the characterization of stationary distributions of SDER in bounded domains (see 57, 63, for an introduction).

The purpose of this section is to present more rigorously the setting of the paper, and supply the relevant definitions and results required to arrive at Lemma D.1. As Lemma D.1 is considered a well-known result, this section is mainly intended for completeness. Specifically, in the following we present some relevant definitions and results by Kang and Ramanan [31, 32], and specifically, ones that relate solutions to SDERs (Definition 2.4 in [32]), to solutions to sub-martingale problems (Definition 2.9 in [32]), and that characterize the stationary distributions of such solutions. For simplicity, we sometimes do not state the results in full generality.

**Setting.** Let $\Omega \subset \mathbb{R}^d$ be a domain (non-empty, connected, and open). Let the drift term $\mathbf{b} : \mathbb{R}^d \to \mathbb{R}^d$ and dispersion coefficient $\mathbf{\Sigma} : \mathbb{R}^d \to \mathbb{R}^{d \times d}$ be measurable and locally bounded. We also denote the diffusion coefficient by $\mathbf{A}(\cdot) = \mathbf{\Sigma}(\cdot) \mathbf{\Sigma}(\cdot)^\top = (a_{ij}(\cdot))_{i,j=1}^d$, and denote its columns by $\mathbf{a}_i(\cdot)$. We say that the diffusion coefficient is uniformly elliptic if there exists $\sigma > 0$ such that

$$\forall \mathbf{v} \in \mathbb{R}^d, \ \forall \mathbf{x} \in \overline{\Omega} \quad \mathbf{v}^\top \mathbf{A}(\mathbf{x}) \mathbf{v} > \sigma \|\mathbf{v}\| . \tag{38}$$

Let $\eta$ be a set valued mapping of allowed reflection directions defined on $\overline{\Omega}$ such that $\eta(\mathbf{x}) = \{\mathbf{0}\}$ for $\mathbf{x} \in \Omega$, and $\eta(\mathbf{x})$ is a non-empty, closed and convex cone in $\mathbb{R}^d$ such that $\{\mathbf{0}\} \subseteq \eta(\mathbf{x})$ for $\mathbf{x} \in \partial\Omega$, and furthermore assume that the set $\{(\mathbf{x}, \mathbf{v}) : \mathbf{x} \in \overline{\Omega}, \mathbf{v} \in \eta(\mathbf{x})\}$ is closed in $\mathbb{R}^{2d}$. In addition, for $\mathbf{x} \in \partial\Omega$ let $\hat{n}(\mathbf{x})$ be the set of inwards normals to $\Omega$ at $\mathbf{x}$,

$$\hat{n}(\mathbf{x}) = \bigcup_{r>0} \hat{n}_r(\mathbf{x}) ,$$

$$\hat{n}_r(\mathbf{x}) = \left\{ \mathbf{n} \in \mathbb{R}^d \mid \|\mathbf{n}\| = 1, \ B_r(\mathbf{x} - r\mathbf{n}) \cap \Omega = \emptyset \right\} .$$

Then, denote the set of boundary points with inward pointing cones

$$\mathcal{U} \triangleq \left\{ \mathbf{x} \in \partial\Omega \mid \exists \mathbf{n} \in \hat{n}(\mathbf{x}) : \forall \boldsymbol{\eta} \in \eta(\mathbf{x}) \ \langle \mathbf{n}, \boldsymbol{\eta} \rangle > 0 \right\} ,$$

and let $\mathcal{V} \triangleq \partial\Omega \setminus \mathcal{U}$. For example, if $\Omega$ is a convex polyhedron and $\eta(\mathbf{x})$ is the cone defined by the positive span of $\hat{n}(\mathbf{x})$ we get that $\mathcal{V} = \emptyset$.

Throughout this section and the rest of the paper, the *stochastic differential equation with reflection* (SDER) in $(\Omega, \eta)$

$$d\mathbf{x}_t = \mathbf{b}(\mathbf{x}_t) dt + \mathbf{\Sigma}(\mathbf{x}_t) d\mathbf{w}_t + d\mathbf{r}_t , \tag{39}$$

where $\mathbf{w}_t$ is a Wiener process, and $\mathbf{r}_t$ is a reflection process with respect to some filtration, is understood as in Definition 2.4 of [32], and the *submartingale problem* associated with $(\Omega, \eta), \mathcal{V}, \mathbf{b}$ and $\mathbf{\Sigma}$, refers to Definition 2.9 of [32]. In addition, we use the following definition.

**Definition H.1** (Piecewise $\mathcal{C}^2$ with continuous reflection; Definition 2.11 in [32]). The pair $(\Omega, \eta)$ is said to be piecewise $\mathcal{C}^2$ with continuous reflection if it satisfies the following properties:

1. $\Omega$ is a non-empty domain in $\mathbb{R}^d$ with representation

$$\Omega = \bigcap_{i \in \mathcal{I}} \Omega^i ,$$

where $\mathcal{I}$ is a finite set and for each $i \in \mathcal{I}$, $\Omega^i$ is a non-empty domain with $\mathcal{C}^2$ boundary in the sense that for each $\mathbf{x} \in \partial\Omega$, there exist a neighborhood $\mathcal{N}(\mathbf{x})$ of $\mathbf{x}$, and functions $\varphi_{\mathbf{x}}^i \in \mathcal{C}^2(\mathbb{R}^d)$, $i \in \mathcal{I}(\mathbf{x}) = \{i \in \mathcal{I} \mid \mathbf{x} \in \partial\Omega^i\}$, such that

$$\mathcal{N}(\mathbf{x}) \cap \Omega^i = \left\{ \mathbf{z} \in \mathcal{N}(\mathbf{x}) \mid \varphi_{\mathbf{x}}^i(\mathbf{z}) > 0 \right\} , \ \mathcal{N}(\mathbf{x}) \cap \partial\Omega^i = \left\{ \mathbf{z} \in \mathcal{N}(\mathbf{x}) \mid \varphi_{\mathbf{x}}^i(\mathbf{z}) = 0 \right\} ,$$

and $\nabla \varphi_{\mathbf{x}}^i \neq \mathbf{0}$ on $\mathcal{N}(\mathbf{x})$. For each $\mathbf{x} \in \partial \Omega^i$ and $i \in \mathcal{I}(\mathbf{x})$, let

$$\mathbf{n}^i(\mathbf{x}) = \frac{\nabla \varphi_{\mathbf{x}}^i}{\|\nabla \varphi_{\mathbf{x}}^i\|}$$

denote the unit inward normal vector to $\partial \Omega^i$ at $\mathbf{x}$.

2. The (set-valued) direction "vector field" $\eta : \overline{\Omega} \to \mathbb{R}^d$ is given by

$$\eta(\mathbf{x}) = \begin{cases} \{\mathbf{0}\} & \mathbf{x} \in \Omega, \\ \left\{ \sum_{i \in \mathcal{I}(\mathbf{x})} \alpha_i \boldsymbol{\eta}^i(\mathbf{x}) \mid \alpha_i \geq 0, \ i \in \mathcal{I}(\mathbf{x}) \right\} & \mathbf{x} \in \partial \Omega, \end{cases} \tag{40}$$

where for each $i \in \mathcal{I}$, $\boldsymbol{\eta}^i(\cdot)$ is a continuous unit vector field defined on $\partial \Omega^i$ that satisfies for all $\mathbf{x} \in \partial \Omega^i$

$$\langle \mathbf{n}^i(\mathbf{x}), \boldsymbol{\eta}^i(\mathbf{x}) \rangle > 0.$$

If $\eta^i(\cdot)$ is constant for every $i \in \mathcal{I}$, the the pair $(\Omega, \eta)$ is said to be piecewise $\mathcal{C}^2$ with constant reflection. If, in addition, $\mathbf{n}^i(\cdot)$ is constant for every $i \in \mathcal{I}$, then the pair $(\Omega, \eta)$ is said to be polyhedral with piecewise constant reflection.

In addition, let $\mathcal{S}$ denote the smooth parts of $\partial \Omega$.

*Remark* H.2. It is clear from the definition that if $\Omega$ is polyhedral, i.e. if all $\Omega^i$'s are half-spaces, and $\eta$ consists of inward normal reflections, then $(\Omega, \eta)$ is polyhedral with piecewise constant reflection.

**Theorem H.3** (Theorem 3 in [31], simplified)**.** *Suppose that the pair $(\Omega, \eta)$ is piecewise $\mathcal{C}^2$ with continuous reflection, for all $i \in \mathcal{I}$ and $\mathbf{x} \in \partial \Omega^i$, $\langle \mathbf{n}^i(\mathbf{x}), \boldsymbol{\eta}^i(\mathbf{x}) \rangle = 1$, $\mathcal{V} = \emptyset$, $\mathbf{b}(\cdot) \in \mathcal{C}^1(\overline{\Omega})$ and $\mathbf{A} \in \mathcal{C}^2(\overline{\Omega})$ (elementwise), and the submartingale problem associated with $(\Omega, \eta)$ and $\mathcal{V}$ is well posed. Furthermore, suppose there exists a nonnegative function $p \in \mathcal{C}^2(\overline{\Omega})$ with $Z_p = \int_{\overline{\Omega}} p(\mathbf{x}) \, d\mathbf{x} < \infty$ that solves the PDE defined by the following three relations:*

1. *For $\mathbf{x} \in \Omega$:*

$$0 = \frac{1}{2} \sum_{i,j=1}^d \frac{\partial^2}{\partial x_i \partial x_j} (a_{ij}(\mathbf{x}) p(\mathbf{x})) - \sum_{i=1}^d \frac{\partial}{\partial x_i} (b_i(\mathbf{x}) p(\mathbf{x})). \tag{41}$$

2. *For each $i \in \mathcal{I}$ and $\mathbf{x} \in \partial \Omega \cap \mathcal{S}$,*

$$0 = -2p(\mathbf{x}) \langle \mathbf{n}^i(\mathbf{x}), \mathbf{b}(\mathbf{x}) \rangle + \mathbf{n}^i(\mathbf{x})^\top \mathbf{A}(\mathbf{x}) \nabla p(\mathbf{x}) - \nabla \cdot (p(\mathbf{x}) \mathbf{q}^i(\mathbf{x})) + p(\mathbf{x}) K_i(\mathbf{x}), \tag{42}$$

*where*

$$\mathbf{q}^i(\mathbf{x}) \triangleq \mathbf{n}^i(\mathbf{x})^\top \mathbf{A}(\mathbf{x}) \mathbf{n}^i(\mathbf{x}) \boldsymbol{\eta}^i(\mathbf{x}) - \mathbf{A}(\mathbf{x}) \mathbf{n}^i(\mathbf{x})$$

*and*

$$K_i(\mathbf{x}) \triangleq \langle \mathbf{n}^i(\mathbf{x}), \nabla \cdot \mathbf{A}(\mathbf{x}) \rangle = \sum_{k=1}^d \mathbf{n}^i(\mathbf{x})_k \sum_{j=1}^d \frac{\partial a_{kj}}{\partial x_j}(\mathbf{x}).$$

3. *For each $i, j \in \mathcal{I}$, $i \neq j$, and $\mathbf{x} \in \partial \Omega^i \cap \partial \Omega^j \cap \partial \Omega$,*

$$p(\mathbf{x}) \left( \langle \mathbf{q}^i(\mathbf{x}), \mathbf{n}^j(\mathbf{x}) \rangle + \langle \mathbf{q}^j(\mathbf{x}), \mathbf{n}^i(\mathbf{x}) \rangle \right) = 0. \tag{43}$$

*Then the probability measure on $\overline{\Omega}$ defined by*

$$p_\infty(A) \triangleq \frac{1}{Z_p} \int_A p(\mathbf{x}) \, d\mathbf{x}, \quad A \in \mathcal{B}(\overline{\Omega}), \tag{44}$$

*is a stationary distribution for the well-posed submartingale problem.*

We are now ready to state a characterization of stationary distributions of (39). Note that for simplicity, we do not maintain full generality.

**Corollary H.4** (Stationary distribution of weak solutions to SDERs). *Suppose that, $\Omega$ is convex and bounded, $\mathbf{b} \in \mathcal{C}^1\left(\overline{\Omega}\right)$ and $\mathbf{A} \in \mathcal{C}^2\left(\overline{\Omega}\right)$, $(\Omega, \eta)$ is piecewise $\mathcal{C}^2$ with continuous reflection, $\mathbf{A}$ is uniformly elliptic (see (38)), and $\mathcal{V} = \emptyset$. Then $p \in \mathcal{C}^2$ satisfying the conditions in Theorem H.3 defines a stationary distribution for (39).*

*Proof.* Assumptions compactness of the domain, and continuous differentiability of the drift and dispersion coefficient imply that they are Lipschitz, hence Exercise 2.5.1 and Theorem 2.5.4 of [57] imply that there exists a unique strong solution to the SDER (39). Then, piecewise $\mathcal{C}^2$ with continuous reflection, the uniform ellipticity assumption, Theorems 1 and 3 of [32], and Theorem H.3 imply that if there exists $p \in \mathcal{C}^2$ satisfying (41)-(43), then (44) is a stationary distributions of (39). $\qquad\square$

In the next subsection we use this to derive explicit expressions for the stationary distribution in the setting of this paper.

## H.2 SDER with isotropic diffusion in a box

We proceed to assume that the diffusion term is a scalar matrix of the form $\mathbf{A}\left(\mathbf{x}\right) = 2\sigma^2\left(\mathbf{x}\right)\mathbf{I}_d$, and that $\Omega$ is a bounded box in $\mathbb{R}^d$, i.e. there exist $\{m_i < M_i\}_{i=1}^d$ such that

$$\Omega = \prod_{i=1}^d (m_i, M_i) = \bigcap_{i=1}^d \left(\Omega_m^i \cap \Omega_M^i\right), \tag{45}$$

where

$$\Omega_m^i \triangleq \left\{\mathbf{x} \in \mathbb{R}^d \mid x_i > m_i\right\}, \ \Omega_M^i \triangleq \left\{\mathbf{x} \in \mathbb{R}^d \mid x_i < M_i\right\}, \tag{46}$$

and that the reflecting field is normal to the boundary, i.e. given by (40) with

$$\boldsymbol{\eta}_m^i \equiv \mathbf{n}_m^i \equiv \mathbf{e}_i, \ \text{and} \ \boldsymbol{\eta}_M^i \equiv \mathbf{n}_M^i \equiv -\mathbf{e}_i \tag{47}$$

for $i = 1, \ldots, d$. In this setting, we can considerably simplify the conditions in Theorem H.3, as done in the following corollary.

**Lemma H.5** (Stationarity condition for SDER in a box with normal reflection). *Let $\mathbf{b}\left(\cdot\right) \in \mathcal{C}^1$, and let $\sigma\left(\cdot\right) \in \mathcal{C}^2$ be uniformly bounded away from 0, i.e. there exists $\underline{\sigma}^2 > 0$ such that for all $\mathbf{x} \in \overline{\Omega}$, $\sigma^2\left(\mathbf{x}\right) > \underline{\sigma}^2$. If there exists $p \in \mathcal{C}^2$ such that*

$$\begin{cases} 0 = \nabla \cdot \left(\nabla\left(\sigma^2\left(\mathbf{x}\right)p\left(\mathbf{x}\right)\right) - \mathbf{b}\left(\mathbf{x}\right)p\left(\mathbf{x}\right)\right) & \mathbf{x} \in \Omega, \\ 0 = \left\langle \nabla\left(\sigma^2\left(\mathbf{x}\right)p\left(\mathbf{x}\right)\right) - \mathbf{b}\left(\mathbf{x}\right)p\left(\mathbf{x}\right), \mathbf{n}\left(\mathbf{x}\right)\right\rangle & \mathbf{x} \in \partial\Omega, \end{cases} \tag{48}$$

*and $\int_{\overline{\Omega}} p\left(\mathbf{x}\right)d\mathbf{x} = 1$, then $p$ is a stationary distribution of*

$$d\mathbf{x}_t = \mathbf{b}\left(\mathbf{x}_t\right)dt + \sqrt{2\sigma^2\left(\mathbf{x}_t\right)}d\mathbf{w}_t + d\mathbf{r}_t \tag{49}$$

*in $\Omega$.*

*Remark* H.6. (48) is exactly the stationarity condition derived from the Fokker-Planck equation with Neumann boundary conditions ensuring conservation of mass.

*Proof.* Under the assumptions we see that the conditions of Corollary H.4 are satisfied, and we can use (41)-(43) to find stationary distributions of (49). First, notice that (41) simplifies to

$$\begin{aligned} 0 &= \frac{1}{2}\sum_{i,j=1}^d \frac{\partial^2}{\partial x_i \partial x_j}\left(a_{ij}\left(\mathbf{x}\right)p\left(\mathbf{x}\right)\right) - \sum_{i=1}^d \frac{\partial}{\partial x_i}\left(b_i\left(\mathbf{x}\right)p\left(\mathbf{x}\right)\right) \\ &= \frac{1}{2}\sum_{i=1}^d \frac{\partial^2}{\partial x_i^2}\left(2\sigma^2\left(\mathbf{x}\right)p\left(\mathbf{x}\right)\right) - \sum_{i=1}^d \frac{\partial}{\partial x_i}\left(b_i\left(\mathbf{x}\right)p\left(\mathbf{x}\right)\right) \\ &= \nabla \cdot \left(\nabla\left(\sigma^2\left(\mathbf{x}\right)p\left(\mathbf{x}\right)\right) - \mathbf{b}\left(\mathbf{x}\right)p\left(\mathbf{x}\right)\right). \end{aligned}$$

Next, we can considerably simplify the boundary conditions. First, notice that $\mathcal{S}$ consists of the interior of the domain's faces so for $\mathbf{x} \in \partial\Omega \cap \mathcal{S}$, the set of active boundary regions $\mathcal{I}(\mathbf{x})$ is a singleton $\mathcal{I}(\mathbf{x}) = \{(i, s)\}$, for some $i = 1, \ldots, d$ and $s \in \{m, M\}$. We focus on the lower boundaries $(m)$, as the conditions for the upper boundaries are symmetric.

For $i = 1, \ldots, d$ and $\mathbf{x} \in \partial\Omega \cap \mathcal{S}$, $\boldsymbol{\eta}_m^i(\mathbf{x}) = \mathbf{n}_m^i(\mathbf{x}) = \mathbf{e}_i$ so

$$\begin{aligned}
\mathbf{q}_m^i(\mathbf{x}) &= \mathbf{n}_m^i(\mathbf{x})^\top \mathbf{A}(\mathbf{x}) \mathbf{n}_m^i(\mathbf{x}) \boldsymbol{\eta}_m^i(\mathbf{x}) - \mathbf{A}(\mathbf{x}) \mathbf{n}_m^i(\mathbf{x}) \\
&= \sigma^2(\mathbf{x}) \left(\mathbf{e}_i^\top \mathbf{I}_d \mathbf{e}_i\right) \mathbf{e}_i - \sigma^2(\mathbf{x}) \mathbf{I}_d \mathbf{e}_i \\
&= \mathbf{0},
\end{aligned}$$

so (43) is satisfied. In addition,

$$K_m^i(\mathbf{x}) = \nabla \cdot \mathbf{a}_i(\mathbf{x}) = \frac{\partial}{\partial x_i} \sigma^2(\mathbf{x}),$$

so (42) becomes, for all $i = 1, \ldots, d$,

$$0 = -2p(\mathbf{x}) \langle \mathbf{n}_m^i(\mathbf{x}), \mathbf{b}(\mathbf{x}) \rangle + \mathbf{n}_m^i(\mathbf{x})^\top \mathbf{A}(\mathbf{x}) \nabla p(\mathbf{x}) - \nabla \cdot \left(p(\mathbf{x}) \mathbf{q}_m^i(\mathbf{x})\right) + p(\mathbf{x}) K_m^i(\mathbf{x})$$

$$0 = -2p(\mathbf{x}) b_i(\mathbf{x}) + \mathbf{a}_i^\top \nabla p(\mathbf{x}) + p(\mathbf{x}) \frac{\partial}{\partial x_i} \sigma^2(\mathbf{x})$$

$$0 = -2p(\mathbf{x}) b_i(\mathbf{x}) + \sigma^2(\mathbf{x}) \frac{\partial}{\partial x_i} p(\mathbf{x}) + p(\mathbf{x}) \frac{\partial}{\partial x_i} \sigma^2(\mathbf{x}),$$

which is

$$\begin{aligned}
0 &= -p(\mathbf{x}) b_i(\mathbf{x}) + \frac{1}{2} \frac{\partial}{\partial x_i} \left(p(\mathbf{x}) \sigma^2(\mathbf{x})\right) \\
&= \left\langle \nabla \left(\sigma^2(\mathbf{x}) p(\mathbf{x})\right) - \mathbf{b}(\mathbf{x}) p(\mathbf{x}), \mathbf{n}(\mathbf{x}) \right\rangle.
\end{aligned}$$

$\square$

### H.2.1 Reflected Langevin dynamics in a box

In this section, we derive some useful properties of the SDER

$$d\mathbf{x}_t = -\nabla L(\mathbf{x}_t) + \sqrt{2\beta^{-1} \sigma^2(\mathbf{x}_t)} d\mathbf{w}_t + d\mathbf{r}_t, \tag{50}$$

in a box domain as defined in (45)-(47), where $L \geq 0$ is some (loss/potential) function, and $\beta > 0$ is an inverse temperature parameter. First, we characterize the stationary distribution of this process.

**Recall Lemma D.1.** If $L, \sigma^2 \in \mathcal{C}^2$, $\sigma^2(\cdot) > 0$ is uniformly bounded away from 0 in $\overline{\Omega}$,

$$Z = \int_{\overline{\Omega}} \frac{1}{\sigma^2(\mathbf{x})} \exp\left(-\beta \int \frac{\nabla L(\mathbf{x})}{\sigma^2(\mathbf{x})} d\mathbf{x}\right) < \infty,$$

the integrals exist, and the field $\nabla L / \sigma^2$ is conservative (curl-free), then

$$p_\infty(\mathbf{x}) = \frac{1}{Z} \frac{1}{\sigma^2(\mathbf{x})} \exp\left(-\beta \int \frac{\nabla L(\mathbf{x})}{\sigma^2(\mathbf{x})} d\mathbf{x}\right) \tag{51}$$

is a stationary distribution of (50).

*Proof.* The drift term in this setting is $\mathbf{b} = -\beta \nabla L$. Therefore, from Lemma H.5, we get that any distribution that satisfies

$$0 = \nabla \left(\sigma^2(\mathbf{x}) p_\infty(\mathbf{x})\right) + \beta p_\infty(\mathbf{x}) \nabla L(\mathbf{x})$$

on $\overline{\Omega}$, is a stationary distribution. We can solve this PDE as

$$\begin{aligned}
0 &= \beta \nabla L(\mathbf{x}) p_\infty(\mathbf{x}) + p_\infty(\mathbf{x}) \nabla \sigma^2(\mathbf{x}) + \sigma^2(\mathbf{x}) \nabla p_\infty(\mathbf{x}) \\
&= p_\infty(\mathbf{x}) \left(\beta \nabla L(\mathbf{x}) + \nabla \sigma^2(\mathbf{x})\right) + \sigma^2(\mathbf{x}) \nabla p_\infty(\mathbf{x})
\end{aligned}$$

$$-\sigma^2(\mathbf{x}) \nabla p_\infty(\mathbf{x}) = p_\infty(\mathbf{x})\left(\beta\nabla L(\mathbf{x}) + \nabla\sigma^2(\mathbf{x})\right)$$

$$\frac{\nabla p_\infty}{p_\infty} = -\frac{\beta\nabla L + \nabla\sigma^2}{\sigma^2}$$

$$\nabla\ln p_\infty = -\frac{\beta\nabla L}{\sigma^2} - \nabla\ln\sigma^2$$

$$\nabla\ln\left(p_\infty\cdot\sigma^2\right) = -\frac{\beta\nabla L}{\sigma^2}.$$

Then,

$$\ln\left(p_\infty\cdot\sigma^2\right) = -\beta\int\frac{\nabla L}{\sigma^2} + C$$

where we used the assumption that the integral on the RHS exists, and is well defined. Hence

$$p_\infty(\mathbf{x}) \propto \frac{1}{\sigma^2(\mathbf{x})}\exp\left(-\beta\int\frac{\nabla L(\mathbf{x})}{\sigma^2(\mathbf{x})}d\mathbf{x}\right).$$

$\square$

When the integral in (51) is solvable, we can find an explicit expression for the stationary distribution, as was done in Appendix D.1.

