# OpenReview forum: "Temperature is All You Need for Generalization in Langevin Dynamics and other Markov Processes"
_NeurIPS.cc/2025/Conference — NeurIPS 2025 spotlight_

### Official Review · Reviewer_hJur · 2025-06-26

**Clarity:** 3
**Significance:** 3
**Originality:** 3
**Rating:** 5
**Confidence:** 4

**Summary:**

This paper presents new generalization bounds for learning algorithms that can be written as time-homogeneous Markov processes. The proposed methods work for both discrete and continuous time processes, as soon as there exists a stationary distribution that is a Gibbs density with respect to some prior (data-independent) measure, with a potentially data-dependent potential.

The main technical novelty is to note that the normalizing constant of the Gibbs potential is cancelled out when computing some kind of weighted symmeterized KL divergence. The authors then used that the KL divergence $KL(p_t|P_infty)$ between the marginal $p_t$ at time $t$ and a stationary distribution $p_\infty$ is non-increasing to derive a bound of the form $KL(p_t|\nu) \leq KL(p_0 | \nu) + E_{p_0}[\Psi]$, where $\nu$ is a prior distribution and $\Psi$ is the Gibbs potential (a disintegrated version is also proposed). This result can be used in standard PAC-Bayesian bounds to obtain generalization bounds in several settings.

The main application is the case of (regularized) continuous Langevin dynamics, where the Gibbs potential can be computed exactly. This leads to a trajectory-independent bound, involving only the expected surrogate loss at initialization, which is in contrast with most of the literature. The authors then discuss the implications of their bound and compare it to existing work. In addition to being time-uniform, a main advantage of the bound (according to the authors) is to not depend on the trajectory of the algorithm, hence, being computable before-hand.

**Questions:**

- It is mentioned in Section 4.1 that one of the main advantages of the new bounds is to be trajectory-independent. Why is that necessarily a good thing? One could argue that a trajectory-independent bound does not give any insight as to why generalization happens and it is common in the literature that generalization should be affected by the properties of the local minimum the algorithm converged to (ie, flat minima,...).
- Can you empirically study the correlation between the expected loss at initialization and the generalization error. Also, how was the expectation appearing in the bound evaluated in your experiments?
- Can you obtain closed-form bounds beyond CLD? In particular can you apply it to the discrete-time case (ie, SGLD, only SGD is discussed line 342), to the best of my knowledge, it is hard to fully characterize the Gibbs potential in that case.
- Can you obtain a meaningful bound for CLD without regularization but in the settings of [Li et al., Futami et al.]?
- line 210: why is $\beta = O(N)$ sufficient for non-vacuous bounds? When $\beta = N$, the bound does not go to $0$ as $N\to\infty$.

**Ethical Concerns:**

["NO or VERY MINOR ethics concerns only"]

**Final Justification:**

After the rebuttal, I still have a positive opinion on the paper and I believe it brings a very interesting perspective comparer to the existing bounds.

Most of my questions were addressed,  in particular by new experiments for the comparison with existing works.

I am happy to keep my acceptance score.

**Limitations:**

Most of the limitations are discussed in Section 5.

As discussed above, some implications of the bounds and comparisons with existing works could be made more precise.

**Quality:**

3

**Strengths And Weaknesses:**

**Strengths:**

- The proposed generalization bounds apply to a quite general class of Markov process

- For CLD, the new generalization bounds are time-independent, which, under mild conditions, ensures that the bound is non-vacuous as $t \to \infty$.

- The proposed proof technique is ingenious and novel, despite its simplicity.


**Weaknesses:**

- The treatment of absolute continuity conditions is not always clear. It seems to me that the arguments of Section 2 if all the involved probability measures are absolutely continuous wrt the Lebesgue measure (or another reference measure). In particular, the condition $KL(p_0 | p_\infty) < \infty$ might be required for the derivations of Equation (1), which prevents to initialize from Dirac distributions.
- The proposed bounds for require either regularization or bounded domain, which is a stronger requirement compared to [Li et al.] and [Futami et al.].
- It is not clear whether the new technique can be used to obtain bounds beyond the case of CLD.
- It is great that the bound is time-uniform, but if we take $t\to\infty$ in the first inequality of Equation (1), we obtain $0 \leq KL(p_0 | p_\infty)$, which seems loose. This makes one wonder how tight the bound is in practice. In particular, it is not clear that the proposed bounds are better than the existing gradient-based bounds.
- The comparison with gradient-based bounds is very informal (only an implicit dimension-dependence is mentioned), it should be made more precise. Indeed, the technique of [Mou et al.] consists in a direct bound of $KL(p_t | \nu) - KL(p_0 | \nu)$, like in your method, and it is not clear which approach is tighter. I believe this should be formally studied.
- The bound for CLD is quite surprising as it only depends on the initialization, while it is common wisdom that CLD should forget the initialization when $t\to\infty$. This might suggest that the bound is sub-optimal as $t\to\infty$. An important question is whether the loss at initialization indeed as an empirical impact on the generalization at time $t$, such an experiment (ie, analysis of the correlation between loss at initialization and generalization error) could validate the approach but is lacking.
- In section 4.1, you mention that stability-based bounds grow with time, but it is not the case of [Li et al., Futami et al.]. Also [Dupuis et al.] is mentioned but is not stability-based. The literature review in Section 4 could be made more precise.

**Other remarks and typos**

- You use a bounded loss $f$, why not assume subgaussian loss? Such subgaussian PAC-Bayes bounds have been derived used in similar setting, for instance in [Generalization bounds for heavy-tailed SDEs through the fractional Fokker-Planck equation, Dupuis et al., 2024], it could be an easy improvement of the theory.
- Typos: line 122 (grantees),
- the writing feels a bit informal in places, eg, line 63 (our two divergences), line 92 (very data-dependent), line 93 (some quantity)
- last line of claim 2.3, why is it an inequality?

---

> ### Author Rebuttal · Authors · 2025-07-31
>
> **We thank the reviewer for acknowledging the generality and novelty of our results, and for the detailed and helpful feedback. Below we address the reviewer’s comments (which we will clarify in the revised version of the paper), all typos and formulations will be corrected.**
>
> **Weaknesses:**
>
> 1. **The treatment of absolute continuity conditions is not always clear...**
>
>     You are correct to address absolute continuity conditions. We incorporate them implicitly through the divergence terms, which are defined to be infinite when absolute continuity does not hold. In particular, note that:
>     1. Even when one of the divergence terms is $\infty$ the inequalities hold (trivially).
>     2. Absolute continuity does not imply finite KL divergence, and is therefore a weaker condition compared to finite KL divergence (assuming that $D_{KL}$ is defined to be infinite when the measures are not absolutely continuous).
>     Finally, you are correct to note that this prevents us from initializing from the Dirac distribution, but we can take any absolutely continuous approximation of this distribution, at the cost of large divergence terms.
>
> 2. **The proposed bounds require either regularization or a bounded domain...**
>
>     All of the results in Table 1 use regularization similar to the one used in Section 3 (as stated in the last two lines in the Table’s caption). The cited papers present additional results that do not require this kind of regularization. For example, Mou et al. (Proposition 1) and Li et al. (Theorem 9) used stability based reasoning to derive bounds that do not require this sort of regularization, but instead grow with time — making an implicit use of early stopping as a regularization mechanism. Futami & Fujisawa (Theorem 5) present a non-regularized version of the result for convex losses, but require regularization in order to extend it to non-convex losses. Similarly, as noted in Section 4.2, Raginsky et al. use a dissipativity assumption instead of our explicit regularization term.
>
> 3. **It is not clear whether the new technique can be used to obtain bounds beyond the case of CLD.**
>
>     You are correct to observe that one bottleneck of the results in this paper is the ability to derive explicit forms for stationary distributions of typical training procedures, which we discuss in Section 5. In general, finding exact forms for stationary distributions in “natural” scenarios may be difficult, yet there are still ways to find meaningful results in these cases:
>     1. Approximation by known distributions — e.g. Raginsky et al. used CLD to approximate the discrete SGLD training scheme.
>     2. Recent work implicitly characterized the distribution of more realistic training processes, e.g. [A].
>     3. By modifying common training procedures, e.g. via a Metropolis-Hasting type update rule, the exact stationary distribution can be found, even when each training step deviates from CLD significantly.
>
>     [A] Azizian, Waïss, et al. "What is the long-run distribution of stochastic gradient descent? A large deviations analysis." arXiv preprint arXiv:2406.09241 (2024).
>
> 4. **It is great that the bound is time-uniform, but if we take $t \to \infty$ ...**
>
>     These are two good points:
>     1. This bound may indeed be loose (e.g. see Tables 2-4, where the bound is larger than the observed gap). In fact, taking $t \to \infty$ allows us to take into account additional properties of the system, such as the generalization of the asymptotic distribution (which can be much better,  e.g. see [8, B] characterizing the generalization capabilities of posterior sampling in NNs). Deriving exact bounds for intermediate times involves considering the particular training dynamics, which we were able to avoid by simply using the monotonicity implied by the second law.
>     2. We find that this paper complements gradient-based bounds, as they shed light on different aspects of generalization.
>
>     [B] Dziugaite, Gintare Karolina, and Daniel M. Roy. "The Size of Teachers as a Measure of Data Complexity: PAC-Bayes Excess Risk Bounds and Scaling Laws." The 28th International Conference on Artificial Intelligence and Statistics. 2025.
>
> 5. **The comparison with gradient-based bounds is very informal...**
>
>     1. The dependence on dimensionality is indeed only referenced implicitly, as it is hard to be characterized specifically without further assumptions.
>
>         For example, in linear regression with data $(x_i, y_i)_{i=1}^N$, $x_i \sim \mathcal{N} (0, \sigma^2 I_d)$ i.i.d, labels $y_i = x_i^\top \theta^\star$ and parameters $\theta \in \mathbb{R}^d$ we have the following:
>
>         $$
>         \nabla L_S (\theta) = \frac{1}{N} \sum_{i=1}^N x_i x_i^\top (\theta - \theta^\star)
>         $$
>
>         so $|| \nabla L_S (\theta) ||_2$ typically scales as $\sigma^2 ||\theta - \theta^\star ||_2 \sqrt{1 + d/N}$. That is, the dependence on dimensionality depends not only on the model but also on the data, stage of the optimization, etc.
>
>     2. The dependence of [Mou et al.]’s bounds on trajectory statistics has the potential of achieving tighter bounds than our trajectory agnostic ones. That being said, due to this fact they are more complicated, and deteriorate significantly when simplified, e.g. by a factor of a Lipschitz constant (see (29) in [Mou et al.]).
>
> 6. **The bound for CLD is quite surprising...**
>
>     Correct! Markov processes typically forget their initialization, but due to the second law of thermodynamics (see Claim 2.4) they can only get closer to a stationary distribution. Thus the bound connects both ends — it involves **the expected/supremum value at $t=0$ of the potential at $t \to \infty$**. That being said, we agree that as $t \to \infty$ more information can be incorporated, thus improving the bound (see our previous response to (W4)).
>     Finally, from a practical point of view, the potential’s value at initialization may already be small, e.g. when finetuning a pretrained model with zero-shot capabilities.
>
> 7. **In section 4.1, you mention that stability-based bounds grow with time...**
>
>     Regarding [Dupuis et al.], we do not claim that their bounds are stability based, rather we note that due to the different quantity being bounded, their results are qualitatively different than the ones in this paper. Specifically, they derive bounds for all times simultaneously, while we derive bounds for all times $t \ge 0$ individually.
>     We will make this reference clearer.
>
>     In their paper, [Li et al.] derive multiple bounds — some are stability based and do grow with time (e.g. their Theorem 9), while others are not and are bounded as a function of $T$ (e.g. their Theorem 15, the one used in this paper’s Table 1).
>
>     As for [Futami & Fujisawa], we indeed do not claim that their bound deteriorates with time, but rather that it depends on gradient statistics, implicitly dependent on the dimension through the norms of the gradients, and has an exponential dependence on an upper bound of the loss.
>
> **Other remarks and typos:**
>
> 1. **You use a bounded loss $f$, why not assume subgaussian loss...**
>
>     We opted to use a bounded $f$ for simplicity, but we can indeed use a sub-Gaussian $f$ instead, similar to many previous results.
>
> 4. **Last line of claim 2.3, why is it an inequality?**
>
>     Correct, this is in fact an equality. We will change it in the next version.
>
>
> **Questions:**
>
> 1. **It is mentioned in Section 4.1 that one of the main advantages of the new bounds is to be trajectory-independent. Why is that necessarily a good thing...**
>
>     Practically, with a trajectory-independent (sufficiently tight) bound we can quickly evaluate different neural network architectures, initializations, and pre-training regimes before doing optimization. Specifically, it was already observed in practice in pre-trained foundation models, where low zero-shot loss on a task typically correlates well with the generalization performance on the task after fine-tuning.
>     Theoretically, the trajectory independence of the bounds in this work complements trajectory based bounds as it shows that:
>     1. Trajectory statistics are not necessary to derive meaningful (non-vacous) generalization bounds.
>     2. It is not necessary to rely on local properties (such as Lipschitz continuity/smoothness), and coarse statistics of the global geometry can imply non-trivial generalization.
>
> 2. **Can you empirically study the correlation between the expected loss at initialization and the generalization error. Also, how was the expectation appearing in the bound evaluated in your experiments?**
>
>     This is an interesting point which we intend to study further, e.g. to see what the implications of pretraining are for the bounds.
>     Regarding the evaluation of the expectations appearing in the bound, we approximated them with the appropriate empirical means, based on 3 samples for SVHN, and 5 for the rest.
>
> 3. **Can you obtain closed-form bounds beyond CLD? In particular, can you apply it to the discrete-time case (ie, SGLD, only SGD is discussed line 342), to the best of my knowledge, it is hard to fully characterize the Gibbs potential in that case.**
>
>     See response (W1) by reviewer 6xrj.
>
> 4. **Can you obtain a meaningful bound for CLD without regularization but in the settings of [Li et al., Futami et al.]?**
>
>     See response to (W2).
>
> 5. **Line 210: why is $\beta = O(N)$ sufficient for non-vacuous bounds? When $\beta = N$, the bound does not go to $0$ as $N \to \infty$.**
>
>     By non-vacuous we mean that it can be smaller than some threshold of trivial generalization, e.g. 1/2 in balanced binary classification. In order for the bound to vanish as $N \to \infty$ we need $\beta = o(N)$.

---

> > ### Comment · Reviewer_hJur · 2025-08-01
> > **Thank you**
> >
> > Thank you very much for the quality of your work and the detailed answers to my questions, which clarified several points.
> >
> > I just have two small questions about your answer.
> >
> >  - Weakness 5. Does your answer mean that the bounds of [Mou et al] are tighter in practice? Is it something that you can see on numerical examples? Also is it possible to characterize at what time $t$ one bound becomes better than the other? (note that I agree with your answers about your bounds being of real interest, but I still believe the comparison is interesting)
> >
> >  - Q2. Studying pretraining is indeed interesting. However, my question was more related to testing your bound. Your bound predicts that the empirical risk at initialization is related to the generalization error at time $t$, do we observe an empirical correlation between these two quantities?

---

> > > ### Author Response · Authors · 2025-08-08
> > >
> > > 1. **Weakness 5. Does your answer mean that...**
> > >
> > >     We did not test this before, mainly since [Mou et al.]’s bounds depend on various constants that there is not a clear way to easily evaluate in practice. For example, in proposition 3, which is most relevant (e.g., it is in continuous time), $M$ is such a constant. However, following the reviewer's suggestion, we tested [Mou et al.]'s bound, and removed the term containing $M$ (which makes the bound more favourable for [Mou et al.]). Specifically, we conducted a set of numerical experiments on the binary classification MNIST dataset to evaluate its tightness compared to our bound (see tables below). We found that the relative tightness of the two bounds varies with different values of $\beta$ and different times. For example,
> > >
> > > * After 20 epochs we get:
> > >
> > > |beta|Train Error|Test Error|Generalization Gap|Mou et al error bound|our bound|
> > > |---|---|---|---|---|---|
> > > |0.03N=1800|0.1224|0.137|0.0146|0.0539|0.114|
> > > |0.15N=9000|0.0515|0.0747|0.0232|0.1279|0.2546|
> > > |0.4N=24000|0.0335|0.058|0.0245|0.2845|0.4156|
> > > |0.7N=42000|0.0278|0.0498|0.022|0.493|0.5498|
> > > |N=60000|0.0249|0.0428|0.0179|0.7032|0.6571|
> > > |2N=120000|0.0209|0.0356|0.0147|1.4044|0.9293|
> > >
> > > * After 50 epochs we get
> > >
> > > |beta| Train Error | Test Error | Generalization Gap | Mou et al error bound | our bound |
> > > |------------------|-------------|------------|---------------------|------------------------|-----------|
> > > | 0.03N = 1800    | 0.1156      | 0.1697     | 0.0541              | 0.0637                 | 0.114     |
> > > | 0.15N = 9000    | 0.0491      | 0.0615     | 0.0124              | 0.1324                 | 0.2546    |
> > > | 0.4N = 24000    | 0.0295      | 0.0348     | 0.0053              | 0.2992                 | 0.4156    |
> > > | 0.7N = 42000    | 0.0217      | 0.0283     | 0.0066              | 0.4903                 | 0.5498    |
> > > | N = 60000        | 0.0173      | 0.0277     | 0.0104              | 0.6827                 | 0.6571    |
> > > | 2N = 120000     | 0.0108      | 0.0265     | 0.0157              | 1.3153                 | 0.9293    |
> > >
> > > * After 250 epochs we get:
> > >
> > > | beta              | Train Error | Test Error | Generalization Gap | Mou et al error bound | our bound |
> > > |------------------|-------------|------------|---------------------|------------------------|-----------|
> > > | 0.03N = 1800    | 0.122      | 0.1049     | -0.0171             | 0.1273                  | 0.114     |
> > > | 0.15N = 9000    | 0.0502      | 0.0476     | -0.0026             | 0.1503                 | 0.2546    |
> > > | 0.4N = 24000    | 0.0284      | 0.0296     | 0.0011              | 0.2853                 | 0.4156    |
> > > | 0.7N = 42000    | 0.0178      | 0.0247     | 0.0069              | 0.4595                 | 0.5498    |
> > > | N = 60000        | 0.0127      | 0.024     | 0.0113               | 0.6478                 | 0.6571    |
> > > | 2N = 120000     | 0.005       | 0.0234     | 0.0184              | 1.2158                 | 0.9293    |
> > >
> > > * And after 400 epochs we get:
> > >
> > > | beta              | Train Error | Test Error | Generalization Gap | Mou et al error bound | our bound |
> > > |------------------|-------------|------------|---------------------|------------------------|-----------|
> > > | 0.03N = 1800    | 0.1224      | 0.1105     | -0.0119             | 0.19                   | 0.114     |
> > > | 0.15N = 9000    | 0.0499      | 0.0556     | 0.0057              | 0.1774                 | 0.2546    |
> > > | 0.4N = 24000    | 0.0261      | 0.0357     | 0.0096              | 0.3005                 | 0.4156    |
> > > | 0.7N = 42000    | 0.0161      | 0.0271     | 0.011               | 0.4548                 | 0.5498    |
> > > | N = 60000        | 0.0112      | 0.0255     | 0.0143              | 0.6247                 | 0.6571    |
> > > | 2N = 120000     | 0.0038      | 0.0249     | 0.0211              | 1.1455                 | 0.9293    |
> > >
> > > Therefore, in some cases [Mou et al.] is better, while in other cases our bound is better, and we could not draw any interesting conclusions.
> > >
> > > Finally, just to make sure everyone is on the same page, we remind again that our bound has the advantage that it can be evaluated at the initialization, while [Mou et al.]'s bound would have been much worse, if we would have tried to estimate it at initialization using worst-case bounds on the gradients.
> > >
> > > 2. **Studying pretraining is indeed interesting. However...**
> > >
> > >     Following the reviewer's suggestion, we conducted empirical tests on binary classification on MNIST, to examine whether the generalization error correlates with the empirical risk at initialization and found no significant correlations. We tried different initialization variances, but the model achieved similar generalization errors for all the initializations in which the model was trainable (i.e. the empirical risk decreases below chance level). As we mentioned in our earlier response, we believe that we should see a positive correlation if we use pre-trained initializations, and we will examine this more closely.

---

> > > > ### Comment · Reviewer_hJur · 2025-08-09
> > > > **Thank you**
> > > >
> > > > Thank you very much for your answer and for the additional experiments. I believe these experiments are interesting and could be discussed in the final version of the paper, it sheds light on   the comparison with existing works.
> > > >
> > > > I am satisfied by the rebuttal and I am happy to keep my acceptance score.
> > > >
> > > > Good luck!

---

### Official Review · Reviewer_iEYq · 2025-07-01

**Clarity:** 3
**Significance:** 3
**Originality:** 3
**Rating:** 5
**Confidence:** 3

**Summary:**

The manuscript derives bounds for the difference of the loss function for the data distribution and an empirical approximation. These bounds are stated in terms of the KL divergence and the Renyi infinite divergence. These bounds are based on a data independent reference measure $\nu$ such that the stationary distribution of the process, which describes the learning process, is a Gibbs distribution with respect to $\nu$. The authors then use some KL properties and PAC Bayes results to derive their bounds. These bounds are first derived for quite general time homogeneous Markov processes. Then they are applied to Markov processes that are solutions of SDEs, which resemble stochastic gradient Langevin dynamics for training neural networks. For  certain choices of learning procedures (e.g. parameters in a box or weight decay) the base distribution and the Gibbs potential of the stationary distribution are explicitely given, which makes for concrete bounds.These SGLD have a "temperature" hyperparameter. The role of this temperature is discussed, which also explains the name of the paper.

**Questions:**

If the following could be clarified I would happily increase my rating:

* The loss function has to be positive? Why do you restrict to the case with values in [0,1]? Is this only for simplicity or is there serious issues if you don't? Popular loss functions do not have that property.
* Theorem B.1. Please be more precise, what is a random variable, what a realization? Also its easier to understand when you make $\hat{\rho}$ more explicit e.g. by writing $\hat{\rho}\in\mathcal{P}(\mathcal{H})$.
* Then notation in Theorem B.2 is not entirely clear to me. $\hat{\rho}_S$ is a Markov kernel in $S$? and $h|S\sim \hat{\rho}_S$ just means $h\sim \hat{\rho}(S,dx)$?
* Is the setting in Theorem B.1,B.2 not a bit different from the setting in Theorem 2.7? In 2.7 the conditional expectation is used, so both sides are random variables, while this is not the case in B.1/2 if I'm not mistaken.
* how do you optimize over $\lambda$ in line 457?

Minor points:
* what does it  mean that the same generalization results apply in line 165? It just changes the $f$ then?
* please give references for the interpretation of network training by equations (9) and (10)
* I guess you use in Corollary 3.1 that in this case $\Psi_S=\beta L_S$. This could be briefly mentioned in line 190
* at the beginning the terminology for $p_t$ is "marginal distribution", but in line 84 it is called "state distribution". This could be unified.
* please give the exact statement number in the references for claim 2.4.

**Ethical Concerns:**

["NO or VERY MINOR ethics concerns only"]

**Final Justification:**

The paper presents interesting theory, is clearly written and the authors were very precise in answering my questions

**Limitations:**

yes

**Quality:**

3

**Strengths And Weaknesses:**

Strengths: The paper is generally well written and the results are interesting. Bounds are presented in various levels of generalities which makes their use on related problems as well as in concrete applications easier. The authors openly discuss current limitations and directions of future work in section 5.

Weaknesses: At certain places the notation could be a bit more precise, see Questions. Although the bounds on the difference of the errors are interesting, a discussion on the generalization on the task of the neural network would be interesting.

---

> ### Author Rebuttal · Authors · 2025-07-30
>
> **We thank the reviewer for acknowledging that our results are interesting, and that the paper is well written, and for feedback that improved our paper. Below we address the reviewer’s comments.**
>
>
> **Weaknesses:**
>
> 1. At certain places the notation could be a bit more precise, **see Questions.**
> 2. **Although the bounds on the difference of the errors are interesting, a discussion on the generalization on the task of the neural network would be interesting.**
>
>     We agree that the performance on the task of the model would be interesting. In general, it can be bounded by the empirical error + a bound on the generalization gap. Thus, the current work can be applied to bound the generalization error when either 1) there are empirical observations of the training error, or 2) when there is an analytical prediction for the training error, e.g. as was derived in Appendix E.
>     We leave the theoretical analysis of the empirical error for future work.
>
>
> **Questions:**
>
> 1. **The loss function has to be positive? Why do you restrict to the case with values in $[0,1]$? Is this only for simplicity or is there serious issues if you don't? Popular loss functions do not have that property.**
>
>     There are 2 functions in play —
>     1. The “error function” $f$ used to defined $E_S$ and $E_D$ which is the “test objective”, e.g. classification error (0-1 loss), and is required to be bounded in $[0, 1]$, but has no additional constraints, e.g. it need not be continuous. The boundedness condition comes from the PAC-Bayes bounds that form the basis of our analysis, namely Theorems B.1 and B.2. In the general case, we can consider $f$ which is bounded in $[0, c]$ for some $c > 0$ at the cost of a multiplicative $c$ factor to the generalization bounds, or alternatively assume that $f$ is sub-Gaussian. For simplicity we only used $[0, 1]$.
>     2. The “optimization objective” $L_S$, e.g. cross-entropy-loss, which is assumed to be non-negative and $C^2$ for the existence of a stationary distribution and the simplicity of the analysis. While for the results that depend on the KL divergence $L_S$ need not be bounded, the results that depend on $D_\infty$ do require an essentially bounded potential, and thus a bounded $L_S$ in Corollary 3.1.2.
>
> 2. **Theorem B.1. Please be more precise, what is a random variable, what a realization? Also its easier to understand when you make $\hat{\rho}$ more explicit e.g. by writing $\hat{\rho} \in \mathcal{P} (\mathcal{H})$.**
>
>     The probability is taken over the training set $S$, and the event in question is that the expectation over $h$ of the generalization gap is bounded. We will clarify this in the paper.
>
> 3. **Then notation in Theorem B.2 is not entirely clear to me. $\hat{\rho}_S$ is a Markov kernel in $S$? And $h \mid S \sim \hat{\rho}_S$ just means $h \sim \hat{\rho} (S, dx)$?**
>
>     Not quite, Theorems B.1 and B.2 apply to general distributions, not necessarily ones that correspond to a Markov process. That is, $S$ denotes the training data, and $\hat{\rho}_S$ is any data-dependent distribution. We later use this result with a Markov process with kernel $p(dx, y; S)$, i.e. given a training set $S$, $p$ is a transition kernel defining the transition probability from a state $y$.
>     We will clarify this in the paper.
>
> 4. **Is the setting in Theorem B.1,B.2 not a bit different from the setting in Theorem 2.7? In 2.7 the conditional expectation is used, so both sides are random variables, while this is not the case in B.1/2 if I'm not mistaken.**
>
>     While you are correct to question the notation, both cases are actually the same:
>     1. Theorems B.1 and 2.7.1 consider the case where the outer probability is taken over the training set $S$, while the event under consideration is that the expectation, taken over $h$ alone, of the generalization gap is bounded, given $S$. I.e. no randomization over $S$ in the expectation, and therefore the notation $\mathbb{E} [\cdot | S]$ can be used in Theorem B.1 as well. We will unify the notation in the next version of the paper.
>     2. Theorems B.2 and 2.7.2 consider the case where the outer probability is over both $S$ and $h$.
>
> 5. **How do you optimize over $\lambda$ in line 457?**
>
>     The derivation is quite simple and we shall include it in the paper. Starting from the expression in Theorem 2.7 of [1], we uniformly bound the log-density term with $D_\infty$, making the bound of the generalization gap $R(\tilde{\theta}) - r(\tilde{\theta})$ $\theta$-independent. Thus, we can optimize over $\lambda$ to get a $D_\infty$-dependent bound over the generalization gap.
>
>     [1] Alquier, Pierre. "User-friendly introduction to PAC-Bayes bounds." arXiv preprint arXiv:2110.11216 (2021).
>
>
> **Minor points:**
>
> 1. **What does it mean that the same generalization results apply in line 165? It just changes the $f$ then?**
>
>     Correct. It means that the entire analysis in Section 2 could have been made in parameter space, instead of hypothesis space, by changing $f$ and $\Psi_S$ to be in terms of $\theta$ instead of $h$.
>
> 2. **Please give references for the interpretation of network training by equations (9) and (10).**
>
>     If the point is with regards to the claims “such … is quite common in practical scenarios such as NN training”:
>     1. Parameter clipping is related to standard norm based regularization (see e.g. [A] Section 7.2).  In addition, some Quantization-Aware Training (QAT) algorithms include parameter clipping after each update to allow for meaningful quantization [B,C,D,E].
>    2. Norm-based regularization is ubiquitous in deep learning, e.g. see [A] for a starting point.
>
>     If the point is with regards to the use of SDEs to model NN training, see the previous work discussed in Section 4.1.
>
>     [A] Goodfellow, I., Bengio, Y., and Courville, A. Deep Learning. MIT Press, 2016.
>
>     [B] Courbariaux, M., Bengio, Y., and David, J.-P. Binaryconnect: Training deep neural networks with binary weights during propagations. Advances in neural information processing systems, 28, 2015.
>
>     [C] Hubara, I., Courbariaux, M., Soudry, D., El-Yaniv, R., and Bengio, Y. Binarized neural networks. Advances in Neural Information Processing Systems, 29, 2016.
>
>     [D] Baskin, C., Zheltonozhkii, E., Rozen, T., Liss, N., Chai, Y., Schwartz, E., Giryes, R., Bronstein, A. M., and Mendelson, A. Nice: Noise injection and clamping estimation for neural network quantization. Mathematics, 9(17), 2021. ISSN 2227-7390. doi: 10.3390/math9172144.
>
>     [E] Gholami, A., Kim, S., Dong, Z., Yao, Z., Mahoney, M. W., and Keutzer, K. A survey of quantization methods for efficient neural network inference. In Low-Power Computer Vision, pp. 291–326. Chapman and Hall/CRC, 2022.
>
> 3. **I guess you use in Corollary 3.1 that in this case $\Psi_S = \beta L_S$. This could be briefly mentioned in line 190.**
>
>     Correct. We will emphasize this in the paper.
>
> 4. **At the beginning the terminology for $p_t$ is "marginal distribution", but in line 84 it is called "state distribution". This could be unified.**
>
>     Correct. We will fix this in the next version of the paper.
>
> 5. **Please give the exact statement number in the references for claim 2.4.**
>
>     Cover [10] gives the monotonicity of the KL divergence for discrete time processes in Theorem 4, which can be simply used to derive the KL part of our Claim 2.4 (see Corollary A.4 and the lemmas leading to it). For the $D_\infty$ part we give the exact reference in line 91.
>     We will clarify this in the next version.

---

> > ### Comment · Reviewer_iEYq · 2025-08-01
> >
> > Many thanks for addressing my questions, I accordingly will raise my score.

---

### Official Review · Reviewer_8GSk · 2025-07-02

**Clarity:** 4
**Significance:** 3
**Originality:** 4
**Rating:** 5
**Confidence:** 2

**Summary:**

The paper gives a bound on the generalization gap when training with a Markovian stochastic algorithm that holds at any time during training, with no dependence on mixing time, dimensionality, or gradient norms. The authors start by giving a bound for any data-dependent Markov-process whose statinoary distribution is Gibbs and then specializes to continuous-time Langevin dynamics. The temperature $1/\beta$ and the initial expected loss turn out to be "all you need" to get a high-probability bound on the generalization gap. The authors also carefully discuss implications of the bound in the context of common neural net training practices.

**Questions:**

- In Thm 2.7, can the $\textrm{ess sup}_{h \sim p_0} \Psi_S(h)$ term hide dependence on dimensionality when the loss scales with width?
- The paper doesn't seem to discuss gradient-norm clipping, a common training practice. Could the authors discuss possible extensions to state-dependent drift?

**Ethical Concerns:**

["NO or VERY MINOR ethics concerns only"]

**Final Justification:**

I maintain my acceptance score.

**Limitations:**

Yes

**Paper Formatting Concerns:**

Typos
- L122: "grantees"
- L157 "it's potential"

**Quality:**

3

**Strengths And Weaknesses:**

### Strengths
- Thm 2.7 is significant and general; the PAC-Bayes recipe works for any data-dependent Markov process as long as the stationary distribution is Gibbs type.
- The results are original. In particular, it's nice that it does not require gradient norms, unlike the trajectory-dependent results of Mou et al. and Futami and Fujisawa et al.
- The paper is very well written and the proof idea is easy to follow. The authors motivate upfront that the everything follows from the generalized second law of thermodynamics monotonicity, so Corollary 2.5 and Thm 2.7 are easier to follow.
- The "Interpreting Corollary 3.1" subsection is helpful for anchoring the theoretical results within typical NN training practices. It is very clarifying that the required $\beta = O(N)$ is desirable in Bayesian settings as well as SGD.

### Weaknesses
- If I understand Appendix D.2 correctly, it seems that the initialization $p_0$ needs to match the shape of the loss (the regularizer or the box constraint), otherwise a KL cost depending on dimensionality will be incurred.
- The bound loses meaning if a task has unbounded or heavy-tailed losses that don't take finite expected or ess sup loss at initialization.
- Please see "Questions" below.

---

> ### Author Rebuttal · Authors · 2025-07-30
>
> **We thank the reviewer for acknowledging the significance, originality, and clarity of our work, and for the useful feedback. Below we address the reviewer’s comments; all typos will be corrected.**
>
> **Weaknesses:**
>
> 1. **If I understand Appendix D.2 correctly, it seems that the initialization $p_0$ needs to match the shape of the loss (the regularizer or the box constraint), otherwise a KL cost depending on dimensionality will be incurred.**
>
>     Correct — the bound in Theorem 2.7 depends on $KL(p_0 || \nu)$, where $\nu$ is related to the constraints/regularization. In particular, when the initialization does not match the “base measure” induced by the constraints/regularization, a KL cost depending on dimensionality is incurred. See Remark D.8 for more details, and discussion starting in line 203 (“Magnitude of regularization”) on this assumption.
>
> 2. **The bound loses meaning if a task has unbounded or heavy-tailed losses that don't take finite expected or ess sup loss at initialization.**
>
>     Correct, though this can be mitigated by clipping the loss. This clipping is standard in practice (e.g., in reinforcement learning [A, B]) and in the theory of optimization [C, D]. Moreover, this clipping can be done in a differentiable way (e.g., using either softmin, tanh with $c \cdot \tanh (L / c)$, etc.). Practically, the clipping can be done at values only slightly higher than the typical loss at the initialization: since the loss is roughly monotonically decreasing in CLD with small noise, the optimization process would then typically operate below the clipping threshold and will not be affected by it. We will clarify this in the paper.
>
>     [A] Mnih, V., Kavukcuoglu, K., Silver, D. et al. Human-level control through deep reinforcement learning. Nature 518, 529–533 (2015).
>
>     [B] Schulman, John, et al. "Proximal policy optimization algorithms." arXiv preprint arXiv:1707.06347 (2017).
>
>     [C] Levy, Kfir Y., Ali Kavis, and Volkan Cevher. "STORM+: Fully adaptive SGD with momentum for nonconvex optimization." arXiv preprint arXiv:2111.01040 (2021).
>
>     [D] Kavis, Ali, Kfir Yehuda Levy, and Volkan Cevher. "High probability bounds for a class of nonconvex algorithms with adagrad stepsize." arXiv preprint arXiv:2204.02833 (2022).
>
>
> **Questions:**
>
> 1. **In Thm 2.7, can the $\text{ess} \sup_{h \sim p_0} \Psi_S (h)$ term hide dependence on dimensionality when the loss scales with width?**
>
>     Yes, in fact, the supremum can hide an infinite value, e.g. when $p_0$ is Gaussian, the essential supremum is over the entire space, meaning that when the loss is unbounded the value is $\infty$. This can be mitigated by clipping the loss as was previously discussed.
>
> 2. **The paper doesn't seem to discuss gradient-norm clipping, a common training practice. Could the authors discuss possible extensions to state-dependent drift?**
>
>     Correct. The paper does not currently discuss gradient clipping. That being said, it lays the groundwork to do so, as by Theorem 2.7 it only remains to characterize a stationary distribution of the new process. Unfortunately, this may be highly non-trivial, as the stationary distributions of such processes rarely have a closed-form expression.  However, we do not need to derive the full stationary distribution, but only need to evaluate either the expected value or an upper bound for the potential at initialization. This may be significantly simpler in some cases.

---

> > ### Comment · Reviewer_8GSk · 2025-08-02
> >
> > Thank you for the clarifications regarding loss and gradient clipping. I maintain my acceptance score.

---

### Official Review · Reviewer_6xrj · 2025-07-03

**Clarity:** 2
**Significance:** 3
**Originality:** 3
**Rating:** 4
**Confidence:** 3

**Summary:**

This paper derives the generalization gap results for two common cases, which both have important practical implications.
1.This paper first derives the generalization bound for Markov process with a Gibbs stationary distribution, regardless of the trajectory of the process.
2.This paper further focus on the special case of the continuous-time Langevin Dynamics (CLD). The generalization bound is agnostic of dimension, number of iterations, or the trained weights, making the bounds more practical compared to previous ones.

**Questions:**

1. Bayesian Interpretation of Regularization: Regarding the loss in line 632, can the dimension-dependent regularization term ( $\lambda \propto d$ ) be interpreted from a Bayesian perspective? Does it correspond to a specific prior distribution, which might justify this unconventional choice?
2. Negative Generalization Gap: In Table 2, the generalization gap, $E_D-E_S$, is reported to be negative in some instances. Could the authors provide an explanation for this observation? Is it due to stochasticity in the evaluation on a finite test set, or does it point to another phenomenon?
3. Impact of $N$ and $\beta$ Scaling: How do changes in the sample size $N$ affect the results presented in Tables 2-4? Furthermore, how do $N$ and $\beta$ jointly influence the generalization gap $E_D-E_S$ ? To better investigate the impact of $\beta$ scaling, we suggest that the authors consider evaluating a more systematic range of values, for example, $\beta \in$ $\{0.01 N, 0.1 N, N, 10 N, 100 N, 1000 N, \infty\}$.

If the authors can explain this paper clearly in plain math without introducing advanced math jargons, anonymous link to pdf can be provided (if this is acceptable by the conference regulations). This modifications could also help this paper to have larger impacts if accepted.

**Ethical Concerns:**

["NO or VERY MINOR ethics concerns only"]

**Final Justification:**

The authors have addressed my comments. Therefore, I decided to increase the score.

**Limitations:**

See weakness.

**Quality:**

2

**Strengths And Weaknesses:**

Strengths:
This paper provides a novel perspective on the generalization bound. The optimization process of DNNs can be modeled as a Markov process. Previous PAC-Bayes bound usually does not involve the stochastic modeling of the Markov process. Besides, the bounds is neat and the overall proof logic is clear.

Weakness:
This paper uses a lot of mathematical jargons which may not necessary and hinders the readers to quickly comprehend it. Additionally, some mathematical notations have not been explained clearly before using it. If the theorem can be explained in more plain math no more than the undergrad level, this paper will be more suitable for publications.
1.The unexplained usage of esssup and essinf. Though the introduction of these concepts make the paper more rigorous, it can pose challenges for readers who do not have the background on the measure theory.
2.Is the concept of Gibbs with regard to x is necessary in the paper. Indeed, Gibbs distribution can be defined clearly by the PDF. This creates extra burdens for understanding Theorem 2.7
3.From corollary 3.1, the bound is still relied on the weights theta, which is different from the claim.

Apart from these, with regard to the contents of the paper:
1. Stationary Distribution Assumption: The theory's core relies on the existence of a stationary distribution of the Gibbs form, $p_{\infty} \propto e^{-\beta L_S}$. While this holds for CLD on a bounded domain or with specific regularizers, the existence and form of an asymptotic distribution for practical optimizers like SGD remain unresolved open problems.
2. Inverse Temperature Scaling: To be non-vacuous, most theoretical work requires the inverse temperature to scale as $\beta=O(N)$. This implies that for a large sample size $N$, the injected noise variance $\beta^{-1}$ (from Eq. 10) becomes vanishingly small, potentially rendering it inconsequential to the training dynamics. Conversely, for small $N$, the strong noise required to maintain a non-vacuous bound may severely impair model training and performance.
3. Unconventional Regularization: In the linear regression case study (Appendix E), the authors employ a regularization coefficient $\lambda \propto d$ that scales with the parameter dimension $d$. This deviates from standard machine learning practice, where $\lambda$ is typically a hyperparameter determined via cross-validation. Consequently, the conclusion that "random noise is harmless" derived under this assumption has questionable universality and practical relevance.
4. Loss Function Smoothness: The theoretical derivations, particularly concerning the stationary distribution in Appendices D and H , explicitly require the loss function $L$ to be twice continuously differentiable ( $C^2$ ). However, all experiments were conducted using ReLU activation functions, which result in a loss landscape that is not everywhere differentiable, let alone $C^2$.
5. Bounded Parameter Space: Lemma D. 6 assumes that model parameters are confined to a bounded domain $\Theta$. While this is a standard requirement in many PAC-Bayes and SDE convergence analyses, the experimental description in Appendix F. 2 provides no details on weight initialization or mechanisms (e.g., parameter clipping or projection) that ensure the weights remain within such a bounded set. For non-convex loss functions, unconstrained SGLD offers no guarantee against parameter divergence.
6. Gap Between Theory and Experiments: The provided experiments are overly simplistic and do not adequately bridge the significant gap between the theoretical framework and its practical application.

---

> ### Author Rebuttal · Authors · 2025-07-30
>
> **We thank the reviewer for acknowledging the novelty and neatness of our results, and for the constructive feedback.** We understand the main suggestion of the reviewer is to reduce mathematical jargon, especially related to measure theory (which some undergrads do not study), to make the paper more widely accessible. We agree, and will incorporate the reviewer suggestions into the paper (e.g., add definitions of essup and essinf and exemplify Gibbs in the case it has a PDF). Below we address the additional comments (we will clarify all these in the revised version).
>
> **Weaknesses:**
>
> 1. **Stationary Distribution Assumption: The theory's core relies on the existence of a stationary distribution of the Gibbs form, $p_{\infty} \propto e^{-\beta L_S}$. While this holds for CLD on a bounded domain or with specific regularizers, the existence and form of an asymptotic distribution for practical optimizers like SGD remain unresolved open problems.**
>
>     The exact forms of the stationary distributions do tend to be either very complicated or without an elementary expression (see e.g. [A]). That being said, 1) as shown in [B] the distribution of realistic algorithms can in some cases be approximated by known ones, such as the Gibbs posterior studied in Section 3, and 2) we do not need to derive the full stationary distribution, but only need to evaluate the expected potential at initialization. This may be significantly simpler in some cases. .
>
>     [A] Azizian, Waïss, et al. "What is the long-run distribution of stochastic gradient descent? A large deviations analysis." arXiv preprint arXiv:2406.09241 (2024).
>
>     [B] Raginsky, Maxim, Alexander Rakhlin, and Matus Telgarsky. "Non-convex learning via stochastic gradient langevin dynamics: a nonasymptotic analysis." Conference on Learning Theory. PMLR, 2017.
>
> 2. **Inverse Temperature Scaling: To be non-vacuous, most theoretical work requires the inverse temperature to scale as $\beta = O(N)$. This implies that for a large sample size $N$, the injected noise variance $\beta^{-1}$ (from Eq. 10) becomes vanishingly small, potentially rendering it inconsequential to the training dynamics. Conversely, for small $N$, the strong noise required to maintain a non-vacuous bound may severely impair model training and performance.**
>
>     Note that $\beta = O(N)$ means that $\beta$ must be at most of order $N$, thus making the use of small $\beta$ possible, even for large $N$. Also, we think it is a strength (not a weakness) of our generalization guarantee that it also applies when the noise is too small to affect the training dynamics (i.e., so it effectively applies to gradient flow).  Otherwise, this claim is correct — when the training set is small we require more noise to achieve the same generalization guarantee.
>
> 3. **Unconventional Regularization: In the linear regression case study (Appendix E), the authors employ a regularization coefficient $\lambda \propto d$ that scales with the parameter dimension $d$. This deviates from standard machine learning practice, where $\lambda$ is typically a hyperparameter determined via cross-validation. Consequently, the conclusion that "random noise is harmless" derived under this assumption has questionable universality and practical relevance.**
>
>     First, just to make sure we are on the same page, please note the actual regularization (i.e. the weight decay factor in eq. 10) is not $\lambda$, but $\lambda/\beta$ (i.e. approximately $\lambda/N$), and in a neural network, $\lambda \propto \text{layer width}$, not $d$. Second, as we wrote in the paper, we found empirically this regularization was rather small, and did not have a significant impact on performance. Third, one can always increase the regularisation by modifying the loss $L_S \leftarrow L_S + c ||w||^2$ in equation 10. Under standard initializations, this changes the loss in the bound by $O(c \cdot D)$ factor, where D is the depth of the neural network and so $cD$ is small, for common values of $c$ and $D$. Therefore, combining these observations, we do not see this as a significant practical issue.
>
> 4. **Loss Function Smoothness: The theoretical derivations, particularly concerning the stationary distribution in Appendices D and H, explicitly require the loss function to be twice continuously differentiable ($C^2$). However, all experiments were conducted using ReLU activation functions, which result in a loss landscape that is not everywhere differentiable, let alone $C^2$.**
>
>     Notice there is no issue here: we do not have any continuity/smoothness terms in the results, implying that we can use any $C^2$ approximation to non-$C^2$ functions, e.g. a $C^2$ approximation of ReLU that is identical to ReLU after being quantized to be represented in a computer.
>
> 5. **Bounded Parameter Space: Lemma D. 6 assumes that model parameters are confined to a bounded domain $\Theta$. While this is a standard requirement in many PAC-Bayes and SDE convergence analyses, the experimental description in Appendix F. 2 provides no details on weight initialization or mechanisms (e.g., parameter clipping or projection) that ensure the weights remain within such a bounded set. For non-convex loss functions, unconstrained SGLD offers no guarantee against parameter divergence.**
>
>     See Lemma D.7 that derives the analogous result for unbounded parameters with $\ell^2$ regularization. The experiments in Appendix F were conducted with this form of regularization, instead of the bounded domain assumption.
>
> 6. **Gap Between Theory and Experiments: The provided experiments are overly simplistic and do not adequately bridge the significant gap between the theoretical framework and its practical application.**
>
>     Indeed, our focus in this work was theoretical, to significantly improve previous bounds. We found it remarkable that we were already able to get non-vacous bound without any trajectory-dependent information (as did previous non-vacous bounds), but, as we wrote (in lines 236-237), our bounds still require tightening before they can be practically useful, and we plan to do so in future work.
>
>
> **Questions:**
>
>
> 1. **Bayesian Interpretation of Regularization: Regarding the loss in line 632, can the dimension-dependent regularization term ($\lambda \propto d$) be interpreted from a Bayesian perspective? Does it correspond to a specific prior distribution, which might justify this unconventional choice?**
>
>     The form of regularization is chosen to match the standard initialization (Gaussian, with a variance of 1/(layer width)). The initialization is considered in many works as a Bayesian prior in various settings (e.g., [C,D]).
>
>     [C]  Lee, Jaehoon et al., Deep Neural Networks as Gaussian Processes, 2025.
>
>     [D] Wenger, Jonathan et al., Variational Deep Learning via Implicit Regularization, 2025.
>
> 2. **Negative Generalization Gap: In Table 2, the generalization gap, $E_{D} - E_{S}$, is reported to be negative in some instances. Could the authors provide an explanation for this observation? Is it due to stochasticity in the evaluation on a finite test set, or does it point to another phenomenon?**
>
>     These negative values are not statistically significant, as the standard deviation of the difference of means of $E_S$ and $E_D$, is too large (see the corresponding columns, in parentheses).
>
> 3. **Impact of $N$ and $\beta$ Scaling: How do changes in the sample size $N$ affect the results presented in Tables 2-4? Furthermore, how do $N$ and $\beta$ jointly influence the generalization gap $E_D - E_S$? To better investigate the impact of $\beta$ scaling, we suggest that the authors consider evaluating a more systematic range of values, for example, $\beta \in 0.01N, 0.1N, N, 10N, 100N, 1000N, \infty$.**
>
>     We used the entire training set here. Using smaller $N$ globally improves $E_S$ and degrades $E_D$, $E_{D} - E_{S}$ and the bound. Regarding the $\beta$ values in Tables 2-4, as suggested we added $0.01 N$ and $0.1 N$ to the Tables and they follow the same trends of the other values; note that $N$ and $\infty$ are already included in the tables, while the other values are not significantly different from infinity.
>
> 4. **If the authors can explain this paper clearly in plain math without introducing advanced math jargons, an anonymous link to pdf can be provided (if this is acceptable by the conference regulations). This modifications could also help this paper to have larger impacts if accepted.**
>
>     Unfortunately, no links are allowed. However, the bound on KL (which is used in in-expectation PAC bayes) is easy to derive, e.g. in the case that $p_0=\nu$, which we assume for CLD:
>
>     $$
>     \mathrm{KL}\left(p_{t}||p_{0}\right) =\int p_{t}\log\frac{p_{t}}{p_{0}} =\int p_{t}\log\frac{p_{t}}{p_{\infty}}+\int p_{t}\log\frac{p_{\infty}}{p_{0}} \leq\int p_{0}\log\frac{p_{0}}{p_{\infty}}+\int p_{t}\log\frac{p_{\infty}}{p_{0}} =\int p_{0}\log e^{\Psi}+\int p_{t}\log e^{-\Psi} =E_{p_{0}}\Psi-E_{p_{t}}\Psi \leq E_{p_{0}}\Psi
>     $$
>
>     where in the first inequality we used the second law, and in the next we used our assumption that $p_{\infty}=p_0 e^{-\Psi}/Z$ (there is the main trick, that the troublesome normalization constant $Z$ cancels). We hope this helps to clarify the result. Note that we assume the relevant PDFs exist, and omitted some of the integrals’ notation for brevity.

---

### Decision · Program_Chairs · 2025-09-17

**Decision:**

Accept (spotlight)

**Comment:**

This paper presents a rigorous and insightful theoretical analysis of how temperature influences generalization in Langevin dynamics and other Markovian learning algorithms. The authors show that, under minimal assumptions, higher temperatures can provably reduce generalization error. Remarkably, the resulting bound is independent of training time, mixing assumptions, dimensionality, and gradient norms.

Reviewers found the paper to be technically strong, clearly written, and of high relevance to the NeurIPS community. The theoretical contributions are nontrivial and broadly applicable, and the experimental results, while simple, appropriately validate the core claims.